# WHAT MAKES LOOPED TRANSFORMERS PERFORM BETTER THAN NON-RECURSIVE ONES

## ABSTRACT

While looped transformers (termed as *Looped-Attn*) often outperform standard transformers (termed as *Single-Attn*) on complex reasoning tasks, the theoretical basis for this advantage remains underexplored. In this paper, we explain this phenomenon through the lens of loss landscape geometry, inspired by empirical observations of their distinct dynamics at both sample and Hessian levels. To formalize this, we extend the River-Valley landscape model by distinguishing between U-shaped valleys (flat) and V-shaped valleys (steep). Based on empirical observations, we conjecture that the recursive architecture of *Looped-Attn* induces a ***landscape-level inductive bias*** towards River-V-Valley. Theoretical derivations based on this conjectured inductive bias suggest a better loss convergence along the river due to valley hopping, and further encourage learning about complex patterns compared to the River-U-Valley induced by *Single-Attn*. Building on this insight, we propose **SHIFT** (**S**taged **HI**erarchical **F**ramework for Progressive **T**raining), a staged training framework that accelerates the training process of *Looped-Attn* while achieving comparable performances.

## 1 INTRODUCTION

Transformers (Vaswani et al., 2017) have emerged as a cornerstone across various fields (Devlin et al., 2019; Radford et al., 2019; Liu et al., 2021; He et al., 2022), particularly in Large Language Models (LLMs) (Brown et al., 2020; Achiam et al., 2023). Despite their success, transformers often exhibit challenges in complex reasoning tasks involving arithmetic, commonsense, and symbolic reasoning (Rae et al., 2021; Anil et al., 2022; Wei et al., 2022; Lightman et al., 2023; Ahn et al., 2024). While prompting strategies such as Chain-of-Thought (CoT) have greatly enhanced the reasoning capabilities (Wei et al., 2022; Fu et al., 2022; Chowdhery et al., 2023), the corresponding performances on tasks requiring long reasoning chains are inherently constrained by the fixed-depth transformers (Chen et al., 2025). This limitation motivates the exploration of alternative architectures designed for advanced multi-step reasoning.

It is well-established that standard, non-recursive transformers (Vaswani et al., 2017) (termed as *Single-Attn*) often exhibit a performance plateau on complex problems. This is particularly evident in length generalization issues, where performances of *Single-Attn* drop on sequences longer than those seen during training (Anil et al., 2022; Xiao & Liu, 2023; Jin et al., 2024; Zhou et al., 2024). As an alternative, looped transformers with recursive structure (Dehghani et al., 2018; Lan et al., 2019) (termed as *Looped-Attn*) have demonstrated success on such complex reasoning tasks (Giannou et al., 2023; Fan et al., 2024; Saunshi et al., 2025; Bae et al., 2025). Specifically, *Looped-Attn* deploys recursive self-attention blocks to iteratively refine its internal representations, which helps transformers overcome the performance bottlenecks observed in *Single-Attn*. Although empirical evidence indicates the superiority of *Looped-Attn* over *Single-Attn*, the theoretical understanding of this advantage remains underexplored. This performance gap evidently stems from the recursive mechanism in *Looped-Attn*, but precisely how this structural modification translates into superior reasoning capabilities is still an open question. This motivates the following question:

**What makes looped transformers perform better than non-recursive ones? Specifically, how does the inductive bias from recursion enhance reasoning capabilities?**

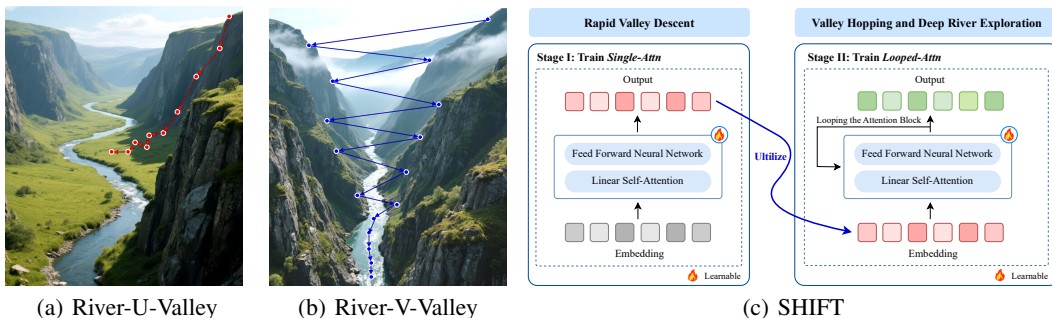

(a) River-U-Valley     (b) River-V-Valley     (c) SHIFT

Figure 1: Loss Landscapes, Optimization Trajectories and SHIFT Strategy.

To theoretically answer this question, we start by empirically investigating the learning processes of *Single-Attn* and *Looped-Attn*. Our investigation examines their behaviors at two levels: a macro-level evaluation of model performance across samples of varying difficulties, and a micro-level examination of the loss landscape's local curvature via Hessian dynamics. These observations reveal two key differences in how *Single-Attn* and *Looped-Attn* learn, which serve as the foundations for our subsequent theoretical analysis. We outline these observations below and provide a detailed discussion in Section 4.1.

---

**Observation 1: Sample-Level Performance**

*(a) Single-Attn. The learning process stops progressing after mastering simple patterns.*
*(b) Looped-Attn. The learning process follows a two-phase curriculum, from simple patterns to complex ones.*

---

**Observation 2: Hessian-Level Dynamics**

*(a) Single-Attn. The eigenspectrum remains relatively static.*
*(b) Looped-Attn. The eigenspectrum undergoes a three-phase evolution: Collapse, Diversification, and Stabilization.*

---

In this paper, we argue that the above two observations potentially originate from the distinct *loss landscapes* induced by different attention architectures. To formalize this, we extend the River-Valley landscape model (Wen et al., 2024) by distinguishing between U-shaped valleys (flat) and V-shaped valleys (steep). Based on this framework, we hypothesize that the *Single-Attn* landscape is dominated by U-shaped valleys, whereas the recursive structure of *Looped-Attn* creates a landscape dominated by V-shaped valleys. This geometric difference accounts for the behaviors observed:

- V-shaped valleys induce a hopping path across valleys, which drives diversification before stabilization of the Hessian eigenspectrum (Observation 2);
- V-shaped valleys might convert hopping to significant progress along the river, which encourages to learn on the complex patterns (Observation 1).

This mechanism comes from the ***landscape-level inductive bias*** of *Looped-Attn*. Figure 1 provides an intuitive illustration, and Sections 4.2∼4.3 detail the formal propositions and theorems.

Furthermore, based on the above understandings, we propose **SHIFT** (**S**taged **HI**erarchical **F**ramework for Progressive **T**raining) that combines *Single-Attn* and *Looped-Attn* to improve the computational efficiency of *Looped-Attn*. Above analysis reveals that both models share the initial phase of mastering simple patterns, and we further demonstrate that their optimization landscapes have a shared river upstream region containing solutions to these patterns. Therefore, SHIFT initially deploys the computationally efficient *Single-Attn* to learn simple patterns, and then switches it to *Looped-Attn*, which enables to explore the river downstream and learn complex patterns. A crucial question remains on when to switch from *Single-Attn* to *Looped-Attn*. We present a **SHIFT Criterion with Patience (SCP)**, established on the performance and optimization stability of *Single-Attn*. Empirical results show that SHIFT achieves reasoning performance comparable to a pure *Looped-Attn* with greater computational efficiency.

Our main contributions are summarized in Appendix A.

## 2  RELATED WORK

**Looped Transformers.**    The principle of recursion in Transformers via cross-layer parameter sharing has been explored in foundational works like Universal Transformers (Dehghani et al., 2018) and ALBERT (Lan et al., 2019). Building on this, looped transformers have demonstrated significant empirical success in complex reasoning (Gao et al., 2024; Bae et al., 2025), such as length generalization capabilities (Giannou et al., 2023; Fan et al., 2024; Saunshi et al., 2025). Theoretical research aiming to understand the advantages of looped transformers can be roughly split into two lines. The first line focuses on expressiveness (Giannou et al., 2023; Gao et al., 2024; Xu & Sato, 2024), showing that looped transformers are Turing complete with universal computational capabilities. The second line analyzes the optimization properties (Gatmiry et al., 2024), proving convergence for linear regression tasks. However, a provable connection between the recursive architecture of looped transformers and the superior reasoning capabilities remains underexplored. Our work addresses this gap by analyzing how the recursive structure shapes the optimization landscape.

**Optimization Landscape and Generalization.**    The geometry of the optimization/loss landscape is fundamental to understanding the training dynamics and generalization capabilities of deep neural networks (Hochreiter & Schmidhuber, 1994; 1997; Li et al., 2021; Lyu et al., 2022; Liu et al., 2023). More recent work has characterized the more complex geometry of the loss landscape, going beyond flat minima. Xing et al. (2018) find that SGD moves in *valley-like* regions of the loss surface to quickly travel far away from the initialization point. Davis et al. (2024) propose that low-loss solutions are not isolated points but lie within connected manifolds, which are defined as *ravines*. Song et al. (2024) characterize the training loss as having an *ill-conditioned-valley-like* structure with a dominant subspace (high curvature) and a bulk subspace (low curvature). This progression culminates in the general *river-valley* theoretical model formulated by Wen et al. (2024), where the river structure is a specific instance of the ravine (Davis et al., 2024) and rooted in the bulk subspace (Song et al., 2024). Building upon this general model, Liu et al. (2025) offer a novel perspective, applying neural thermodynamic laws to understand the river-valley loss landscape. Our work extends the geometry of valleys by *U-shaped* and *V-shaped*, and analyzes these distinct landscapes and training dynamics induced by different architectures.

Additional related work is discussed in Appendix C.

## 3  PRELIMINARIES

This section formalizes the next-token prediction task and specific model architectures.

**Next-token Prediction Task.**    Let the vocabulary $\mathcal{V} = \{1, \cdots, V\}$ be a finite index set of $V$ tokens (*e.g.* words, characters). We consider a training set $\mathcal{T}_N = \{(X^i, y^i)\}_{i=1}^N$ of input sequences $X = [x_1, x_2, \cdots, x_n] \in \mathcal{V}^n$ and target tokens $y \in \mathcal{V}$. Model parameters $\theta$ are trained by minimizing the empirical cross-entropy loss: $\widehat{L}(\theta) = -\frac{1}{N} \sum_{i=1}^N \log\left(\mathsf{S}_{y^i}(\hat{y}^i)\right)$, where $\mathsf{S}_y(\hat{y})$ is the softmax probability of the ground-truth token $y$ given the model's logit output $\hat{y}$. The input sequence $X$ is first mapped to an embedding matrix $E \in \mathbb{R}^{d \times n}$. For theoretical convenience, we consider a simplified setting where the core component for both *Single-Attn* and *Looped-Attn* is a single-layer linear self-attention function $f_\theta$:

$$f_\theta(E, z) = W_V E E^\top W_K^\top W_Q z,$$

where $z \in \mathbb{R}^d$ is a query vector (typically the embedding of the last token) and $W_V, W_K, W_Q \in \mathbb{R}^{d \times d}$ are the value, key, query matrices, respectively.

***Single-Attn* and *Looped-Attn* models.**    The two models are distinguished by how they apply this attention layer. The *Single-Attn* model applies the attention operation once to produce its final state: $z_1 = z_0 + f_\theta(E_0, z_0)$, where $z_0$ is the initial query vector from the input embedding $E_0$. In contrast, the *Looped-Attn* model refines the representation iteratively over $T$ loops. At each step $t \in [T]$, both the query state $z$ and the embedding matrix $E$ for all tokens are updated. We define $E_{t-1}$ as the embedding matrix resulting from the $(t-1)$-th loop. Starting with the initial query state $z_0$ and the input embedding matrix $E_0$, the state update is as follows:

$$z_t = z_{t-1} + f_\theta(E_{t-1}, z_{t-1}).$$

For both models, a final linear head $W_h$ maps the final state ($z_1$ or $z_T$) to the output logits: $\hat{y} = W_h z_1$ for *Single-Attn* and $\hat{y} = W_h z_T$ for *Looped-Attn*. More details are presented to Appendix D.

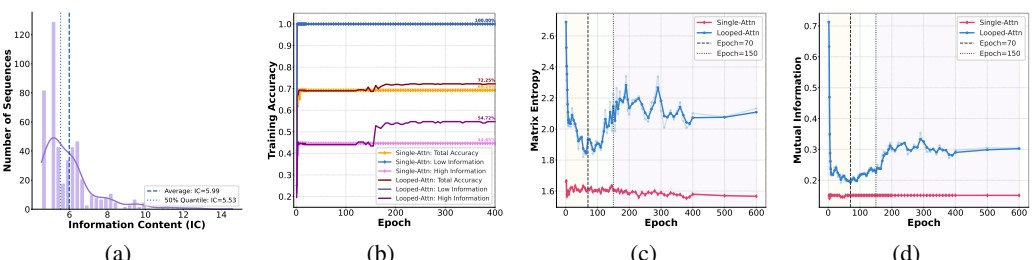

Figure 3: Data Distribution, Task-Level Performance and Hessian-Level Dynamic. **(a)** Long-tail distribution of the dataset shown by Information Content. **(b)** Training accuracy on low information, high information and total sequences. **(c)** Matrix entropy metric. **(d)** Mutual information metric.

## 4 WHAT MAKES LOOPED TRANSFORMERS PERFORM BETTER

This section addresses the fundamental question posed in Section 1. Specifically, we begin by empirical observations of sample-level performances and Hessian-level dynamics (Section 4.1). Motivated by these findings, we introduce two theoretical landscape models, River-U-Valley and River-V-Valley, to characterize landscape-level inductive biases of *Single-Attn* and *Looped-Attn* (Section 4.2). We then present formal theorems and corollaries showing that the River-V-Valley landscape of *Looped-Attn* leads to superior optimization performance (Section 4.3). Finally, we discuss the implications of our theoretical framework for length generalization (Section 4.4).

### 4.1 KEY OBSERVATIONS ON TASK-LEVEL AND HESSIAN-LEVEL

**Experimental Setup.**   We analyze the learning dynamics of two toy models aligned with our theoretical formulation (Section 3): a non-recursive transformer with a single attention layer (*Single-Attn*), a looped transformer consisting of iterating a single attention layer for three loops (*Looped-Attn*). The learning task for both models is to predict the final token $x_3$, given the first three $(x_0, x_1, x_2)$ as input. Detailed experiments are provided in the Appendix E.1. More experimental results on practical models and reasoning tasks are provided in Appendix E.2.

To establish a controllable task difficulty, we design a synthetic Markov language dataset, where each sequence $X$ is generated following a Markov process (Figure 2). The difficulty of predicting a given sequence is quantified by its information content (IC), where $IC(X) = -\log P(X)$.

**Sample-Level Performances.**   To evaluate sample-level performances, sequences are categorized by difficulty using the IC metric into 'low information' (simple; lowest 40%) and 'high information' (complex; highest 40%). The training performances of both *Single-Attn* and *Looped-Attn* are presented in Figure 3(b), with a summary in Observation 1.

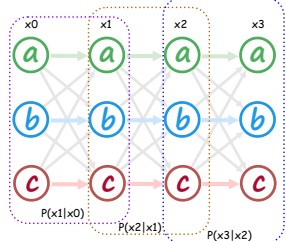

Figure 2: Generation of Markov Language Sequences.

**(a) *Single-Attn*. The learning process stops progressing after mastering simple patterns.** *Single-Attn* exhibits a performance bottleneck. The model rapidly achieves perfect accuracy on low-information sequences. However, its performance on high-information sequences stagnates early in training, showing no subsequent improvement.

**(b) *Looped-Attn*. The learning process follows a two-phase curriculum, from simple patterns to complex ones.** *Looped-Attn* demonstrates a distinct two-phase learning process. In the first 150 epochs, the model masters low-information sequences similar to *Single-Attn*. After epoch 150, it makes significant progress on the high-information sequences, with accuracy rising from $44.65\%$ to $54.72\%$. This dynamic suggests that the recursive architecture exhibits a two-phase learning process, enabling the model to learn more complex patterns.

**Hessian-Level Dynamics.**   To characterize the optimization process, we examine the loss landscape's local curvature through the eigenspectrum $\{\lambda\}$ of Hessian matrix $H$. The evolution of this spectrum is quantified using two information-theoretic metrics: Hessian Matrix Entropy $E(H)$,

which measures landscape diversity or complexity, and Mutual Information $I(H_s; H_{s+1})$, which measures landscape stability between consecutive epoch $s$ and $s + 1$.

$$E(H) = -\sum_i p(|\lambda_i|) \log p(|\lambda_i|),\ I(H_s; H_{s+1}) = \sum_{i,j} p(|\lambda_i|_s, |\lambda_j|_{s+1}) \log \frac{p(|\lambda_i|_s, |\lambda_j|_{s+1})}{p(|\lambda_i|_s) p(|\lambda_j|_{s+1})}.$$

A combined analysis of these two metrics and eigenspectra reveals fundamentally different Hessian-level dynamics for *Single-Attn* and *Looped-Attn*. These findings are presented in Figures 3(c)∼3(d) and Figures 7∼8, with a summary in Observation 2.

**(a)** *Single-Attn.* **The eigenspectrum remains relatively static.** The Hessian eigenspectrum of *Single-Attn* stabilizes almost immediately after training begins. The model rapidly converges to a region where the eigenspectrum is dominated by a spike of near-zero eigenvalues, indicating a relatively flat local geometry (Figures 7(f)∼7(j)). Meanwhile, both Matrix Entropy and Mutual Information metrics keep static (Figures 3(c)∼3(d)). This rapid convergence to a simple geometry suggests that the model fails to explore more regions of the loss landscape after mastering simple patterns, explaining its performance bottleneck.

**(b)** *Looped-Attn.* **Three-phase in eigenspectrum: Collapse, Diversification, and Stabilization.**

**Phase I.** The initial phase involves a collapse of the eigenspectrum, as many eigenvalues shrink toward zero to form a dominant spike (Figures 8(a)∼8(e)). It is also reflected by a significant drop in Matrix Entropy (Figure 3(c)). In this phase, the model moves into a flat region of the landscape, which is a low-dimensional subspace associated with simple patterns. A concurrent decrease in Mutual Information indicates the landscape's variation during this phase (Figure 3(d)).

**Phase II.** Subsequently, the eigenspectrum diversifies as new, larger eigenvalues emerge (Figures 8(f)∼8(j)). It also corresponds to an increase and fluctuation in Matrix Entropy (Figure 3(c)). This activity suggests an exploration of more complex regions along the river. Despite no immediate accuracy gains, the rise in Mutual Information suggests this exploration is a stable search rather than a random process (Figure 3(d)), which makes *Looped-Attn* fundamentally different from *Single-Attn*.

**Phase III.** In the final phase, the eigenspectrum stabilizes (Figures 8(k)∼8(o)). Matrix Entropy converges, indicating that the landscape's geometry has settled (Figure 3(c)). Concurrently, Mutual Information increases to a high plateau, confirming that the landscape's evolution has become stable (Figure 3(d)). This geometric stabilization signifies the arrival at a flatter region, which enables the model to learn complex patterns and ultimately improve its accuracy.

## 4.2 LANDSCAPE-LEVEL INDUCTIVE BIAS

This section extends the River-Valley landscape model by Wen et al. (2024), which formally characterizes the loss landscapes and optimization dynamics suggested by our empirical observations. For a loss function $\widehat{L}(\theta)$ over model parameters $\theta$, the local geometry of loss landscape is captured by its Hessian matrix $H(\theta) = \nabla^2 \widehat{L}(\theta)$. Our analysis focuses on the Hessian eigenspectrum, where $\lambda_i$ denotes its $i$-th largest eigenvalue and $r_i$ or $v_i$ denotes the corresponding eigenvector.

**Definition 1** (River-Valley Loss Landscape)**.** *We define a River-Valley Landscape by specifying two subspaces constructed from the Hessian eigenspectrum with a small threshold $\epsilon > 0$:*

- *River: The river subspace $S_{River}$ is spanned by eigenvectors with eigenvalues below the small threshold: $S_{River} = span\{r_i \mid \lambda_i \leq \epsilon\}$.*
- *Valley: The valley subspace $S_{Valley}$ is spanned by eigenvectors with eigenvalues above the small threshold: $S_{Valley} = span\{v_i \mid \lambda_i > \epsilon\}$.*

*The geometry of valley is further classified by the spectral properties of Hessian restricted to this subspace, denoted $H_{Valley}$, with eigenvalues $\{\lambda_1, \ldots, \lambda_{d_V}\}$. Define condition number as $\kappa(H_{Valley}) = \lambda_1/\lambda_{d_V}$ and Inverse Hessian Average Energy as $\mathcal{E}(H_{Valley}) \triangleq 1/d_V \|H_{Valley}^{-1}\|_F^2 = 1/d_V \sum_{i=1}^{d_V} 1/\lambda_i^2$.*

- *U-shaped Valley (Flat Valley [1]): A valley is U-shaped if it is well-conditioned and has small average energy. With constants $\delta, \zeta \geq 0$: $\kappa(H_{Valley}) \leq 1 + \delta$ and $0 < \mathcal{E}(H_{Valley}) \leq \zeta$.*

---

[1]Here we use 'flat' to represent valleys with uniformly relatively small eigenvalues (U-shaped), and 'steep' to represent valleys with both relatively large and small eigenvalues (V-shaped).

- **_V-shaped Valley (Steep Valley_ [1]_):_** _A valley is V-shaped if it is ill-conditioned and has large average energy. With a constant $\zeta \geq 0$: $\kappa(H_{Valley}) \gg 1$ and $\mathcal{E}(H_{Valley}) \gg \zeta$._

Definition 1 provides a formal characterization of the landscape's features. The river corresponds to directions with near-zero eigenvalues, forming a flat manifold where the loss value changes slowly, while the valley corresponds to directions with large eigenvalues. The geometry within the valley is determined by the condition number of the valley Hessian and inverse Hessian average energy. Specifically, a U-shaped valley is characterized by a broad and flat floor through which the river flows. This valley is surrounded by uniformly steep cliffs, ensuring that movement in any direction within this subspace leads to a comparable loss. In contrast, a V-shaped valley is characterized by a narrow river channel, with cliffs of highly varied steepness. An intuitive illustration is presented in Figure 1. We discuss the hyperparameters and representative loss examples in Appendix E.2.2.

The spectral experiments presented in Figure 27 (with $\epsilon = 0.02$) reveal that _Looped-Attn_ exhibits a larger $\mathcal{E}(H_{\text{Valley}})$ than _Single-Attn_. Building on Definition 1, we then formalize the distinct optimization landscapes and specific dynamics in _Single-Attn_ and _Looped-Attn_ models.

**Conjecture 1** (**Single-Attn: Flat Valley Trapping**). _The Single-Attn model creates a River-U-Valley landscape. After a rapid descent, the optimizer becomes trapped in the valley's broad and flat floor, stopping further exploration within this low-gradient region._

**Empirical Justifications for Conjecture 1.** The River-U-Valley model is empirically supported by the Hessian-level dynamics in _Single-Attn_ (Observation 2). The river component is evidenced by a dominant spike of near-zero eigenvalues from the early epochs, which confirms the existence of a flat subspace. Surrounding this river, large eigenvalues of similar magnitudes form uniformly steep cliffs that enclose a broad and flat floor, characterizing the valley as U-shaped. This landscape geometry is captured by Matrix Entropy and Mutual Information metrics, which indicate a simple and static landscape structure. Such a geometry determines a specific optimization dynamic: the optimizer initially descends rapidly along the steep cliffs. However, the broad and flat valley floor constitutes an optimization trap where weak gradient signals provide insufficient guidance for exploration along the river, resulting in flat valley trapping.

**Conjecture 2** (**Looped-Attn: From Steep Valley Hopping to River Convergence**). _The Looped-Attn model creates a River-V-Valley landscape. The optimizer exhibits significant hopping between the valley's varied and steep cliffs, guiding its trajectory along the river toward convergence._

**Empirical Justifications for Conjecture 2.** The River-V-Valley model is empirically justified by the three-phase evolution of Hessian-level dynamics in _Looped-Attn_ (Observation 2). The model initially enters the river subspace from a complex valley, evidenced by the gradually dominant spike of near-zero eigenvalues. A diversifying set of large eigenvalues forms the V-shaped valley's varied and steep cliffs, where a narrow river channel exists at the valley floor. The complex and evolving geometry is also captured by Matrix Entropy and Mutual Information. Such a geometry leads to a specific optimization dynamic: the optimizer initially descends by hopping between the valleys. After reaching the valley floor, the narrow river channel enables sustained exploration, avoiding getting trapped in the broad U-shaped valley of _Single-Attn_.

### 4.3 River-V-Valley Brings Superior Optimization Performance

In this section, we prove that the River-V-Valley landscape in _Looped-Attn_ provides a superior performance than _Single-Attn_. Before the formal theoretical analysis, we provide an intuition for the connection between loss landscapes and sample-level performances (Observation 1).

**Intuition for Superior Performance.** The River-U-Valley landscape of _Single-Attn_ induces Flat Valley Trapping, which might account for its performance bottleneck. The initial rapid descent along the cliffs converts into progress along the river, corresponding to mastering simple patterns. However, the optimizer subsequently becomes trapped in the flat valley floor, preventing it from discovering the path to more complex patterns. In contrast, the River-V-Valley landscape of _Looped-Attn_ facilitates Steep Valley Hopping dynamics, which might drive its two-phase learning curriculum. After an initial descent for learning simple patterns, its enhanced performance might stem from two key factors: (a) The hopping dynamic converts descent into more forward progress along the river; (b) The narrow river channel prevents the optimizer from becoming trapped. These together ensure deep exploration in the river downstream, enabling the model to learn complex patterns.

We now proceed with a formal analysis to mathematically demonstrate ***how these hopping dynamics lead to more effective optimization***. Our analysis begins by modeling the loss landscape using a structured quadratic form that captures its essential geometry (The general loss is later in Setting 2). The parameter space is decomposed into two orthogonal subspaces: the valley subspace $S_{\text{Valley}} = \text{span}\{v_1, \ldots, v_{d_V}\}$ and the river subspace $S_{\text{River}} = \text{span}\{r_1, \ldots, r_{d_R}\}$, with dimensions $d_V, d_R$, and parameters $\theta_V, \theta_R$ respectively.

**Setting 1** (**Quadratic Loss**). *One simple example of a River-Valley landscape (Definition 1) is the quadratic loss:*

$$\widehat{L}(\theta_V, \theta_R) = \frac{1}{2} \begin{pmatrix} \theta_V \\ \theta_R \end{pmatrix}^\top \begin{pmatrix} H_{\textit{Valley}} & H_{VR} \\ H_{RV} & \mathbf{0} \end{pmatrix} \begin{pmatrix} \theta_V \\ \theta_R \end{pmatrix} - h_R^\top \theta_R,$$

*where $[H_{\textit{Valley}}]_{ij} = \frac{\partial^2 \widehat{L}}{\partial v_i \partial v_j}, [H_{VR}]_{ij} = \frac{\partial^2 \widehat{L}}{\partial v_i \partial r_j}, [H_{RV}]_{ij} = \frac{\partial^2 \widehat{L}}{\partial r_i \partial v_j}$ (Definition 2 in Appendix G.1). We assume the coupling strength along the valley eigenvectors $v_i$ satisfies $\underline{h} \leq \|H_{RV} v_i\| \leq \bar{h}$ for constants $\underline{h}, \bar{h} > 0$, and the valley parameters are initialized as $\theta_{V,0} \sim \mathcal{N}(0, \bar{\alpha}^2 I / d_V)$ with $\|\theta_{V,0}\| \leq \bar{\alpha}$ for a constant $\bar{\alpha} > 0$.*

Setting 1 formalizes a structured quadratic loss, which is characterized by three key components. Specifically, this includes **(a)** The valley Hessian $H_{\text{Valley}}$: This matrix captures the valley's curvature. Its condition number quantitatively distinguishes between the well-conditioned U-shaped valley of *Single-Attn* and the ill-conditioned V-shaped valley of *Looped-Attn*; **(b)** The Coupling Matrix $H_{RV}$: This matrix quantifies the critical interaction that allows movement in the valley to induce a gradient in the river; **(c)** The river gradient $-h_R^\top$: This term represents the intrinsic optimization drive along the river. More details are deferred to Remark 5 in Appendix G.1.

**Theorem 1** (**Cumulative Force under Quadratic Loss**). *Under Setting 1, we define $\mathcal{C}$ as the upper bound of cumulative force $\|C_K\|$ generated by the valley dynamics on the river subspace after $K$ optimization steps, then it holds that*

$$\|C_K\| \approx \left\| \eta \sum_{k=0}^{K-1} H_{RV} \Phi^k \theta_{V,0} \right\| \leq \sqrt{d_V}\, \bar{h}\, \bar{\alpha} \sum_{i=1}^{d_V} \frac{1}{|\lambda_i|} \triangleq \mathcal{C},$$

*where $\Phi = I - \eta H_{\textit{Valley}}$ with a learning rate $\eta$, and $\{\lambda_i\}$ is the spectrum of valley Hessian $H_{\textit{Valley}}$.*

Theorem 1 establishes the relationship between the potential/maximal cumulative force on the river parameters and the valley's geometry, as encoded in the valley eigenvalues $\lambda_i$. The theorem indicates that this force is determined by the nuclear norm of inverse Hessian, alongside a scaling factor of valley dimension.

**Corollary 1** (**Greater Maximal Cumulative Force of *Looped-Attn***). *Under Theorem 1 and Definition 1, the maximal cumulative force generated by Looped-Attn ($\mathcal{C}^{(2)}$) is significantly greater than that of Single-Attn ($\mathcal{C}^{(1)}$): $\mathcal{C}^{(2)} \gg \mathcal{C}^{(1)}$.*

**Theorem 2** (**Expected Squared Cumulative Force under Quadratic Loss**). *Under Setting 1, after a sufficient large $K$ optimization steps, the expected squared cumulative force $\mathbb{E}\left[\|C_K\|^2\right]$ holds that*

$$\frac{\bar{\alpha}^2}{d_V} \underline{h}^2 \sum_{i=1}^{d_V} \frac{1}{\lambda_i^2} \leq \mathbb{E}\left[\|C_K\|^2\right] \leq \frac{\bar{\alpha}^2}{d_V} \bar{h}^2 \sum_{i=1}^{d_V} \frac{1}{\lambda_i^2},$$

*where $\{\lambda_i\}$ is the spectrum of valley Hessian $H_{\textit{Valley}}$.*

**Corollary 2** (**Superior Asymptotic Optimization Performance of *Looped-Attn***). *Under Theorem 2, Definition 1 and Assumption 1, for the same initialization, after a sufficiently large $K$ optimization steps, the expected squared loss values for Looped-Attn ($\widehat{L}_K^{(2)}$) is smaller than for Single-Attn ($\widehat{L}_K^{(1)}$): $\mathbb{E}[(\widehat{L}_K^{(2)})^2] < \mathbb{E}[(\widehat{L}_K^{(1)})^2]$.*

Based on Definition 1, the V-shaped valley of *Looped-Attn* possesses a larger average energy, which creates a larger potential force $\mathcal{C}$ than that of *Single-Attn* (Corollary 1). Furthermore, when taking expectation over initialization, the cumulative force is ultimately reflected in the asymptotic training loss (Corollary 2). The larger force in *Looped-Attn* facilitates sustained river progress via valley hopping, enabling the model to learn both simple and complex patterns. The detailed proof is deferred to Appendix G.2 and Appendix G.3.

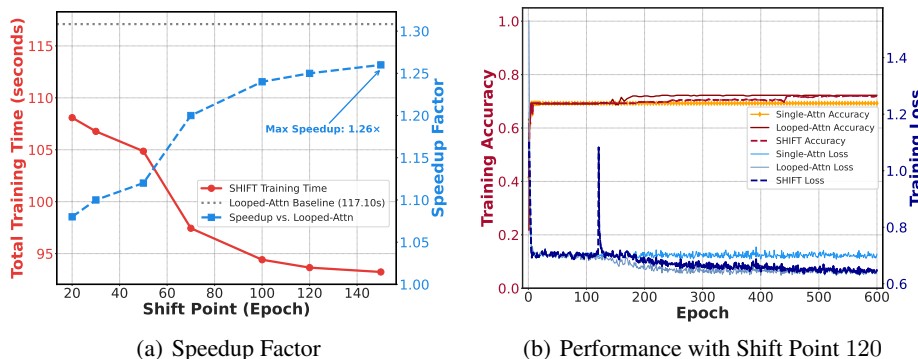

(a) Speedup Factor  (b) Performance with Shift Point 120

Figure 4: SHIFT Efficiency and Performance on Markov Dataset.

**Setting 2** (**General Loss**). *A general Loss of River-Valley landscape (Definition 1) is defined as:*

$$\widehat{L}(\theta_V, \theta_R) = \widehat{L}_{Valley}(\theta_V) + \widehat{L}_{River}(\theta_R) + \widehat{L}_{Coupling}(\theta_V, \theta_R).$$

*We assume the valley parameters are initialized as $\theta_{V,0} \sim \mathcal{N}(0, \bar{\alpha}^2 I / d_V)$ with $\|\theta_{V,0}\| \le \bar{\alpha}$ for a constant $\bar{\alpha} > 0$. Further technical assumptions are detailed in Appendix G.1 (Assumptions 2∼4).*

Setting 2 considers a general loss, which is an extension to Setting 1.

**Theorem 3** (**Superior Optimization Performance of *Looped-Attn* under General Loss**). *Under Setting 2 and Definition 1, the following results hold:*

*(a) **Cumulative Force.** The maximal cumulative force $\mathcal{C}_{gen}$ generated by the valley dynamics on the river subspace is given by: $\mathcal{C}_{gen} = \sqrt{d_V}\,\bar{h}_{gen}\,\bar{\alpha} \sum_{i=1}^{d_V} 1/|\lambda_i^B|$, where $\{\lambda_i^B\}$ is the spectrum of the lower-bound valley Hessian $H^B$ (Assumption 2).*

*(b) **Greater Maximal Cumulative Force.** The maximal cumulative force generated by Looped-Attn ($\mathcal{C}^{(2)}$) is significantly greater than that of Single-Attn ($\mathcal{C}^{(1)}$): $\mathcal{C}^{(2)} \gg \mathcal{C}^{(1)}$.*

*(c) **Lower Asymptotic Training Loss.** For the same initialization and a sufficiently large $K$, after $K$ optimization steps, the expected squared training loss for Looped-Attn ($\widehat{L}_K^{(2)}$) is lower than for Single-Attn ($\widehat{L}_K^{(1)}$): $\mathbb{E}[(\widehat{L}_K^{(2)})^2] < \mathbb{E}[(\widehat{L}_K^{(1)})^2]$.*

Theorem 3 extends the provably superior optimization performance of *Looped-Attn* to a general loss function. The detailed proof of Theorem 3 is deferred to Appendix G.4.

## 4.4 DISCUSSION IN LENGTH GENERALIZATION

This section introduces how our theoretical framework relates to *Looped-Attn*'s success in length generalization. Figure 5 illustrates the Information Content (IC) distributions for the test datasets with different sequence lengths. As length increases, the total space of possible sequences expands, which causes two primary effects on the IC distribution: its mean value shifts to the right (indicating a higher average complexity), and its variance increases (the distribution becomes broader). A direct consequence is that the low-IC sequences during training may become rare or non-existent in longer test sequences, which frames the core challenge of length generalization: a model must find a generalizable solution capable of mastering sufficiently complex patterns.

Empirical performances are provided in Figure 18 and Table 1, and theoretical results provide an explanation for how *Looped-Attn* achieves this. As established in Corollaries 1∼2 and Theorem 3, the River-V-Valley landscape of *Looped-Attn* enables exploration deeper into the downstream river (a manifold of flat minima). Thus it guides *Looped-Attn* towards solutions that inherently generalize better. We connect this to the finding that the superior optimization dynamic brings better performance on length generalization tasks for the *Looped-Attn* model. Detailed experiments are provided in the Appendix E.1.3 and E.2.

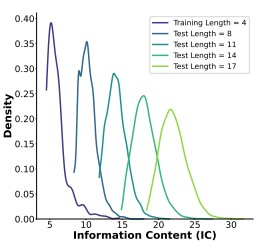

Figure 5: Length Generalization.

## 5 STAGED HIERARCHICAL FRAMEWORK FOR PROGRESSIVE TRAINING

This section proposes SHIFT (Staged HIerarchical Framework for Progressive Training), a computationally efficient two-stage training strategy motivated by our theoretical analysis of River-U-Valley and River-V-Valley landscapes. The strategy utilizes distinct model architectures at different learning stages, as illustrated in Figure 1(c).

**Stage I: Rapid Valley Descent with *Single-Attn*.** Training begins with the *Single-Attn* architecture. The objective is to move efficiently from a random initialization (the clifftop) to a low-loss region (the valley floor). We thus adopt *Single-Attn* which facilitates initial convergence on simple tasks with computational efficiency.

**Stage II: Valley Hopping and Deep River Exploration with *Looped-Attn*.** Training is transitioned to *Looped-Attn* when *Single-Attn* reaches loss plateaus. This transition reshapes the optimization within a V-shaped valley. As established in Corollaries 1~2 and Theorem 3, the V-shaped valley induces a hopping descent mechanism, enabling further exploration in the river direction. This allows the model to find solutions to complex tasks that are less accessible to *Single-Attn*.

A key component of SHIFT is determining the moment to transition between architectures. To this end, we introduce the SHIFT Criterion with Patience (SCP), which consists of two steps.

**(a) Plateau Detection.** First, SCP detects a performance plateau. The validation loss for *Single-Attn* reaches plateaus after initial epochs (Figure 19(a)). The plateau point $E_{\text{plateau}}$ is identified when the validation loss fails to decrease by a threshold $\delta_1$ over $P$ consecutive epochs.

**(b) Gradient Stabilization Wait.** Second, SCP incorporates a patience period $W$ for gradient stabilization. The gradient norm initially exhibits high variance, which would make an unstable transition (Figure 19(b)). This period ensures the optimizer norm has settled by a threshold $\delta_2$. Consequently, the shift point is calculated as $E_{\text{shift}} = E_{\text{plateau}} + W$.

Figure 4(a) reveals that an immediate transition is suboptimal on Markov dataset. A delayed transition yields greater speedup, but an excessive delay prevents *Looped-Attn* from converging in Stage II. To address this trade-off, SCP selects a shift point between 100 and 150 epochs. This achieves a training speedup of approximately $1.26\times$ without compromising final performance (Figure 4(b)). The hyperparameter sensitivity analysis of $\delta_1$, $P$, $\delta_2$ and $W$ are provided in Appendix E.1.4.

We next provide the theoretical foundation for this architectural transition in Theorem 4, by establishing a connection between their landscapes.

**Theorem 4 (Shared River Upstream).** *Let $\nabla_W \widehat{L}_1(\theta)$ and $\nabla_W \widehat{L}_2(\theta)$ be the gradients of the Single-Attn and Looped-Attn models with a weight matrix $W \in \{W_K, W_Q\}$. Under Assumption 5~6 (Appendix H.1.1), the gradients of the two models are positively aligned:*

$$\langle \nabla_{W_K} \widehat{L}_1(\theta), \nabla_{W_K} \widehat{L}_2(\theta) \rangle \geq 0, \quad \langle \nabla_{W_Q} \widehat{L}_1(\theta), \nabla_{W_Q} \widehat{L}_2(\theta) \rangle \geq 0.$$

**Justification for SHIFT.** Theorem 4 ensures the feasibility of this architectural transition. It establishes that the gradients of both architectures are positively aligned, implying that optimization within their respective valleys corresponds to progress along a shared upstream river in the loss landscape. This shared foundation guarantees that the parameters learned by *Single-Attn* in Stage I provide a effective initialization for the deeper exploration by *Looped-Attn* in Stage II. A detailed proof is available in Appendix H. Furthermore, Theorem 1~3 and Corollary 1~2 guarantee the superiority of this two-stage strategy. These results prove that the V-shaped valley of *Looped-Attn* generates a greater cumulative optimization force along the river. Therefore, SHIFT combines the training speed of *Single-Attn* with the superior optimization performance of *Looped-Attn*. In practice, SHIFT is implemented that progressively increases computational depth (*i.e.*, loop iterations from $T = 1$ to $T > 1$). This approach can be viewed as a form of curriculum learning (Bengio et al., 2009; Wang et al., 2021), where an efficient model (*Single-Attn*) first learns simple patterns before a more powerful model (*Looped-Attn*) is deployed for further refinement.

## 6 CONCLUSION

This paper theoretically answers what makes looped transformers perform better than non-recursive ones. We investigate their distinct dynamics and formalize these by extending the River-Valley

model to distinguish between U-shaped valleys and V-shaped valleys. We provably demonstrate that the landscape-level inductive bias of River-V-Valley facilitates superior convergence on complex patterns. Building on this, we propose SHIFT, a framework that achieves comparable reasoning performance compared to *Looped-Attn* but with greater computational efficiency. Overall, our work provides a new perspective and a theoretical framework for understanding the advantages of looped transformers, potentially inspiring more effective and principled training paradigms. More discussions and future work are provided in Appendix B.

## ETHICS STATEMENT

This paper presents a fundamental research focusing on the theoretical and empirical analysis of neural network architectures. Our work is methodological, investigating the mathematical properties of loss landscapes for different types of transformer models. The experiments are conducted on two categories of datasets: (a) a synthetic Markov language dataset, created specifically for controlled analysis of learning dynamics, and (b) publicly available algorithmic reasoning datasets. Our research does not involve the use of human subjects, personally identifiable information, or any form of sensitive data. Therefore, this work does not raise ethical concerns related to data privacy, algorithmic bias in social contexts, or potential societal harm.

## REPRODUCIBILITY STATEMENT

We are committed to ensuring the full reproducibility of our research. To this end, we have provided detailed descriptions of our theoretical frameworks and experimental procedures.

**Theoretical Results.** The theoretical formalization of the River-Valley landscape (Section 4.2) is motivated by empirical observations (Section 4.1). The superiority of *Looped-Attn* (Section 4.3) is supported by mathematical proofs. Detailed derivations for Theorem 1, Corollaries 1∼2 and Theorem 3 are available in Appendix G. The foundation for the SHIFT framework is established in Theorem 4 with proof in Appendix H.

**Experimental Setup.** We provide a comprehensive description of our experimental design. The experimental setup in the synthetic dataset with toy models, including the data generation process, model details, and hyperparameters, is described in Section 4.1 and further detailed in Appendix E.1. The experimental setup for the practical models and the standard algorithmic reasoning tasks is detailed in Appendix E.2.

**Source Code.** To facilitate the verification of our findings and support further research in this area, the source code used for all experiments will be made publicly available upon publication.

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

# Appendix

## A    CONTRIBUTIONS

Our main contributions are summarized as follows.

**(a) A Refined Geometric View of Loss Landscape.**    Inspired by distinct empirical observations in sample-level performance and Hessian-level dynamics (Section 4.1), we enrich the River-Valley landscape model by introducing a geometric characterization of U-shaped and V-shaped Valleys (formal definition in Section 4.2). This characterization is essential for attributing these observations to the landscape-level inductive biases of *Single-Attn* and *Looped-Attn* models.

**(b) Distinct Landscape-Level Inductive Biases.**    To our knowledge, we are the first to formally hypothesize inductive bias of *Looped-Attn* from the perspective of loss landscape. Specifically, in Section 4.2, we reveal that the River-U-Valley landscape of *Single-Attn* leads to flat valley trapping. In contrast, the River-V-Valley landscape of *Looped-Attn* creates an effective path characterized by steep valley hopping and river convergence.

**(c) Theoretical Illustration of Superior Performance in *Looped-Attn*.**    Building upon our findings on inductive bias, we theoretically illustrate the superior performance that would arise from the conjectured River-V-Valley landscape in *Looped-Attn* under the landscape framework (Section 4.3 and Appendix G). Furthermore, we leverage this optimization analysis to explain its strong length generalization ability, empirically demonstrating that the effective optimization path leads to generalizable solutions (Section 4.4).

**(d) An Effective Progressive Training Framework.**    Based on the aforementioned landscape-level inductive biases, we design SHIFT, an intuitive framework that combines *Single-Attn* and *Looped-Attn* (Section 5). The framework's feasibility is grounded in a provable shared river upstream between the two landscapes (detailed proof in Appendix H). We present a shifting criterion with patience (SCP) and demonstrate that SHIFT achieves a balance between computational efficiency and final performance.

## B    DISCUSSIONS AND FUTURE WORK

We present more necessary discussions on our work, which might be helpful for understanding our contributions and existing limitations, and highlight valuable directions for future research.

**Model Simplification.**    Our analysis employs a simplified model with a single linear attention layer for two key purposes: (a) It provides a controlled setting for our experiments to investigate the Hessian dynamics. (b) It ensures the gradient calculations for Theorem 4 (Section 5) are mathematically tractable, which is the theoretical foundation of our SHIFT framework.

It is curial to note that our core theoretical framework is general and does not rely on this specific model architecture. This landscape framework characterizes loss landscapes using River-U-Valley and River-V-Valley to show the optimization advantage of *Looped-Attn* (Sections 4.2 and 4.3). These insights are corroborated by our experiments on GPT-2 based models in Appendix E.2. Although it is hard to directly analyze the Hessian in these practical settings, the superior performance of *Looped-Attn* aligns with the optimization advantage predicted by our River-V-Valley conjecture. We can also explain the training dynamics within our landscape framework, reinforcing its applicability to more complex, non-linear models.

Nevertheless, extending the formal proof of gradient alignment from the simplified model to deep, nonlinear transformers remains a promising direction for future work.

**Landscape Conjectures.**    Conjectures 1∼2 formalize the loss landscapes for *Single-Attn* and *Looped-Attn* by proposing the River-U-Valley and River-V-Valley models. These conjectures are empirically motivated. We justify these with the analysis of Hessian dynamics (Section 4.1), which reveals different evolutionary eigenspectrum of the two architectures. Given the complexity of optimization process, grounding theoretical analysis in empirically-inspired landscape model is a crucial step toward formal understanding  (Wen et al., 2024). A key direction for future work is to move beyond empirical motivation and establish a formal proof for these landscape conjectures. This would involve theoretically deriving the geometric properties of the Hessian from the recursive architecture, potentially by extending emerging mathematical tools such as Dong et al. (2025). Proving this

formally is highly challenging beyond our current scope, which remains a promising direction for future study.

**Landscape Transition Dynamics of SHIFT.** Our landscape model provides a geometric perspective on why the SHIFT framework achieves performance comparable to *Looped-Attn*. Stage I begins with *Single-Attn* in a River-U-Valley landscape, where the optimizer rapidly descends from a high-loss clifftop to a low-loss valley floor near the river. The architectural switch to *Looped-Attn* then induces a geometric transformation: the flat valley floor suddenly becomes the steep slopes of a V-shaped valley. This landscape change forces the optimizer to perform valley hopping which is unique for *Looped-Attn*. This temporary hopping enables it to escape the flat valley floor and reach the narrow river channel. Once in the river, it can proceed with deep downstream exploration. While both models share an upstream river (Theorem 4), their distinct architectures determine the final performance. *Single-Attn* traps in the flat valley floor, whereas SHIFT (*Looped-Attn* in Stage II) successfully navigates downstream, leading to different solutions.

**Practical Implications of SHIFT.** The principles behind SHIFT suggest a promising paradigm for enhancing pre-trained foundation models. We begin with a well-trained standard, non-recursive model (equivalent to Stage I). To improve its performance on tasks requiring complex, multi-step reasoning, we could introduce recursion into some of its blocks and continue to train (equivalent to Stage II). This approach leverages the base model's existing knowledge while reshaping the optimization landscape to unlock more powerful reasoning abilities, guided by the principles of the River-V-Valley. It represents a computationally efficient alternative to training a large recursive model from scratch and offers a valuable direction for future empirical investigation.

## C  ADDITIONAL RELATED WORK

This section provides a more detailed discussion of the related work for Section 2 in the main text.

**Looped Transformers.** The principle of recurrence in Transformers, achieved via cross-layer parameter sharing, has been explored in foundational works like Universal Transformers (Dehghani et al., 2018) and ALBERT (Lan et al., 2019). Building on this, looped transformers have demonstrated significant empirical success in diverse applications, from in-context learning (ICL) (Yang et al., 2023; Chen et al., 2024; Gatmiry et al., 2024) to length generalization that enables them to process sequences much longer than those seen during training (Giannou et al., 2023; Fan et al., 2024; Gao et al., 2024; Saunshi et al., 2025; Bae et al., 2025).

Theoretical research aiming to understand these empirical advantages can be roughly split into two lines. The first line focuses on expressiveness (Giannou et al., 2023; Gao et al., 2024; Xu & Sato, 2024), showing that looped transformers are Turing complete with universal computational capabilities. The second line analyzes the optimization properties (Gatmiry et al., 2024), proving optimization convergence for linear regression tasks within the ICL framework. However, a provable connection between the recursive architectural prior of looped transformers, optimization landscape, and superior reasoning capabilities remains missing, particularly under the general next-token prediction paradigm. Our work addresses this gap by analyzing how the recursive structure shapes the optimization landscape, ultimately seeking to combine the length generalization benefits of looped transformers with the efficiency of standard, non-recursive models.

**Optimization Landscape and Generalization.** The geometry of the optimization/loss landscape is fundamental to understanding the training dynamics and generalization capabilities of deep neural networks. Empirically, Hochreiter & Schmidhuber (1994; 1997) first demonstrate that SGD can typically find flat minima among various solutions. Theoretically, much research has provided strong evidence supporting this idea, reporting that models converging to flat minima exhibit better generalization performance across various tasks and architectures (Keskar et al., 2016; Wu et al., 2017; Neyshabur et al., 2017; Kleinberg et al., 2018; Xie et al., 2020; Li et al., 2021; Lyu et al., 2022; Andriushchenko et al., 2023; Liu et al., 2023).

More recent work has characterized the more complex geometry of the loss landscape, going beyond flat minima. Xing et al. (2018) find that SGD moves in ***valley-like*** regions of the loss surface to quickly travel far away from the initialization point. Davis et al. (2024) propose that low-loss solutions are not isolated points but lie within connected manifolds, which are defined as ***ravines***. Song et al. (2024) characterize the training loss as having an ***ill-conditioned-valley-like*** structure with a

dominant subspace (high curvature) and a bulk subspace (low curvature). This progression culminates in the general ***river-valley*** theoretical model formulated by Wen et al. (2024), where the river structure is a specific instance of the ravine (Davis et al., 2024) and rooted in the bulk subspace (Song et al., 2024). Building upon this general model, Liu et al. (2025) offer a novel perspective, applying neural thermodynamic laws to understand the river-valley loss landscape. Our work extends the geometry of valleys by ***U-shaped*** and ***V-shaped***, and analyzes these distinct landscapes and training dynamics induced by different architectures.

These two perspectives, flat minima and river-valley landscapes, are highly compatible. We argue that the river downstream locates flatter minima, which is potentially corresponding to better generalization (Hochreiter & Schmidhuber, 1994; 1997).

**Inductive Bias.** Implicit bias and inductive bias are fundamental concepts in deep learning theory. Implicit bias is an emergent property of the optimization algorithm (*e.g.*, gradient descent) that guides the model toward a particular minimum that does generalize well (Soudry et al., 2018; Gunasekar et al., 2018a; Ji & Telgarsky, 2019; Woodworth et al., 2020; HaoChen et al., 2021; Ataee Tarzanagh et al., 2023; Tarzanagh et al., 2023; Thrampoulidis, 2024). In contrast, inductive bias is induced by the model architecture. For example, weight sharing and locality inherently bias convolutional neural networks (CNNs) over fully-connected networks (FCN) by breaking the learning algorithm's symmetry (Gunasekar et al., 2018b; Li et al., 2020; Jagadeesan et al., 2022; Wang & Wu, 2023). Jelassi et al. (2024) reveal an inductive bias in transformers that makes it easier for them to copy from the context. Saunshi et al. (2024) uncover an inductive bias of stacking for improving downstream reasoning tasks, but without a theoretical basis. Gatmiry et al. (2024) also study looped transformers, showing their inductive biases in optimization convergence for linear regression tasks. Distinct from above, we introduce ***landscape-level inductive bias***, where the model architecture fundamentally reshapes the optimization landscape (River-U-Valley and River-V-Valley). These different landscapes induce unique training dynamics. From this perspective, we reveal the advantages of *Looped-Attn* over *Single-Attn* supported by both empirical observations and theoretical analysis (Section 4).

# D  DETAILED PRELIMINARIES

This section provides more details for Section 3 in the main text.

We formalize the next-token prediction task, specify the objective function, and present the mathematical characterizations of *Single-Attn* and *Looped-Attn* models.

Let the vocabulary $\mathcal{V} = \{1, \cdots, V\}$ be a finite index set of $V$ tokens (*e.g.* words, characters). An input sequence is denoted by $X = [x_1, x_2, \cdots, x_n] \in \mathcal{V}^n$, where each token $x_s \in \mathcal{V}$. The task is to predict the next token, $y \in \mathcal{V}$, given the context $X$. We consider a training set of $N$ sequences $\mathcal{T}_N := \{(X^i, y^i)\}_{i=1}^N$, where $X^i \in \mathcal{V}^n$ and $y^i \in \mathcal{V}$ for all $i \in [N]$. A model with parameter $\theta$ is trained by minimizing the empirical cross-entropy loss. Let $\hat{y} \in \mathbb{R}^V$ be the logit vector output by the model, then the loss function is defined as:

$$\widehat{L}(\theta) = -\frac{1}{N} \sum_{i=1}^N \log\left(\mathsf{S}_{y^i}(\hat{y}^i)\right) = \widehat{\mathbb{E}}\left[-\log\left(\mathsf{S}_y(\hat{y})\right)\right],$$

where $\mathsf{S}_y(\hat{y}) = \exp(\hat{y}_y)/\sum_{j=1}^V \exp(\hat{y}_j)$ denotes the softmax probability for the ground-truth token $y$, with $\hat{y}_y$ being the $y$-th component of the logit vector $\hat{y}$.

**Input Embeddings and Self-Attention Module.**  The input sequence $X$ is mapped to $d$-dimensional embedding matrix $E$ via an embedding map $g : \mathcal{V}^n \to \mathbb{R}^{d \times n}$ parameterized by $\theta_{\text{emb}}$, so that $E = g(X; \theta_{\text{emb}})$. We assume that $g$ is fixed (*i.e.*, not trainable) and focus our analysis on the self-attention module.

Both *Single-Attn* and *Looped-Attn* utilize a fundamental self-attention function $f_\theta$, implemented as a single-layer linear attention block (without residual connections), defined as:

$$f_\theta(E, z) = W_V E E^\top W_K^\top W_Q z, \quad f_\theta(E) = W_V E E^\top W_K^\top W_Q E,$$

where $E \in \mathbb{R}^{d \times n}$ is the embedding matrix, $z \in \mathbb{R}^d$ is the query vector, *i.e.,* the $n$-th column of $E$, and $W_V, W_K, W_Q \in \mathbb{R}^{d \times d}$ are the value, key, query matrices, respectively.

**Single-Attn Model and Looped-Attn Model.**  The *Single-Attn* model applies the self-attention operation once, then

$$z_1 = z_0 + f_\theta(E_0, z_0),$$

where $z_0$ is the $n$-th column of the input embedding matrix $E_0$ and $z_1$ is the final state.

The *Looped-Attn* model iteratively refines representations over $T$ steps. For each loop $t \in [T]$, the representations are updated via residual connections and gating mechanisms:

$$z_t = z_{t-1} + f_\theta(E_{t-1}, z_{t-1}), \quad E_t = E_{t-1} + f_\theta(E_{t-1}).$$

We have the recursive definition for the final state $z_T$ after $T$ loop iterations, *i.e.,*

$$z_T = z_0 + \sum_{t=1}^T f_\theta(E_{t-1}, z_{t-1}).$$

**Prediction Head.**  The final logit output $\hat{y} \in \mathbb{R}^V$ is generated by a linear projection head $h : \mathbb{R}^d \to \mathbb{R}^V$, parameterized by $W_h \in \mathbb{R}^{V \times d}$. Finally, the output logits are $\hat{y} = W_h z_1$ for *Single-Attn* and $\hat{y} = W_h z_T$ for *Looped-Attn*.

## E  DETAILED EXPERIMENTS

### E.1  EXPERIMENTS ON TOY MODELS AND SYNTHETIC MARKOV LANGUAGE DATASET

#### E.1.1  EXPERIMENTAL SETUP

**Toy Models and Hyperparameter Details.**  To conduct the motivating experiments and investigate the learning dynamics of different architectures, we employ simplified toy models. Specifically, we adopt a non-recursive transformer with a single attention layer (*Single-Attn*), and a looped transformer consisting of iterating a single attention layer for three loops (*Looped-Attn*). These toy models are aligned with our theoretical formulation in Section 3. We train both models for 600 epochs, using Adam optimizer with the learning rate 0.001. Each experiment is conducted on a single 24GB NVIDIA GeForce RTX 3090.

**Markov Language Dataset.**  We utilize a synthetic Markov language dataset, specifically designed to provide a controllable spectrum of task difficulty. As illustrated in Figure 2, each sample is a sequence of four tokens, $X = (x_0, x_1, x_2, x_3)$ (*e.g.*, 'aaaa','aaab','abbc'), drawn from a vocabulary of three discrete symbols $\{a, b, c\}$. The sequences are generated according to a homogeneous Markov process, where the probability of a full sequence is given by

$$P(X) = P(x_0)P(x_1|x_0)P(x_2|x_1)P(x_3|x_2).$$

The initial state probabilities $P(x_0)$ are uniform, while the transition probabilities at each step are governed by three distinct, randomly generated transition matrices.

The learning task for both *Single-Attn* and *Looped-Attn* is to predict the final token $x_3$, given the first three $(x_0, x_1, x_2)$ as input. We quantify the difficulty of each prediction by the information content (IC) of its corresponding ground-truth sequence:

$$IC(X) = -\log P(X).$$

To create a dataset with a mixture of simple and complex tasks, we begin by generating all $3^4$ possible sequences. The initial set is then expanded to a larger dataset size of $N = 500$ through a weighted oversampling process. This sampling probability for each sequence is proportional to its ground-truth probability raised to the power of 2. This ensures that high-probability (low-information, or simple) sequences are sampled more frequently, resulting in a long-tail training distribution, as shown in Figure 6. Consequently, simple patterns are abundant while complex patterns are rare, posing a generalization challenge.

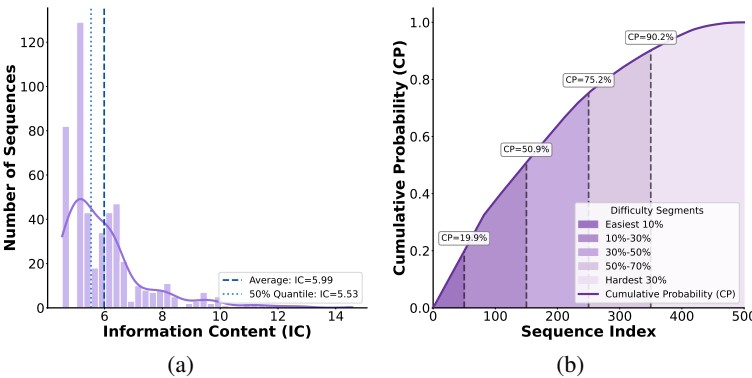

Figure 6: Data Distribution. **(a,b)** Long-tail distribution of the dataset shown by IC and CP.

#### E.1.2  EMPIRICAL OBSERVATIONS

By combining two information-theoretic metrics (Hessian Matrix Entropy and Mutual Information) with a direct analysis of the eigenspectrum, we investigate different Hessian-level dynamics for *Single-Attn* and *Looped-Attn*.

**More Discussion on Hessian-Level Dynamics.** The metrics of matrix entropy and mutual information based on Hessian *w.r.t.* the value matrix $W_V$, are presented in Figures 3(c)~3(d). Regarding Figure 3(d), it is important to understand that we cannot directly compare the absolute values of Mutual Information (MI) for *Single-Attn* and *Looped-Attn*. This is because they have a different baseline level of Matrix Entropy. In information theory, the mutual information between two random variables is fundamentally bounded by the entropy of each variable. Specifically, we have $I(H_s; H_{s+1}) \leq \min(E(H_s), E(H_{s+1}))$. This means that the absolute values of MI is limited by the complexity of landscape itself, as measured by Matrix Entropy.

This helps explain the low final MI value for *Single-Attn*. Even though the state at epoch $s + 1$ is similar to the state at epoch $s$, the overall landscape is simple (low entropy) thus the absolute MI value remains small. However, notice that both models ultimately reach a stable state of high MI within the limits set by its own entropy. It represents a stagnation, not exploration.

**Eigenspectra of Hessian *w.r.t.* the Value Matrix $W_V$.** We present the eigenspectra of Hessian with respect to (*w.r.t.*) the value matrix $W_V$ in Figure 7~11 for three models: *Single-Attn*, *Looped-Attn* and *Deep-Attn* (a non-recursive transformer with three attention layers).

We find that the spectral shape and evolution of *Single-Attn* (Figure 7) and *Deep-Attn* (Figure 9~11) are nearly identical. Both converge to a simple and static landscape, and their valley eigenspectra contain uniformly relatively small eigenvalues, with maximum eigenvalues of a similar small magnitude (*e.g.*, $\lambda_{\max} \approx 0.83$ for *Single-Attn* and $\lambda_{\max} \approx 0.28$ for *Deep-Attn* Layer 1). Based on Definition 2, both *Single-Attn* and *Deep-Attn* create River-U-Valley landscapes. In contrast, *Looped-Attn* (Figure 8) exhibits the distinct three-phase evolution. Its valley eigenspectra contain both relatively large and small eigenvalues, with a significantly larger $\lambda_{\max} \approx 2.84$. Based on Definition 2, *Looped-Attn* creates a River-V-Valley landscape.

This comparison demonstrates that the River-V-Valley landscape is a unique inductive bias of the recursive architecture, not simply a product of computational depth.

**Eigenspectra of Hessian *w.r.t.* the Key Matrix $W_K$.** The metrics of matrix entropy and mutual information based on Hessian *w.r.t.* the key matrix $W_K$, are presented in Figure 12. We present the eigenspectra of Hessian with respect to (*w.r.t.*) the key matrix $W_K$ in Figure 13~17 for three models: *Single-Attn*, *Looped-Attn* and *Deep-Attn*.

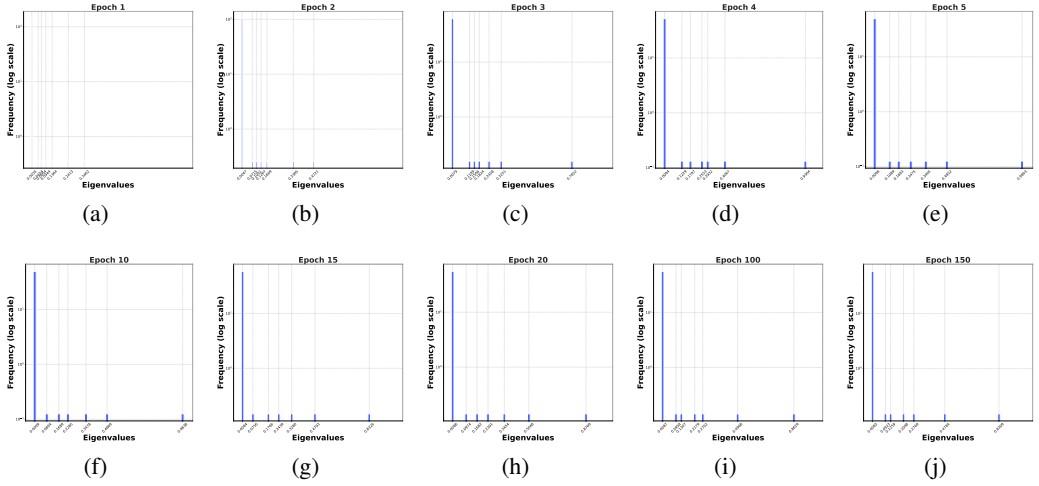

Figure 7: Single-Attn Eigenspectra (Hessian *w.r.t.* the Value Matrix $W_V$).

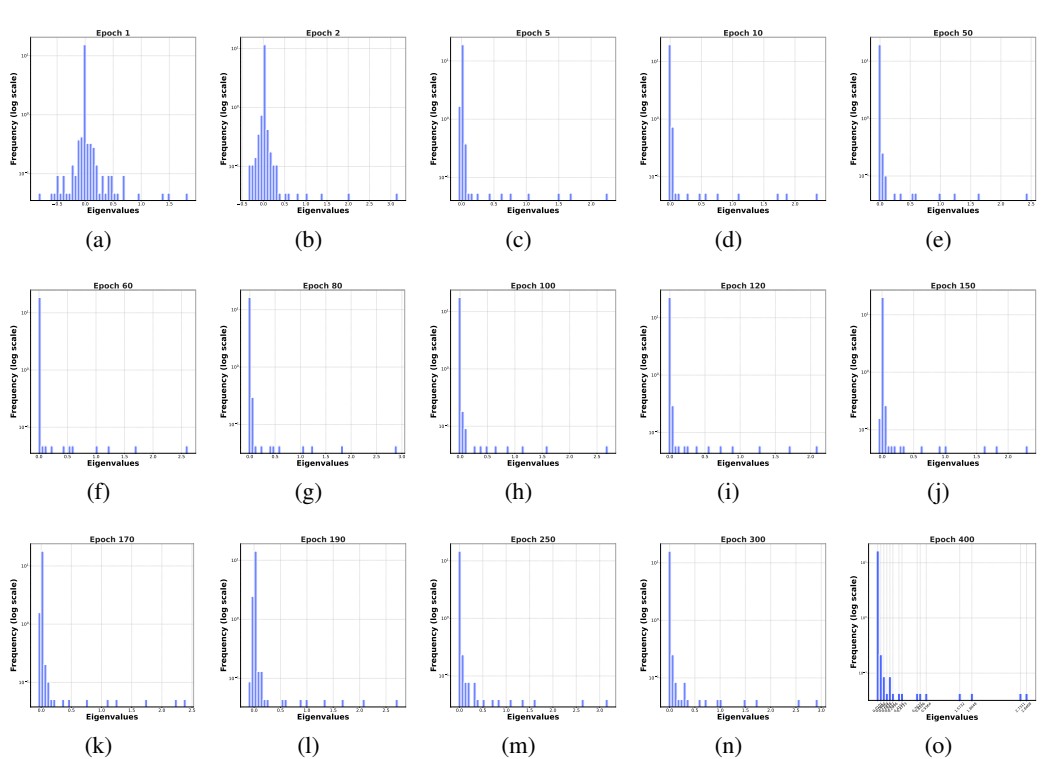

Figure 8: Looped-Attn Eigenspectra (Hessian *w.r.t.* the Value Matrix $W_V$).

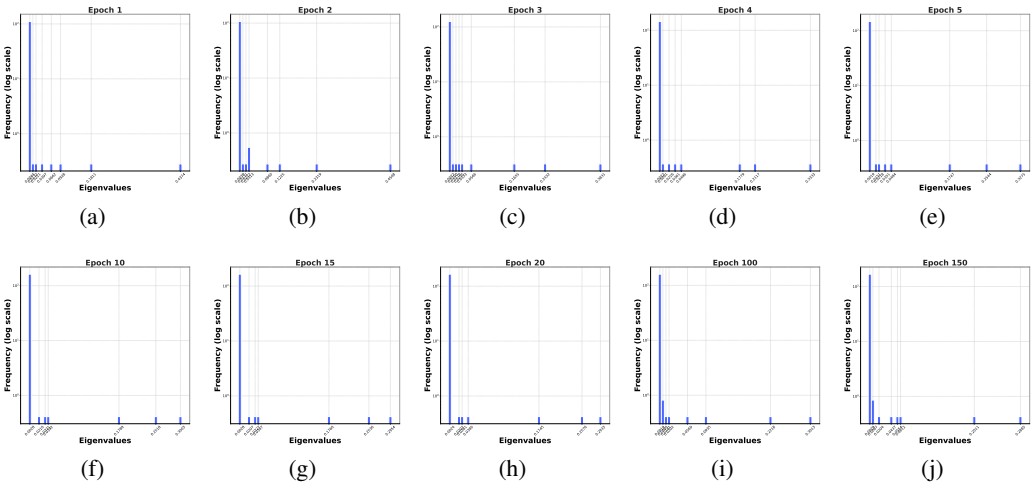

Figure 9: Deep-Attn Eigenspectra (Hessian *w.r.t.* the Value Matrix $W_V$ in Layer 1).

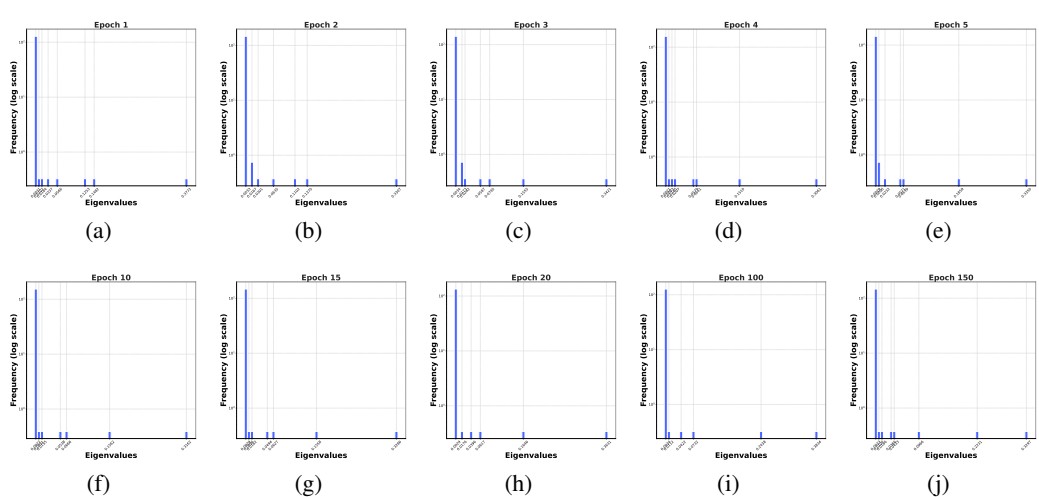

Figure 10: Deep-Attn Eigenspectra (Hessian *w.r.t.* the Value Matrix $W_V$ in Layer 2).

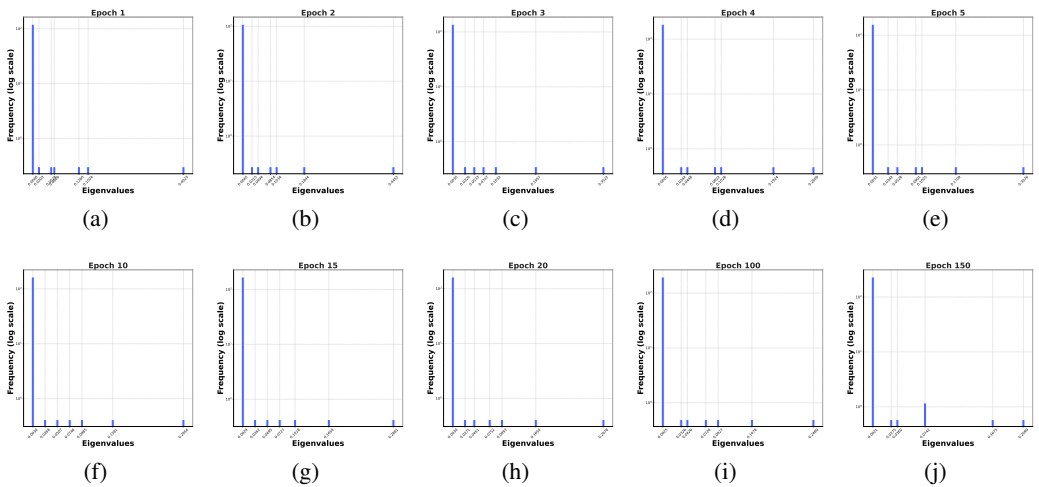

Figure 11: Deep-Attn Eigenspectra (Hessian *w.r.t.* the Value Matrix $W_V$ in Layer 3).

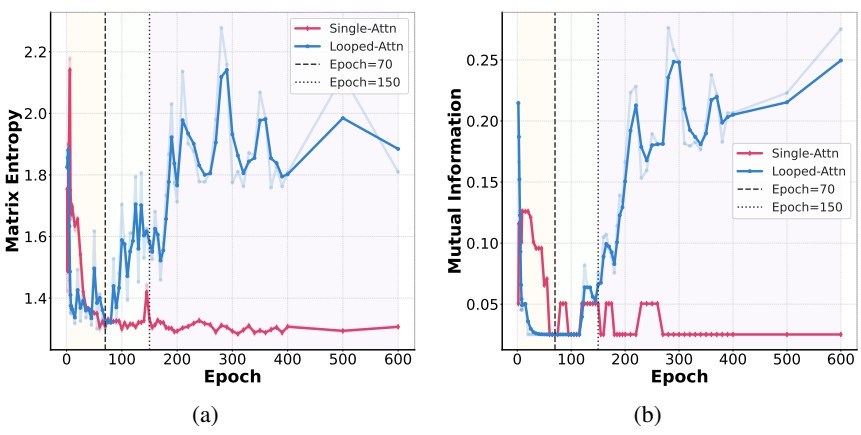

Figure 12: **(a)** Matrix Entropy metric. **(b)** Mutual Information Metric.

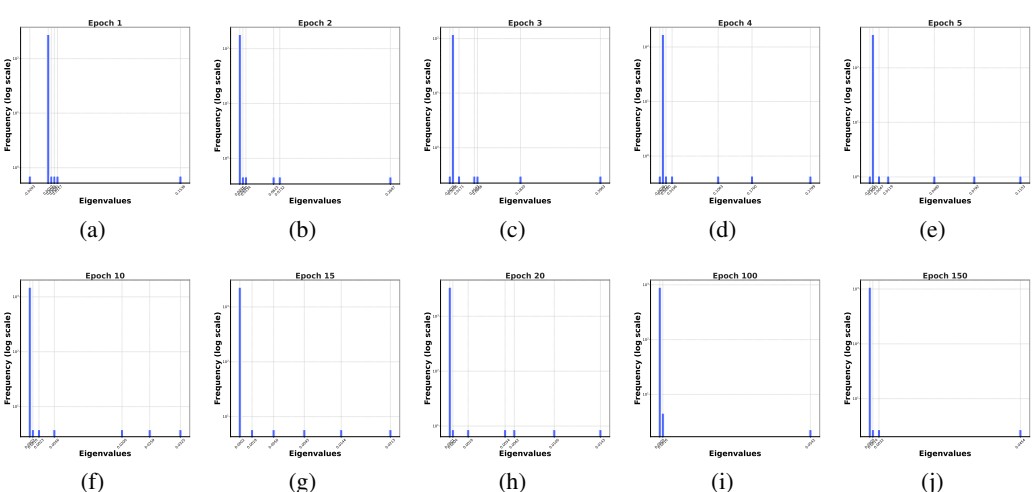

Figure 13: Single-Attn Eigenspectra (Hessian *w.r.t.* the Key Matrix $W_K$).

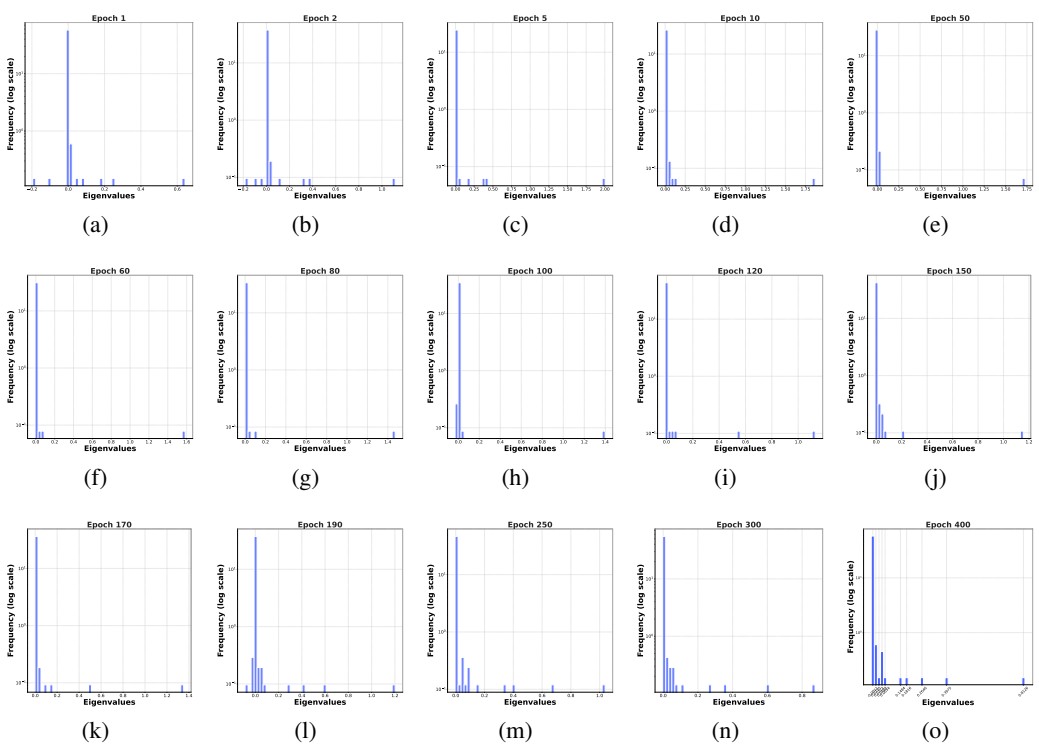

Figure 14: Looped-Attn Eigenspectra (Hessian *w.r.t.* the Key Matrix $W_K$).

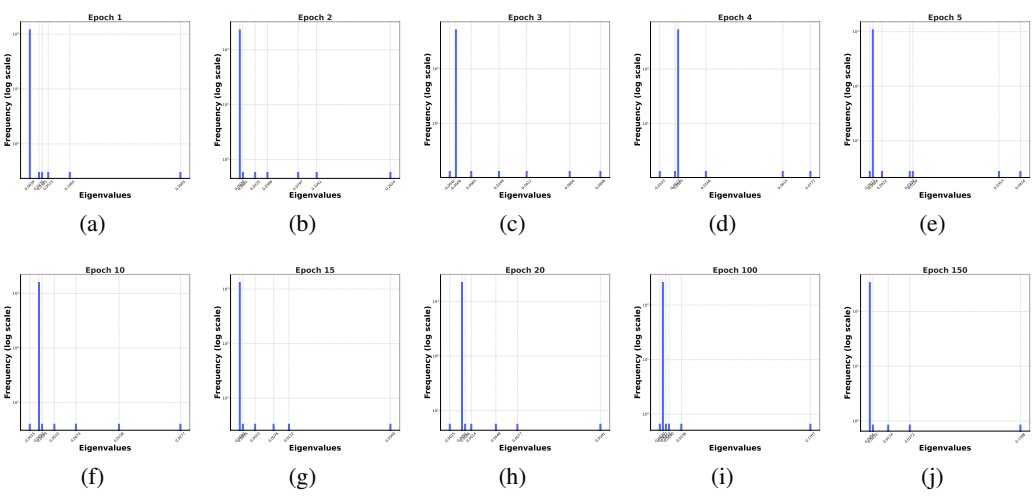

Figure 15: Deep-Attn Eigenspectra (Hessian *w.r.t.* the Key Matrix $W_K$ in Layer 1).

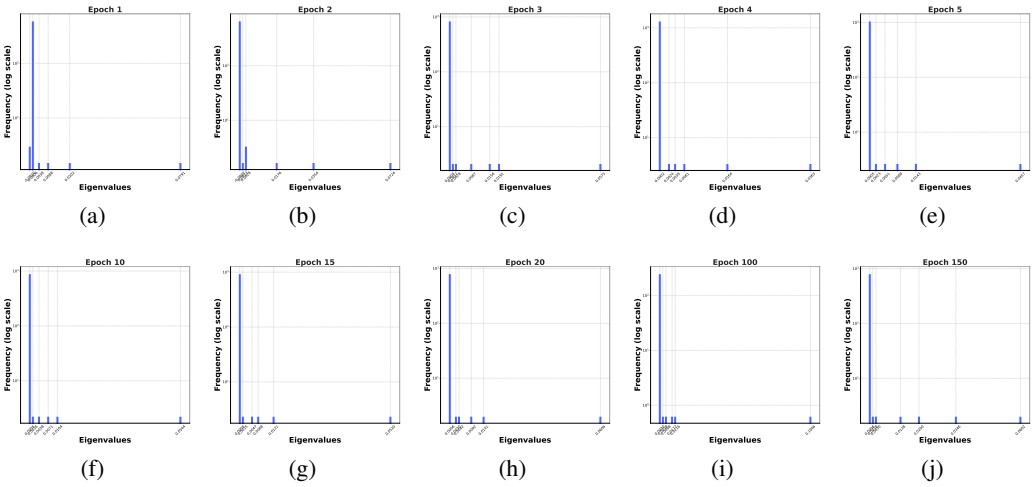

Figure 16: Deep-Attn Eigenspectra (Hessian *w.r.t.* the Key Matrix $W_K$ in Layer 2).

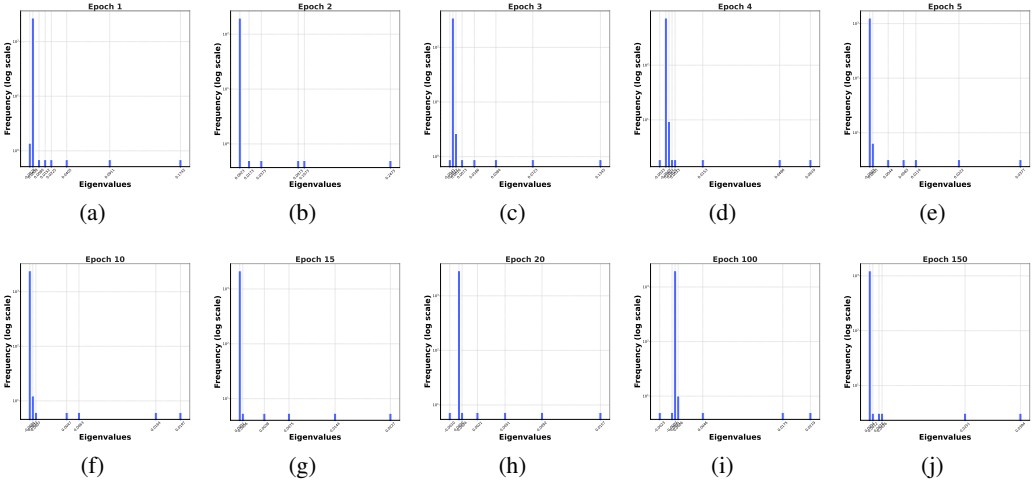

Figure 17: Deep-Attn Eigenspectra (Hessian *w.r.t.* the Key Matrix $W_K$ in Layer 3).

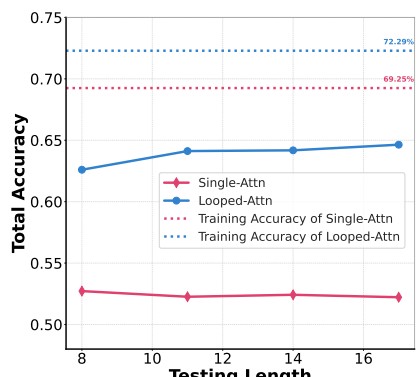

Figure 18: Length Generalization Performances.

Table 1: Accuracy on Relatively Simple Sequences.

| Datasets | Sequence Length | # Simple Sequences | *Single-Attn* | *Looped-Attn* |
|---|---|---|---|---|
| **Training** | $L = 4$ | 100% | 69.25% | 72.29% |
| **Testing** | $L = 8$ | 99.5% | 52.83% | 62.78% |
| | $L = 11$ | 64.5% | 55.57% | 70.28% |
| | $L = 14$ | 0 | N/A | N/A |
| | $L = 18$ | 0 | N/A | N/A |

### E.1.3 LENGTH GENERALIZATION

To bridge optimization with generalization, we design a controlled experiment on the synthetic Markov language dataset to evaluate the length generalization capabilities of the *Single-Attn* and *Looped-Attn* models.

**Testing Datasets.** We generate a series of test datasets with sequence lengths $L \in \{8, 11, 14, 17\}$. To specifically isolate the challenge of generalizing a learned rule to longer sequences, rather than adapting to entirely new dynamics (where our designed simplified Single-Attn and Looped-Attn might be completely failed), we generate all test datasets using the same transition dynamics $\{T_1, T_2, T_3\}$ employed for the training data. For sequence lengths $L > 4$, the transition matrices are applied cyclically. Furthermore, to ensure consistent evaluation across lengths, each dataset is generated by sampling a fixed number of $N_{\text{test}} = 5000$ sequences, following the same long-tail sampling rules ($\alpha = 2$) as the training dataset. With these rules, we present the Information Content (IC) distributions for the test datasets with different sequence lengths in Figure 5.

**Evaluation Metrics.** We analyze model performance based on the IC of each sequence. This allows us to distinguish between simple (low-IC) and complex (high-IC) tasks. Based on the IC distribution of the training data ($L = 4$), we establish a fixed complexity threshold $IC = 14.57$, which represents the maximum IC in the training sequences. We then evaluate both models on the following metrics:

- **Total Accuracy:** The accuracy on the total test datasets.
- **Accuracy on Relatively Simple Sequences:** The accuracy on the subset of test sequences with an IC below the fixed threshold ($IC \leq 14.57$).

Figure 18 and Table 1 present the length generalization performance of the *Single-Attn* and *Looped-Attn* models. We find that:

**(a) Total Accuracy.** As shown in Figure 18, *Looped-Attn* significantly outperforms *Single-Attn* on out-of-distribution testing datasets with sequence lengths greater than the training length. This performance gap confirms that the inductive bias of *Looped-Attn* leads to a more generalizable so-

lution, aligning with our theoretical findings that its optimization landscape guides toward a more flatter minimum.

An interesting observation from Figure 18 is that the accuracy of both models does not strictly decrease as testing length increases (and even increases slightly). This phenomenon originates from our specific design which employs cyclic transitions. In this setup, a longer sequence provides the model with more in-context examples of the underlying repeating rule. This may temporarily counteract the performance drops from increasing complexity. However, we point out on more general datasets, a clearer trend of performance dropping with increasing sequence length would be observed (Fan et al., 2024). Here, we focus more on the consistently superior performance of *Looped-Attn* over *Single-Attn*.

**(b) Accuracy on Relatively Simple Sequences.** The '# Simple Sequences' column reveals a critical length generalization challenge: the low-IC sequences during training become rare or non-existent in longer test sequences. This confirms that longer sequences are inherently more complex.

We consider the accuracy on these relatively simple sequences. Specifically, at $L = 11$ where a significant portion of simple sequences still exists, *Looped-Attn* maintains a higher accuracy compared to *Single-Attn*. This indicates that *Single-Attn* struggles to apply its knowledge even to tasks of comparable complexity when the sequence is longer. In contrast, *Looped-Attn* generalizes better to longer sequences. This aligns with our theory that *Looped-Attn* finds a more generalizable solution by exploring further into the river downstream with flat minima.

### E.1.4 SHIFT CRITERION WITH PATIENCE

**Motivation for SCP Design.** As discussed in Section 5, Figure 4 empirically validates the motivation behind SCP by illustrating the trade-off between computational efficiency and reasoning accuracy. Specifically, Figure 4(a) reveals that while a delayed transition increases the speedup factor, an excessive delay prevents *Looped-Attn* from converging in Stage II. In Figure 4(b), we visualize the training dynamics at a specific shift point (Epoch 120) to compare SHIFT, *Single-Attn*, and *Looped-Attn*. These experiments indicate that relying solely on the loss plateau is insufficient for determining the optimal transition timing. Since *Single-Attn* exhibits a long loss plateau, it is difficult to identify a precise moment that balances accuracy and efficiency based on loss alone. This observation motivates the design of the second stage of SCP.

**Hyperparameter Sensitivity of $\delta_1$, $P$, $\delta_2$ and $W$.** We conduct a detailed sensitivity analysis of the SCP criterion's hyperparameters. Specifically, the baseline configuration is established at $\delta_1 = 0.001$, $P = 10$, $\delta_2 = 0.03$ and $W = 5$, with experimental ranges in $\delta_1 \in \{0.0001, 0.0005, 0.001, 0.005, 0.01, 0.05, 0.1, 0.5\}$, $P \in \{5, 6, 7, 8, 9, 10, 15, 20\}$, $\delta_2 \in \{0.025, 0.03, 0.035, 0.04, 0.045, 0.05\}$ and $W \in \{3, 4, 5, 6, 7\}$.

As shown in Figure 20, for the Plateau Detection phase, the model exhibits robustness with the shift point consistently stabilizing around epoch 119 regardless of variations in the loss threshold $\delta_1$ and patience $P$. For the Gradient Stabilization Wait phase, a larger gradient norm threshold $\delta_2$ relaxes the stability constraint, resulting in earlier transitions. To maximize total training efficiency, we recommend selecting $\delta_2$ slightly above the intrinsic gradient norm rather than using arbitrarily loose thresholds. The window $W$ serves primarily to filter out single-step stochastic outliers. We advise against setting $W$ too large unless the gradient curve is exceptionally smooth.

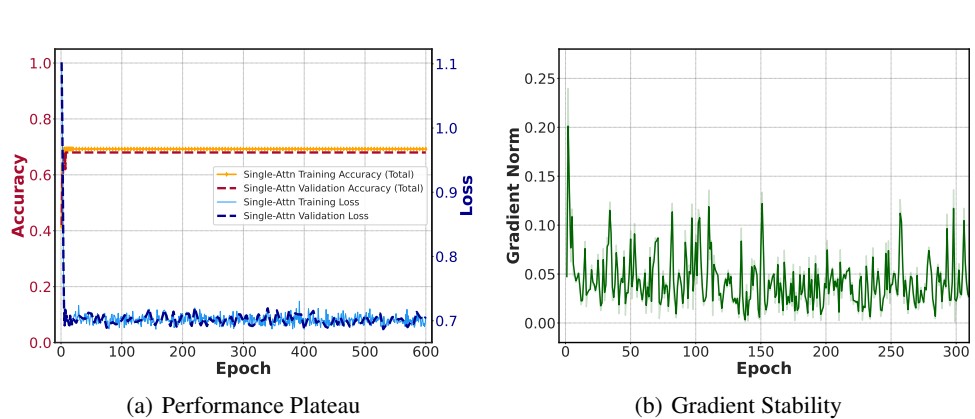

(a) Performance Plateau      (b) Gradient Stability

Figure 19: SHIFT Criterion with Patience (SCP).

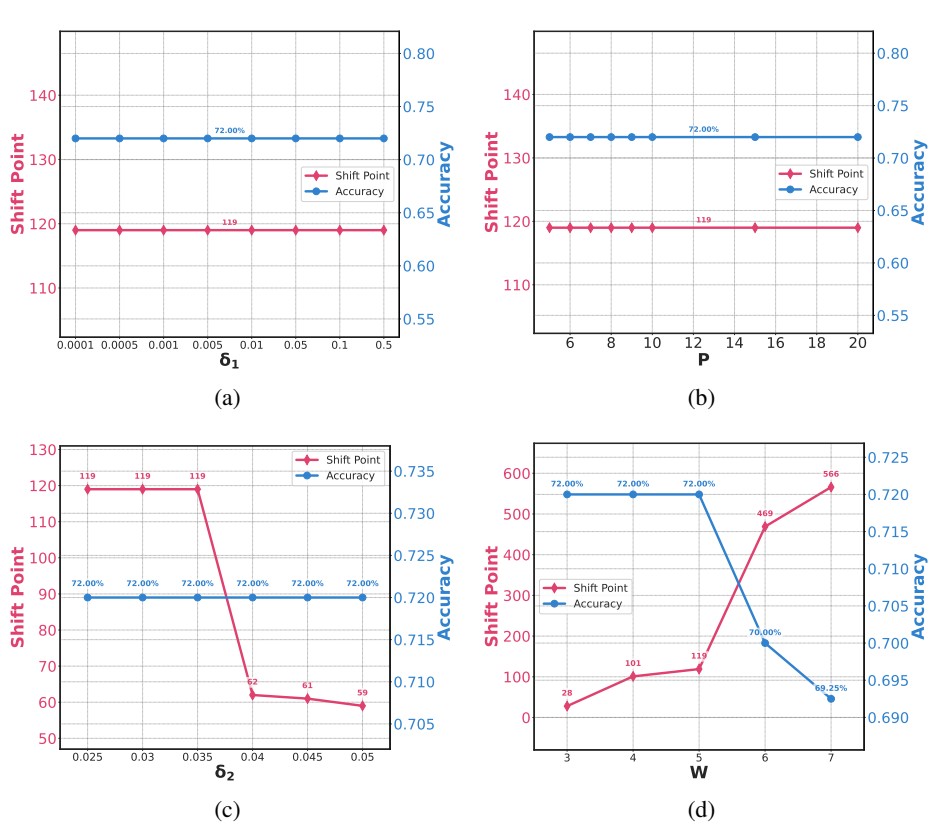

Figure 20: Hyperparameter Sensitivity in SCP.

### E.2 EXPERIMENTS ON PRACTICAL MODELS AND DATASETS

#### E.2.1 EXPERIMENTAL SETUP

This section details the experimental setup for evaluating three training paradigms on practical models and datasets: *Single-Attn*, *Looped-Attn*, and our proposed *SHIFT* framework. Our experimental design follows the methodology for length generalization in looped transformers established by Fan et al. (2024).

**Architectures and Training Paradigms.** To ensure a fair comparison, all experiments are conducted under the equal parameter count principle. We employ a decoder-only GPT-2 architecture as the foundational building block for all models.

- *Single-Attn*: This model is a standard, non-recursive Transformer trained via Full-Output Prediction to generate the entire output sequence in a single forward pass.
- *Looped-Attn*: This model uses the same Transformer block as *Single-Attn* but applies it iteratively. We adopt a recursive variant "FOP-Loop-Adaptive" from Fan et al. (2024). Unlike our toy model with a fixed number of loops (Section E.1), this more advanced setup allows the model to adapt its computational depth. During training, the model is trained to produce the output after exactly $T$ loops for a training sequence of length $T$, with the loss computed only at the $T$-th loop. During inference, it uses an adaptive stopping criterion to select the number of loops for test sequences of different lengths.
- SHIFT: This is our proposed two-stage training strategy that transitions from *Single-Attn* to *Looped-Attn* at a shift point guided by SCP (Section 5).

**Datasets and Tasks.** The datasets and tasks are adapted from Fan et al. (2024). We mainly evaluate models on five algorithmic reasoning tasks: Parity, Addition, Copy, Binary Sum, and Unique Set. These tasks require multi-step reasoning, sequential computation and serve as benchmarks for assessing a model's ability to learn underlying patterns and generalize to sequence lengths not seen during training (length generalization).

**Hyperparameters and Implementation Details.** Across all experiments, the model block is configured with an embedding dimension of 256. The number of attention heads and block depth are task-specific, following the settings in Fan et al. (2024). We use the AdamW optimizer with a learning rate of 1e-4. All models are trained for a total of 50,001 steps. Each experiment is conducted on a single 24GB NVIDIA GeForce RTX 3090.

#### E.2.2 EXPERIMENTAL RESULTS

In the following, we present the experimental results on the above five datasets in Figure 22∼26. For each dataset, we compare the training, validation, and length generalization performances of the three models. Figure 21 summarizes the computational efficiency of the SHIFT framework compared to the *Looped-Attn* baseline.

**Performances of *Single-Attn* and *Looped-Attn*.** We observe two interesting different behaviors on training accuracy curves compared to the experiments on our synthetic Markov language datasets (Figures 22∼26). However, our central findings remain consistent: *Looped-Attn* creates a River-V-Valley landscape and thus demonstrates superior performance compared to the River-U-Valley landscape in *Single-Attn*.

**(a)** On practical models and tasks, the training accuracy for all models achieves near $100\%$ early, which contrasts with the distinct two-phase accuracy curve observed on the toy dataset (Figure 3(b)). This difference stems from the intrinsic structures of the tasks. Specifically,

- An algorithmic task like Parity is governed by a single, recursive underlying rule (*e.g.*, a sequential XOR operation) for all training samples, regardless of length. The initial descent in the valley corresponds to the model learning this core operation, which is sufficient to solve nearly all in-distribution short sequences and causes the training accuracy to quickly plateau. However, this plateau masks a critical divergence in the optimization dynamics. Even after the accuracy metric no longer improves, *Looped-Attn* continues its optimization by exploring river downstream, which is essential for refining the learned core operation into a truly generalizable algorithm. In contrast, *Single-Attn* gets trapped in the flat valley floor which explains its failure in length generalization.

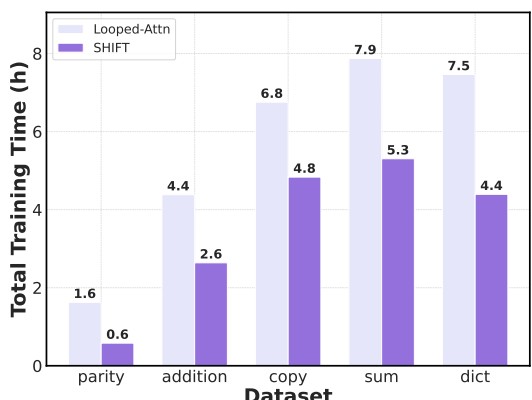

Figure 21: SHIFT Computational Efficiency on Algorithmic Datasets.

- Our synthetic Markov dataset is designed to contain a diverse set of distinct generative rules with varying complexities. This naturally separates the training process: during the valley descent, the model masters the simple rules, while the subsequent downstream exploration is required to learn the more complex rules, resulting in a clear two-phase accuracy progression (if the model learns the complex ones).

**(b)** On practical models and tasks, the accuracy drop upon shifting is significant, but minimal in our toy model experiments (Figure 4(b)). This phenomenon does not contradict the validity of the Stage I initialization in SHIFT, as the accuracy recovers rapidly. It reveals a crucial interaction between the complexity of base architecture and the change of loss landscape.

In both experimental setups, the SHIFT transition reshapes the landscape from a U-shaped valley to a V-shaped valley. However, the magnitude of this geometric shift appears to depend on the complexity of the base architecture.

- On practical tasks, *Looped-Attn* and *Single-Attn* are built upon GPT-2. Applying the recursive principle to this complex base architecture creates a V-shaped valley that is greatly different from the U-shaped valley of its non-recursive ones. This causes the optimizer to significantly push the parameters far from the stable region, leading to the observed temporary collapse in accuracy.

- On our synthetic dataset, *Looped-Attn* and *Single-Attn* are built from a single attention layer. For these simplified models, the geometric distinction between the U-shaped valley and V-shaped valley leads to a ***relatively*** smooth architectural transition and a stable accuracy trajectory.

This initial instability is the short-term cost of transitioning to a more powerful optimization path.

**Effectiveness of SHIFT.** Figures 22∼26 consistently validate the performance effectiveness of our proposed SHIFT framework across all evaluated tasks. As shown in the **(c)** subfigures, *Single-Attn* fails to generalize to longer sequences, while capable of achieving high accuracy on in-distribution training data. In contrast, *Looped-Attn* demonstrates great length generalization capabilities by maintaining high accuracy on longer test sequences. Our SHIFT framework successfully combines the rapid initial convergence of *Single-Attn* with a final performance comparable to the *Looped-Attn* baseline. Furthermore, as shown in Figure 21, SHIFT achieves this strong performance with significantly greater computational efficiency, reducing training time across evaluated algorithmic tasks.

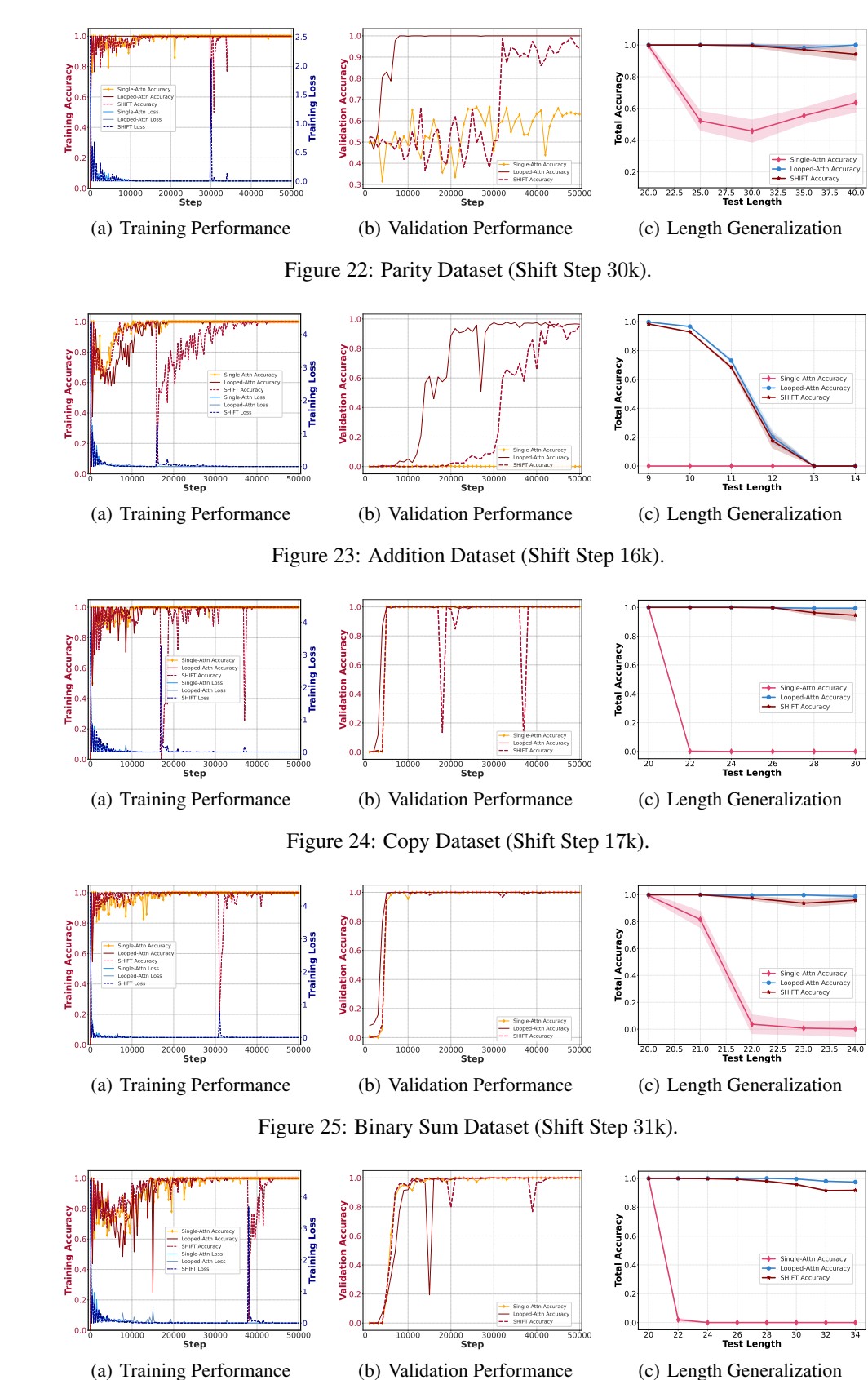

(a) Training Performance  (b) Validation Performance  (c) Length Generalization

Figure 22: Parity Dataset (Shift Step 30k).

(a) Training Performance  (b) Validation Performance  (c) Length Generalization

Figure 23: Addition Dataset (Shift Step 16k).

(a) Training Performance  (b) Validation Performance  (c) Length Generalization

Figure 24: Copy Dataset (Shift Step 17k).

(a) Training Performance  (b) Validation Performance  (c) Length Generalization

Figure 25: Binary Sum Dataset (Shift Step 31k).

(a) Training Performance  (b) Validation Performance  (c) Length Generalization

Figure 26: Unique Set Dataset (Shift Step 38k).

# F    ADDITIONAL DISCUSSIONS ON DEFINITION 1

Figure 27: Eigenvalues of Valley Hessian ($\epsilon = 0.02$).

**Hyperparameters.**    The constants $\epsilon$, $\delta$, and $\zeta$ in Definition 1 serve as descriptive symbols to characterize the intrinsic landscape geometry. Specifically, $\epsilon$ partitions the parameter space into the river (the optimum exists in the river downstream) and valley (it generates driving force on river). Its selection needs to respect the intrinsic spectral gap of the model. A reasonable $\epsilon$ is essential for our theoretical results: setting $\epsilon$ too large would misclassify small eigenvalues as river components. This artificially excludes the primary contributors to the valley's energy $\mathcal{E}$, thereby hiding the driving force inherent in the V-shaped valley on river. Conversely, setting $\epsilon$ too close to zero risks including numerical noise into the valley analysis. Furthermore, $\delta$ and $\zeta$ quantify geometric distinctions: $\delta$ distinguishes between the well-conditioned and ill-conditioned geometries, and $\zeta$ serves as a baseline for energy magnitude.

**Representative Examples.**    We assume a small threshold $\epsilon$ (*e.g.*, $\epsilon = 0.02$) separates the River and Valley subspaces. Let $\{\lambda_1, \lambda_2, \lambda_3\}$ denote the eigenvalues and $\{v_1, v_2, v_3\}$ denote the corresponding eigenvectors. We analyze four representative functions to illustrate the standard River-U-Valley, River-V-Valley, and other special landscapes beyond the scope of Definition 1.

**Case A: River-U-Valley ($\kappa$ and $\mathcal{E}$ are small).**

$$f_A = 0.001x_1^2 + x_2^2 + x_3^2.$$

The eigenvalues are $\lambda_1 = 0.002 \leq \epsilon, \lambda_2, \lambda_3 = 2 > \epsilon$. With Definition 1, the subspaces are $S_{\text{River}} = \text{span}\{v_1\}$ and $S_{\text{Valley}} = \text{span}\{v_2, v_3\}$. For the valley subspace, the condition number of valley Hessian is $\kappa = 1$ (well-condition), and the inverse Hessian average energy is $\mathcal{E} = 0.25$ (small energy). This geometry corresponds to a U-shaped Valley: an isotropic bowl with uniformly steep cliffs. In total, the landscape of $f_A$ is River-U-Valley.

**Case B: River-V-Valley ($\kappa$ and $\mathcal{E}$ are large).**

$$f_B = 0.001x_1^2 + 0.02x_2^2 + 2x_3^2.$$

The eigenvalues are $\lambda_1 = 0.002 \leq \epsilon, \lambda_2 = 0.04 > \epsilon, \lambda_3 = 4 > \epsilon$. With Definition 1, the subspaces are $S_{\text{River}} = \text{span}\{v_1\}$ and $S_{\text{Valley}} = \text{span}\{v_2, v_3\}$. For the valley subspace, the condition number of valley Hessian is $\kappa = 100$ (ill-condition), the inverse Hessian average energy is $\mathcal{E} = 312.625$ (large energy). This geometry corresponds to a V-shaped Valley, characterized by varied and steep cliffs. In total, the landscape of $f_B$ is River-V-Valley.

**Case C: Anisotropic Valley with Low Energy (Large $\kappa$, Small $\mathcal{E}$).**

$$f_C = 0.001x_1^2 + x_2^2 + 100x_3^2.$$

The eigenvalues are $\lambda_1 = 0.002 \leq \epsilon, \lambda_2 = 2 > \epsilon, \lambda_3 = 200 > \epsilon$. With Definition 1, the subspaces are $S_{\text{River}} = \text{span}\{v_1\}$ and $S_{\text{Valley}} = \text{span}\{v_2, v_3\}$. For the valley subspace, the condition number

of valley Hessian is $\kappa = 100$ (ill-condition), the inverse Hessian average energy is $\mathcal{E} \approx 0$ (small energy). Consequently, this geometry fits neither the U-shaped nor the V-shaped definition. For the optimization dynamic, the large condition number induces hopping within the valley. However, unlike the V-shaped valley, this hopping does not convert into effective river exploration because the valley lacks a sufficiently small eigenvalue to drive the update. This case represents a suboptimal anisotropic optimization landscape where the model endures instability without facilitating river exploration. In total, this case is beyond the scope of this paper (Definition 1). In other words, *Single-Attn* and *Looped-Attn* does not possess such landscapes.

**Case D: Isotropic Valley with High Energy (Small $\kappa$, Large $\mathcal{E}$).**

$$f_D = 0.001x_1^2 + 0.02x_2^2 + 0.02x_3^2.$$

The eigenvalues are $\lambda_1 = 0.002 \leq \epsilon, \lambda_2, \lambda_3 = 0.04 > \epsilon$. With Definition 1, the subspaces are $S_{\text{River}} = \text{span}\{v_1\}$ and $S_{\text{Valley}} = \text{span}\{v_2, v_3\}$. For the valley subspace, the condition number of valley Hessian is $\kappa = 1$ (well-condition), the inverse Hessian average energy is $\mathcal{E} = 625$ (large energy). Consequently, this geometry fits neither the U-shaped nor the V-shaped definition. For the optimization dynamic, although the high energy implies a large potential driving force, the small condition number induces a rapid smooth descent to the valley floor ($\theta_V \to 0$). Unlike the V-shaped valley where oscillation keeps the valley parameters active, the rapid decay of $\theta_V$ causes the coupling force on the river ($H_{RV}\theta_V$) to vanish quickly. Therefore, despite the high energy, the model quickly becomes trapped at the valley floor, failing to explore river downstream. In total, this case is beyond the scope of this paper (Definition 1). In other words, *Single-Attn* and *Looped-Attn* does not possess such landscapes.

# G RIVER-V-VALLEY BRINGS SUPERIOR OPTIMIZATION PERFORMANCE

## G.1 DEFINITIONS AND ASSUMPTIONS

**Definition 2** (**Block-Structured Hessian**). *Let the Hessian matrix $H$ be represented in the orthonormal basis of the Valley $\{v_i\}$ and River $\{r_j\}$ subspaces. Its block components are defined by the second directional derivatives of the loss $\widehat{L}$ as follows:*

$$[H_{Valley}]_{ij} = \frac{\partial^2 \widehat{L}}{\partial v_i \partial v_j}, \quad [H_{VR}]_{ij} = \frac{\partial^2 \widehat{L}}{\partial v_i \partial r_j}, \quad [H_{RV}]_{ij} = \frac{\partial^2 \widehat{L}}{\partial r_i \partial v_j}, \quad [H_{River}]_{ij} = \frac{\partial^2 \widehat{L}}{\partial r_i \partial r_j}.$$

*Proof.* This block structure is formally derived through a change of basis, transforming the standard Hessian into the coordinate system defined by the River-Valley subspaces.

**From standard basis to the River-Valley subspaces.** Let $H_{\text{old}}$ be the Hessian of the loss function $\widehat{L}(\theta)$ with respect to the standard basis of $\mathbb{R}^d$, where

$$[H_{\text{old}}]_{ij} = \frac{\partial^2 \widehat{L}}{\partial \theta_i \partial \theta_j}.$$

We introduce a new orthonormal basis aligned with the geometry of the landscape, formed by the basis vectors of the valley subspace, $S_{\text{Valley}} = \text{span}\{v_1, \ldots, v_{d_V}\}$, and the river subspace, $S_{\text{River}} = \text{span}\{r_1, \ldots, r_{d_R}\}$.

The change of basis from the River-Valley coordinates to the standard coordinates is given by the orthonormal matrix $U$:

$$U = (V, R) = (v_1, \cdots, v_{d_V}, r_1, \cdots, r_{d_R}) \in \mathbb{R}^{d \times (d_V + d_R)},$$

where $V \in \mathbb{R}^{d \times d_V}$ and $R \in \mathbb{R}^{d \times d_R}$ are matrices whose columns are the basis vectors of the respective subspaces.

**The Hessian in the new basis.** The representation of the Hessian $H$ in this new basis is

$$H = U^\top H_{\text{old}} U.$$

Substituting the block form of $U$ yields the block structure of $H$:

$$H = \begin{pmatrix} V^\top \\ R^\top \end{pmatrix} H_{\text{old}} \begin{pmatrix} V & R \end{pmatrix} = \begin{pmatrix} V^\top H_{\text{old}} V & V^\top H_{\text{old}} R \\ R^\top H_{\text{old}} V & R^\top H_{\text{old}} R \end{pmatrix}.$$

From this, we can identify each block:

- $H_{\text{Valley}} = V^\top H_{\text{old}} V$: The projection of the Hessian onto the Valley subspace.

- $H_{VR} = V^\top H_{\text{old}} R$: The coupling term from the River to the Valley subspace.

- $H_{RV} = R^\top H_{\text{old}} V$: The coupling term from the Valley to the River subspace.

- $H_{\text{River}} = R^\top H_{\text{old}} R$: The projection of the Hessian onto the River subspace.

Thus, we have

$$[H_{\text{Valley}}]_{ij} = v_i^\top H_{\text{old}} v_j = \frac{\partial^2 \widehat{L}}{\partial v_i \partial v_j},$$

$$[H_{VR}]_{ij} = v_i^\top H_{\text{old}} r_j = \frac{\partial^2 \widehat{L}}{\partial v_i \partial r_j},$$

$$[H_{RV}]_{ij} = r_i^\top H_{\text{old}} v_j = \frac{\partial^2 \widehat{L}}{\partial r_i \partial v_j},$$

$$[H_{\text{River}}]_{ij} = r_i^\top H_{\text{old}} r_j = \frac{\partial^2 \widehat{L}}{\partial r_i \partial r_j}.$$

$\square$

**Setting 1 (Quadratic Loss).** *One simple example of a River-Valley landscape (Definition 1) is the quadratic loss:*

$$\widehat{L}(\theta_V, \theta_R) = \frac{1}{2} \begin{pmatrix} \theta_V \\ \theta_R \end{pmatrix}^\top \begin{pmatrix} H_{Valley} & H_{VR} \\ H_{RV} & \mathbf{0} \end{pmatrix} \begin{pmatrix} \theta_V \\ \theta_R \end{pmatrix} - h_R^\top \theta_R,$$

*where $[H_{Valley}]_{ij} = \frac{\partial^2 \widehat{L}}{\partial v_i \partial v_j}, [H_{VR}]_{ij} = \frac{\partial^2 \widehat{L}}{\partial v_i \partial r_j}, [H_{RV}]_{ij} = \frac{\partial^2 \widehat{L}}{\partial r_i \partial v_j}$ (Definition 2 in Appendix G.1). We assume the coupling strength along the valley eigenvectors $v_i$ satisfies $\underline{h} \le \|H_{RV} v_i\| \le \bar{h}$ for constants $\underline{h}, \bar{h} > 0$, and the valley parameters are initialized as $\theta_{V,0} \sim \mathcal{N}(0, \bar{\alpha}^2 I/d_V)$ with $\|\theta_{V,0}\| \le \bar{\alpha}$ for a constant $\bar{\alpha} > 0$.*

**Remark 5.** The structure of this loss model is a principled abstraction of our theoretical model and empirical observations. Each component of the function corresponds to a specific geometric hypothesis.

**The valley component $\widehat{L}_{\mathbf{Valley}}(\theta_V)$.** The valley is a subspace with high curvature. Any movement away from the valley floor should result in a significant increase in the loss value. We adopt a simplest quadratic function to capture this behavior and landscape:

$$\widehat{L}_{Valley}(\theta_V) = \frac{1}{2} \theta_V^\top H_{Valley} \theta_V.$$

The matrix $H_{Valley}$ is the valley Hessian. Its spectral properties (condition number) directly model the shape of the valley: U-shape and V-shape defined in Definition 1.

**The river component $\widehat{L}_{\mathbf{River}}(\theta_R)$.** The river corresponds to the subspace with near-zero eigenvalues, forming a flat manifold. While the true landscape may possess non-zero curvature in these directions, empirical observations in Figures 7~8 reveal a massive spectral gap between valley and river directions (*i.e.*, $\lambda_{Valley} \gg \lambda_{River}$). This suggests that along the full optimization trajectory (including regions outside the idealized flat manifold), the quadratic confinement provided by the curvature in the river direction is negligible compared to the driving force of the gradient. Consequently, optimization dynamics within the river are dominated by the first-order gradient term. We thus adopt the approximation $H_{River} \approx \mathbf{0}$ and model the river using a linear term:

$$\widehat{L}_{River}(\theta_R) = -h_R^\top \theta_R.$$

Here, the vector $h_R$ represents the intrinsic gradient flow along the River. The negative sign indicates that moving in the direction of $h_R$ decreases the loss. It effectively captures the slow dynamics along the river relative to the fast dynamics in the valley.

**The coupling component $\widehat{L}_{\mathbf{Coupling}}(\theta_V, \theta_R)$.** The optimization in valley and river subspaces are not perfectly independent. To model their interaction, we adopt $H_{RV}$ to construct a simple quadratic form:

$$\widehat{L}_{Coupling}(\theta_V, \theta_R) = \theta_R^\top H_{RV} \theta_V = \theta_V^\top H_{VR} \theta_R,$$

since Hessian is symmetric, *i.e.*, $H_{RV} = H_{VR}^\top$. The matrix $H_{RV}$ is the Coupling Matrix that quantifies the strength of the interaction between the subspaces. Specifically, $H_{RV}$ describes how a movement in the valley induces a gradient in the river.

**Assembling the final model.** By combining these three principled components, we arrive at our final quadratic loss function:

$$\widehat{L}(\theta_V, \theta_R) = \frac{1}{2} \theta_V^\top H_{Valley} \theta_V - h_R^\top \theta_R + \theta_R^\top H_{RV} \theta_V.$$

This can be expressed compactly in the block-matrix form:

$$\widehat{L}(\theta_V, \theta_R) = \frac{1}{2} \begin{pmatrix} \theta_V \\ \theta_R \end{pmatrix}^\top \begin{pmatrix} H_{Valley} & H_{VR} \\ H_{RV} & \mathbf{0} \end{pmatrix} \begin{pmatrix} \theta_V \\ \theta_R \end{pmatrix} - h_R^\top \theta_R.$$

In addition, for the initialization $\theta_{V,0}$, we derive that

$$\mathbb{E}\left[\|\theta_{V,0}\|^2\right] = \mathbb{E}\left[\sum_{i=1}^{d_V} \theta_i^2\right] = d_V \frac{\bar{\alpha}^2}{d_V} = \bar{\alpha}^2.$$

According to the Law of Large Numbers, as $d_V$ is large, the norm of $\theta_{V,0}$ is concentrated around its expected value $\bar{\alpha}$. This initialization guarantees that $\theta_{V,0}$ possesses non-zero projections onto the eigenvectors associated with small valley eigenvalues. These components are essential for activating the significant cumulative driving force of *Looped-Attn*.

**Assumption 1** (**Dominant Effect in Average Energy**). *Let $\mathcal{E}^{(1)}$ and $\mathcal{E}^{(2)}$ denote the Inverse Hessian Average Energy (Definition 1) for Single-Attn and Looped-Attn, respectively. With $\underline{h} \leq \|H_{RV}v_i\| \leq \bar{h}$ (Setting 1), assume that $\mathcal{E}^{(2)}/\mathcal{E}^{(1)} \gg \bar{h}^2/\underline{h}^2$.*

**Remark 6.** Assumption 1 ensures that the landscape advantage of *Looped-Attn* compared to *Single-Attn*, characterized by the significant magnitude of inverse eigenvalues ($\mathcal{E}^{(2)} \gg \mathcal{E}^{(1)}$), dominates the scaling effects in the coupling strength. Specifically, the ratio $\bar{h}^2/\underline{h}^2$ is of a constant order, since $\bar{h}$ and $\underline{h}$ correspond to the projection strengths of $H_{RV}$ onto different valley eigenvectors, which typically share the same magnitude. In contrast, the energy ratio $\mathcal{E}^{(2)}/\mathcal{E}^{(1)}$ exceeds this constant order due to the significant structural differences between *Single-Attn* and *Looped-Attn* models.

**Assumption 2** (**Bounded Time-Varying Valley Hessian**). *Let $\{H_{Valley}(\theta_k)\}_{k \geq 0}$ be the sequence of Valley Hessians during the optimization trajectory. There exist constant, positive semi-definite matrices $H^B$ and $H^T$ sharing a common stable basis with $\{H_{Valley}(\theta_k)\}_{k \geq 0}$, such that for all steps $k$: $H^B \preceq H_{Valley}(\theta_k) \preceq H^T$, where $\preceq$ denotes the Loewner order.*

**Remark 7.** Assumption 2 posits a structurally stable valley subspace where the eigenvectors of $H_{\text{Valley}}(\theta_k)$ do not rotate significantly, while the eigenvalues vary during the optimization phase. We use matrices $H^B$ and $H^T$ to bound this evolving eigenspectrum. Specifically, with $H^B \preceq H_{\text{Valley}}(\theta_k) \preceq H^T$, we have that the sorted eigenvalues satisfy $\lambda_i(H^B) \leq \lambda_i(H_{\text{Valley}}(\theta_k)) \leq \lambda_i(H^T), \forall i = 1, \cdots, d$. Intuitively, the lower bound $\lambda_i(H^B)$ guarantees that the valley directions do not become infinitely flat, ensuring the landscape possesses sufficient curvature to drive optimization. The upper bound $\lambda_i(H^T)$ ensures that the steepest directions do not become infinitely steep, *i.e.*, the Hessian satisfies Lipschitz smoothness.

**Assumption 3** (**Bounded Time-Varying Coupling Hessian**). *Let $H_{RV}(\theta_k)$ be the time-varying coupling matrix at step $k$. There exist constant matrices $\underline{H}$ and $\overline{H}$, such that $\underline{H}^\top \underline{H} \preceq H_{RV}^\top H_{RV} \preceq \overline{H}^\top \overline{H}$, The coupling strength along the stable valley eigenvectors (Assumption 2) satisfy $\underline{h}_{gen} \leq \|\underline{H}v_i^T\|, \|\overline{H}v_i^B\| \leq \overline{h}_{gen}$ for constants $\underline{h}_{gen}, \overline{h}_{gen} > 0$.*

**Remark 8.** With Assumption 2, the eigenvectors $\{v_i^T\}$ or $\{v_i^B\}$ are stable, where $\{v_i^T\}$ denote the eigenvalues of Hessian upper bound $H^T$, and $\{v_i^B\}$ denote the eigenvalues of Hessian lower bound $H^B$. Assumption 3 bounds the coupling energy $H_{RV}^\top H_{RV}$ and coupling strength along these stable eigenvectors. Specifically, it guarantees that the interaction between the valley and river subspaces is well-behaved. The upper bounds ensure Lipschitz smoothness and the lower bounds ensure that the gradient conversion from valley to river does not vanish.

**Assumption 4** (**Dominant Effect in Average Energy**). *Let $\mathcal{E}^{(1)}$ and $\mathcal{E}^{(2)}$ denote the Inverse Hessian Average Energy (Definition 1) for Single-Attn and Looped-Attn, respectively. With $\|\underline{H}v_i^T\| \geq \underline{h}_{gen}, \|\overline{H}v_i^B\| \leq \overline{h}_{gen}$ (Assumption 3), assume that $\mathcal{E}^{(2)}/\mathcal{E}^{(1)} \gg \bar{h}_{gen}^2/\underline{h}_{gen}^2$.*

**Remark 9.** Assumption 4 ensures that the landscape advantage of *Looped-Attn* compared to *Single-Attn*, characterized by the significant magnitude of inverse eigenvalues ($\mathcal{E}^{(2)} \gg \mathcal{E}^{(1)}$), dominates the scaling effects in the coupling strength.

## G.2 Proof for Theorem 1 and Corollary 1

We aim to prove that over $K$ iterations (the stage where the valley's dynamics largely drive progress in the river, i.e., before reaching the river), the total progress made in the river subspace is significantly greater for *Looped-Attn* than for *Single-Attn*. In our theoretical model, superior convergence performance is defined as the ability to explore further along the river, thus reaching a better optimization performance.

With the quadratic loss model from Setting 1, $\widehat{L}(\theta_V, \theta_R) = \frac{1}{2}\theta_V^\top H_{\text{Valley}}\theta_V - h_R^\top\theta_R + \theta_R^\top H_{RV}\theta_V$, we derive the gradients:

$$\frac{\partial\widehat{L}(\theta_V, \theta_R)}{\partial\theta_V} = H_{\text{Valley}}\theta_{V,k} + H_{VR}\theta_{R,k},$$

$$\frac{\partial\widehat{L}(\theta_V, \theta_R)}{\partial\theta_R} = H_{RV}\theta_{V,k} - h_R.$$

Therefore the GD update rules for the two subspaces are:

$$\theta_{V,k+1} = \theta_{V,k} - \eta\left(H_{\text{Valley}}\theta_{V,k} + H_{VR}\theta_{R,k}\right) = (I - \eta H_{\text{Valley}})\theta_{V,k} - \eta H_{VR}\theta_{R,k}, \tag{1}$$

$$\theta_{R,k+1} = \theta_{R,k} - \eta\left(H_{RV}\theta_{V,k} - h_R\right). \tag{2}$$

**Derivation of the cumulative change in the river subspace.** Our goal is to quantify the total progress made within the river subspace during $K$ iterations. From Equation 2, the total change in $\theta_R$ after $K$ steps is:

$$\Delta\theta_{R,K} \triangleq \theta_{R,K} - \theta_{R,0} = \sum_{k=0}^{K-1}(\theta_{R,k+1} - \theta_{R,k})$$

$$= \sum_{k=0}^{K-1}(\eta h_R - \eta H_{RV}\theta_{V,k})$$

$$= K\eta h_R - \eta\sum_{k=0}^{K-1}H_{RV}\theta_{V,k}. \tag{3}$$

The first term represents progress driven by the river's intrinsic constant gradient. The second term represents the influence from the valley. We define $C_K$ to be the cumulative effect induced by the valley dynamics on the river, *i.e.*, movement in the valley $\theta_{V,k}$ induces a gradient in the river:

$$C_K \triangleq \eta\sum_{k=0}^{K-1}H_{RV}\theta_{V,k}.$$

**Spectral analysis of the dominant dynamics.** The cumulative effect $C_K$ depends on the trajectory of $\theta_{V,k}$. The recurrence for $\theta_{V,k}$ in Equation 1 can be solved as

$$\theta_{V,k} = \Phi^k\theta_{V,0} - \eta\sum_{j=0}^{k-1}\Phi^{k-1-j}H_{VR}\theta_{R,j}, \tag{4}$$

where $\Phi \triangleq I - \eta H_{\text{Valley}}$. The trajectory of $\theta_V$ is composed of two parts: **(a)** the unforced update, $\Phi^k\theta_{V,0}$, represents the intrinsic decay of the valley component; and **(b)** the summation term represents the cumulative influence on the valley from the river. During the early and intermediate stages of optimization (bouncing between the valleys), the magnitude of $\theta_V$ grows rapidly and remains significantly larger than that of $\theta_R$. Consequently, the term $-\eta H_{RV}\theta_V$ generates a significant driving force on the river, while the term $-\eta H_{VR}\theta_R$ acts only as a minor perturbation on the valley. Thus, the dynamics of the valley dominate and drive the exploration of the river, while the dynamics of the river can be regarded as a secondary perturbation to the valley. We mainly consider the dominant part (a) in the following.

Let $H_{\text{Valley}} = Q \Lambda Q^\top$ be the spectral decomposition, where $Q = [v_1, \ldots, v_{d_V}]$ is the orthonormal matrix of eigenvectors and $\Lambda = \text{diag}(\lambda_1, \ldots, \lambda_{d_V})$ is the diagonal matrix of corresponding eigenvalues. The dominant dynamics of $\theta_V$ are governed by the unforced update $\Phi^k \theta_{V,0}$, which can be expressed in the eigen-space as:

$$
\begin{aligned}
\Phi^k \theta_{V,0} &= (I - \eta Q \Lambda Q^\top)^k \theta_{V,0} \\
&= (Q I Q^\top - \eta Q \Lambda Q^\top)^k \theta_{V,0} \\
&= Q(I - \eta \Lambda)^k Q^\top \theta_{V,0} \\
&= \sum_{i=1}^{d_V} (1 - \eta \lambda_i)^k v_i^\top \theta_{V,0} v_i
\end{aligned}
\tag{5}
$$

**Dominant cumulative effect for *Single-Attn* and *Looped-Attn*.** We denote the dominant part of the cumulative effect, arising from the unforced update, as $\widehat{C}_K$:

$$
\widehat{C}_K \triangleq \eta \sum_{k=0}^{K-1} H_{RV} \Phi^k \theta_{V,0}.
\tag{6}
$$

Let $\rho_i \triangleq 1 - \eta \lambda_i$ be the decay rate of the $i$-th component. Substituting Equation 5 into Equation 6 yields:

$$
\begin{aligned}
\widehat{C}_K = \eta \sum_{k=0}^{K-1} H_{RV} \Phi^k \theta_{V,0} &= \eta \sum_{k=0}^{K-1} H_{RV} \left( \sum_{i=1}^{d_V} \rho_i^k v_i^\top \theta_{V,0} v_i \right) \\
&= \eta \sum_{i=1}^{d_V} H_{RV} v_i^\top \theta_{V,0} v_i \left( \sum_{k=0}^{K-1} \rho_i^k \right) \\
&= \eta \sum_{i=1}^{d_V} H_{RV} v_i^\top \theta_{V,0} v_i \left( \frac{1 - \rho_i^K}{1 - \rho_i} \right) \\
&= \sum_{i=1}^{d_V} \frac{v_i^\top \theta_{V,0}}{\lambda_i} H_{RV} v_i \left( 1 - \rho_i^K \right).
\end{aligned}
$$

As $K \to \infty$, $\widehat{C}_K$ is asymptotic to $C_\infty$:

$$
\begin{aligned}
C_\infty = \lim_{K \to \infty} \widehat{C}_K &= \lim_{K \to \infty} \sum_{i=1}^{d_V} \frac{1}{\lambda_i} H_{RV} v_i^\top \theta_{V,0} v_i \left( 1 - \rho_i^K \right) \\
&= \sum_{i=1}^{d_V} \frac{1}{\lambda_i} H_{RV} v_i^\top \theta_{V,0} v_i.
\end{aligned}
$$

With $\|H_{RV}\| \leq \bar{h}$ and $\|\theta_{V,0}\| \leq \bar{\alpha}$ in Setting 1, we consider the norm of asymptotic value $C_\infty$:

$$
\begin{aligned}
\|C_\infty\| &= \left\| \sum_{i=1}^{d_V} \frac{1}{\lambda_i} H_{RV} v_i^\top \theta_{V,0} v_i \right\| \\
&\leq \sum_{i=1}^{d_V} \frac{1}{|\lambda_i|} |v_i^\top \theta_{V,0}| \cdot \|H_{RV} v_i\| \\
&\leq \|H_{RV}\| \sum_{i=1}^{d_V} \frac{1}{|\lambda_i|} |v_i^\top \theta_{V,0}| \\
&\leq \sqrt{d_V} \|H_{RV}\| \|\theta_{V,0}\| \sum_{i=1}^{d_V} \frac{1}{|\lambda_i|} \\
&\leq \sqrt{d_V} \, \bar{h} \, \bar{\alpha} \sum_{i=1}^{d_V} \frac{1}{|\lambda_i|} \triangleq \mathcal{C}
\end{aligned}
\tag{7}
$$

It means that after $K$ iterations, the driving force from valley is limited to $\mathcal{C}$. In other words, $\mathcal{C}$ quantifies the total potential driving force the valley can generate, which is primarily related to the inverse of the eigenvalues of the valley subspace.

With the expression of cumulative force, $\mathcal{C} = \sqrt{d_V}\,\bar{h}\,\bar{\alpha}\sum_{i=1}^{d_V}\frac{1}{|\lambda_i|}$, we then compare two models.

The spectral experiments presented in Figure 27 (with $\epsilon = 0.02$) reveal that *Looped-Attn* exhibits a larger $\mathcal{E}(H_{\text{Valley}})$ than *Single-Attn*. Thus with Definition 1, we summarize the characteristics of two models in Conjecture 1~2.

For *Single-Attn* with River-U-Valley, we have $\frac{1}{d_V^{(1)}}\sum_{i=1}^{d_V^{(1)}}\frac{1}{(\lambda_i^{(1)})^2} \leq \zeta$. With inequality $\|x\|_1 \leq \sqrt{d}\|x\|_2$ for vector $x \in \mathbb{R}^d$, the maximal cumulative force satisfies

$$\mathcal{C}^{(1)} = \bar{h}\bar{\alpha}\sqrt{d_V^{(1)}}\sum_{i=1}^{d_V^{(1)}}\frac{1}{|\lambda_i^{(1)}|} \leq \bar{h}\bar{\alpha}\sqrt{d_V^{(1)}}\sqrt{d_V^{(1)}}\sqrt{\sum_{i=1}^{d_V^{(1)}}\frac{1}{(\lambda_i^{(1)})^2}} \leq \bar{h}\bar{\alpha}(d_V^{(1)})^{3/2}\sqrt{\zeta}.$$

For *Looped-Attn* with River-V-Valley, we have $\frac{1}{d_V^{(2)}}\sum_{i=1}^{d_V^{(2)}}\frac{1}{(\lambda_i^{(2)})^2} \gg \zeta$. With inequality $\|x\|_1 \geq \|x\|_2$ for vector $x \in \mathbb{R}^d$, the maximal cumulative force satisfies

$$\mathcal{C}^{(2)} = \bar{h}\bar{\alpha}\sqrt{d_V^{(2)}}\sum_{i=1}^{d_V^{(2)}}\frac{1}{|\lambda_i^{(2)}|} \geq \bar{h}\bar{\alpha}\sqrt{d_V^{(2)}}\sqrt{\sum_{i=1}^{d_V^{(2)}}\frac{1}{(\lambda_i^{(2)})^2}} \gg \bar{h}\bar{\alpha}d_V^{(2)}\sqrt{\zeta}.$$

The valley dimensions of two model are typically of the same order, thus we conclude that

$$\mathcal{C}^{(2)} \gg \mathcal{C}^{(1)}.$$

This proves that the ill-conditioned nature of the V-shaped valley provides a larger potential for driving exploration in the river subspace. This continued and powerful exploration allows *Looped-Attn* to navigate further down the river, overcoming performance plateaus and achieving a superior optimization performance compared to the rapidly trapped *Single-Attn* model.

### G.3 PROOF FOR COROLLARY 2

The quadratic loss in Setting 1 is:

$$\widehat{L}(\theta_V, \theta_R) = \frac{1}{2}\theta_V^\top H_{\text{Valley}}\theta_V - h_R^\top \theta_R + \theta_R^\top H_{RV}\theta_V.$$

From Theorem 1, as $K \to \infty$, the cumulative force converges to:

$$C_\infty = \sum_{i=1}^{d_V} \frac{1}{\lambda_i} H_{RV} v_i^\top \theta_{V,0} v_i = \sum_{i=1}^{d_V} \frac{c_i}{\lambda_i} u_i,$$

where $c_i \triangleq v_i^\top \theta_{V,0}$ and $u_i \triangleq H_{RV} v_i \in \mathbb{R}^{d_R}$.

With $\theta_{V,0} \sim \mathcal{N}(0, \bar{\alpha}^2 I/d_V)$ in Setting 1, we have

$$\mathbb{E}[c_i c_j] = \mathbb{E}[v_i^\top \theta_{V,0}\theta_{V,0}^\top v_j] = v_i^\top \mathbb{E}[\theta_{V,0}\theta_{V,0}^\top]v_j.$$

If $i = j$, $\mathbb{E}[c_i^2] = \bar{\alpha}^2/d_V$. If $i \neq j$, $\mathbb{E}[c_i^2] = 0$. Then the expected norm is

$$\mathbb{E}\left[\|C_\infty\|^2\right] = \mathbb{E}\left[\left\langle \sum_{i=1}^{d_V}\frac{c_i}{\lambda_i}u_i, \sum_{j=1}^{d_V}\frac{c_j}{\lambda_j}u_j \right\rangle\right]$$

$$= \sum_{i=1}^{d_V}\sum_{j=1}^{d_V}\frac{\mathbb{E}[c_i c_j]}{\lambda_i \lambda_j}\langle u_i, u_j\rangle$$

$$= \sum_{i=1}^{d_V}\frac{\mathbb{E}[c_i^2]}{\lambda_i^2}\|u_i\|^2 = \frac{\bar{\alpha}^2}{d_V}\sum_{i=1}^{d_V}\frac{\|u_i\|^2}{\lambda_i^2}.$$

With $\underline{h} \le \|H_{RV}v_i\| \le \bar{h}$ in Setting 1, we have

$$\frac{\bar{\alpha}^2}{d_V}\underline{h}^2\sum_{i=1}^{d_V}\frac{1}{\lambda_i^2} \le \mathbb{E}\left[\|C_\infty\|^2\right] \le \frac{\bar{\alpha}^2}{d_V}\bar{h}^2\sum_{i=1}^{d_V}\frac{1}{\lambda_i^2}.$$

Thus,

$$\mathbb{E}\left[\|C_\infty^{(2)}\|^2\right] \ge \frac{\bar{\alpha}^2}{d_V^{(2)}}\underline{h}^2\sum_{i=1}^{d_V^{(2)}}\frac{1}{(\lambda_i^{(2)})^2}, \quad \frac{\bar{\alpha}^2}{d_V^{(1)}}\bar{h}^2\sum_{i=1}^{d_V^{(1)}}\frac{1}{(\lambda_i^{(1)})^2} \ge \mathbb{E}\left[\|C_\infty^{(1)}\|^2\right].$$

With Definition 1 and Assumption 1, it leads to

$$\mathbb{E}\left[\|C_\infty^{(2)}\|^2\right] \gg \mathbb{E}\left[\|C_\infty^{(1)}\|^2\right].$$

*i.e.*, as $K \to \infty$, $\mathbb{E}\left[\|C_K^{(2)}\|^2\right] \gg \mathbb{E}\left[\|C_K^{(1)}\|^2\right].$

Let $K$ be a number of iterations large enough such that the valley parameters for both models have converged to the bottom of their respective valleys.

The well-conditioned U-shaped valley of *Single-Attn* leads to converge rapidly in the valley subspace (within $K_1$ steps). The ill-conditioned V-shaped valley of *Looped-Attn* leads to slower convergence in the valley (within $K_2$ steps, where $K_2 > K_1$). We consider $K = K_2$.

At iteration $K$, for both models, the valley parameters are effectively zero:

$$\theta_{V,K}^{(1)} \approx \mathbf{0} \quad \text{and} \quad \theta_{V,K}^{(2)} \approx \mathbf{0}.$$

Given $\theta_{V,K} \approx \mathbf{0}$, the valley and coupling terms become negligible for both models:

$$\widehat{L}_{\text{Valley},K} = \frac{1}{2}\theta_{V,K}^\top H_{\text{Valley}}\theta_{V,K} \approx 0.$$

$$\widehat{L}_{\text{Coupling},K} = \theta_{R,K}^\top H_{RV} \theta_{V,K} \approx 0.$$

Therefore, the total loss for each model at step $K$ is dominated by its river component:

$$\widehat{L}_K^{(1)} \approx \widehat{L}_{\text{River},K}^{(1)} = -h_R^\top \theta_{R,K}^{(1)}.$$

$$\widehat{L}_K^{(2)} \approx \widehat{L}_{\text{River},K}^{(2)} = -h_R^\top \theta_{R,K}^{(2)}.$$

From Equation 3,

$$\Delta\theta_{R,K} = K\eta h_R - \eta \sum_{k=0}^{K-1} H_{RV}\theta_{V,k} = K\eta h_R - C_K,$$

where $C_K$ represents the cumulative effect from the valley dynamics over $K$ iterations.

**Loss Comparison.** We analyze the change in the river loss, $\Delta\widehat{L}_{\text{River},K} = \widehat{L}_{\text{River},K} - \widehat{L}_{\text{River},0}$.

$$\mathbb{E}\left[ \left|\Delta\widehat{L}_{\text{River},K}^{(1)}\right|^2 - \left|\Delta\widehat{L}_{\text{River},K}^{(2)}\right|^2 \right]$$

$$=\mathbb{E}\left[ \left|-h_R^\top(\theta_{R,K}^{(1)} - \theta_{R,0}^{(1)})\right|^2 - \left|-h_R^\top(\theta_{R,K}^{(2)} - \theta_{R,0}^{(2)})\right|^2 \right]$$

$$=\mathbb{E}\left[ K^2\eta^2\|h_R\|^4 + \|h_R\|^2\|C_K^{(1)}\|^2 - K^2\eta^2\|h_R\|^4 - \|h_R\|^2\|C_K^{(2)}\|^2 \right]$$

$$=\|h_R\|^2\mathbb{E}\left[ \|C_K^{(1)}\|^2 - \|C_K^{(2)}\|^2 \right].$$

As $K \to \infty$, we have $\mathbb{E}\left[\|C_K^{(2)}\|^2\right] \gg \mathbb{E}\left[\|C_K^{(1)}\|^2\right]$, then

$$\mathbb{E}\left[ \left|\Delta\widehat{L}_{\text{River},K}^{(1)}\right|^2 - \left|\Delta\widehat{L}_{\text{River},K}^{(2)}\right|^2 \right] = \|h_R\|^2\mathbb{E}\left[ \|C_K^{(2)}\|^2 - \|C_K^{(1)}\|^2 \right] \ll 0,$$

which yields $\mathbb{E}[|\Delta\widehat{L}_{\text{River},K}^{(1)}|^2] \ll \mathbb{E}[|\Delta\widehat{L}_{\text{River},K}^{(2)}|^2]$ and demonstrates that *Looped-Attn* achieves a significantly greater loss reduction. Starting from the same initialization, a greater loss reduction implies a lower final loss value:

$$\mathbb{E}[(\widehat{L}_K^{(2)})^2] \ll \mathbb{E}[(\widehat{L}_K^{(1)})^2].$$

During the phase $K = K_2$, *Looped-Attn* has exhibited significant advantages over *Single-Attn*. Furthermore, for subsequent steps $K > K_2$, *Looped-Attn* continues to explore the river downstream while *Single-Attn* remains trapped in the flat valley.

### G.4 PROOF FOR THEOREM 3

We extend the analysis in Theorem 1 and Corollary 1∼2 to the more general loss model introduced in Setting 2:

$$\widehat{L}(\theta_V, \theta_R) = \widehat{L}_{\text{Valley}}(\theta_V) + \widehat{L}_{\text{River}}(\theta_R) + \widehat{L}_{\text{Coupling}}(\theta_V, \theta_R).$$

**Time-Varying Hessian and Dynamics Analysis.** To analyze the optimization dynamics for the general loss, we approximate the landscape locally around each iterate $\theta_k = (\theta_{V,k}, \theta_{R,k})$ using a second-order Taylor expansion. This approximation is justified since each step of GD $\eta\partial\widehat{L}(\theta_k)$ is typically small, the subsequent parameter $\theta_{k+1}$ remains within this neighborhood.

The Taylor expansion of $\widehat{L}(\theta)$ around $\theta_k$ is given by:

$$\widehat{L}(\theta) \approx \widehat{L}(\theta_k) + \partial_\theta \widehat{L}(\theta_k)^\top (\theta - \theta_k) + \frac{1}{2}(\theta - \theta_k)^\top H(\theta_k)(\theta - \theta_k), \qquad (8)$$

where $\partial_\theta \widehat{L}(\theta_k)$ and $H(\theta_k)$ are the gradient and Hessian evaluated at $\theta_k$. And we have:

$$\theta - \theta_k = \begin{pmatrix} \theta_V - \theta_{V,k} \\ \theta_R - \theta_{R,k} \end{pmatrix}, \quad \partial_\theta \widehat{L}(\theta_k) = \begin{pmatrix} \partial_{\theta_V} \widehat{L}(\theta_k) \\ \partial_{\theta_R} \widehat{L}(\theta_k) \end{pmatrix}, \quad H(\theta_k) = \begin{pmatrix} H_{\text{Valley}}(\theta_k) & H_{VR}(\theta_k) \\ H_{RV}(\theta_k) & H_{\text{River}}(\theta_k) \end{pmatrix}.$$

Substituting these into Equation 8 yields the local quadratic approximation:

$$
\begin{aligned}
\widehat{L}(\theta_V, \theta_R) \approx \; & \widehat{L}(\theta_{V,k}, \theta_{R,k}) + \left(\partial_{\theta_V} \widehat{L}(\theta_k)\right)^\top (\theta_V - \theta_{V,k}) + \left(\partial_{\theta_R} \widehat{L}(\theta_k)\right)^\top (\theta_R - \theta_{R,k}) \\
& + \frac{1}{2}(\theta_V - \theta_{V,k})^\top H_{\text{Valley}}(\theta_k)(\theta_V - \theta_{V,k}) \\
& + \frac{1}{2}(\theta_R - \theta_{R,k})^\top H_{\text{River}}(\theta_k)(\theta_R - \theta_{R,k}) \\
& + (\theta_R - \theta_{R,k})^\top H_{RV}(\theta_k)(\theta_V - \theta_{V,k}).
\end{aligned}
$$

From this approximation, we have

$$\partial_{\theta_V} \widehat{L}(\theta_V, \theta_R) \approx \partial_{\theta_V} \widehat{L}(\theta_k) + H_{\text{Valley}}(\theta_k)(\theta_V - \theta_{V,k}) + H_{VR}(\theta_k)(\theta_R - \theta_{R,k}),$$

$$\partial_{\theta_R} \widehat{L}(\theta_V, \theta_R) \approx \partial_{\theta_R} \widehat{L}(\theta_k) + H_{RV}(\theta_k)(\theta_V - \theta_{V,k}),$$

where we assume $H_{\text{River}}(\theta_k) \approx \mathbf{0}$ since river is an extremely flat region.

We find that the river update at the point near $\theta_k$, is approximately linearly dependent $\theta_V$ and the linear coefficient is $H_{RV}(\theta_k)$. Thus we assume that the river gradient at $\theta_k$ is also following:

$$\partial_{\theta_R} \widehat{L}(\theta_{V,k}, \theta_{R,k}) \approx H_{RV}(\theta_k)\theta_{V,k} - h_{R,k}, \qquad (9)$$

where $h_{R,k}$ is the inherent driving force of the river itself, independent of the valley position. This term is similar to the residual term in linear model. Similarly, we assume that the valley gradient at $\theta_k$ is following:

$$\partial_{\theta_V} \widehat{L}(\theta_{V,k}, \theta_{R,k}) \approx H_{\text{Valley}}(\theta_k)\theta_{V,k} + H_{VR}(\theta_k)\theta_{R,k}. \qquad (10)$$

We here assume the valley and river gradient as a linear function of $\theta_V$ or $\theta_R$. Equation 9 corresponds to a first-order Taylor expansion of the gradient function expanded at the river manifold $\tilde{\theta} = (\mathbf{0}, \theta_{R,k})$, *i.e.*,

$$\partial_{\theta_R} \widehat{L}(\theta_{V,k}, \theta_{R,k}) = \partial_{\theta_R} \widehat{L}(0, \theta_{R,k}) + H_{RV}(\theta_k)\theta_{V,k} + \mathbf{R}_1(\theta_{V,k}),$$

where $\partial_{\theta_R} \widehat{L}(0, \theta_{R,k}) \triangleq -h_{R,k}$ is the intrinsic driver force along the river, and $\mathbf{R}_1(\theta_{V,k}) = \frac{1}{2}[\partial_{\theta_V} H_{RV}]\theta_{V,k}^\top \theta_{V,k}$ is the Taylor remainder. The approximation in Equation 9 corresponds to retaining the first two terms and neglecting the remainder. Assuming the Hessian $H_{RV}$ is $\rho_1$-Lipschitz continuous *w.r.t.* $\theta_V$, the remainder satisfies

$$\|\mathbf{R}_1(\theta_{V,k})\| \leq \frac{\rho_1}{2}\|\theta_{V,k}\|^2.$$

We find that the linear term is of order $\mathcal{O}(\|\theta_{V,k}\|)$, while the remainder is of order $\mathcal{O}(\|\theta_{V,k}\|^2)$. Therefore, as the valley parameters $\|\theta_{V,k}\|$ decay during optimization, the error term vanishes at a significantly faster rate than the linear term, ensuring the validity of Equation 9.

Equation 10 corresponds to a first-order Taylor expansion of the gradient function expanded at $\tilde{\theta} = (\mathbf{0}, \mathbf{0})$, *i.e.*,

$$\partial_{\theta_V}\widehat{L}(\theta_{V,k}, \theta_{R,k}) = \partial_{\theta_V}\widehat{L}(0,0) + H_{\text{Valley}}(\theta_k)(\theta_{V,k} - 0) + H_{VR}(\theta_k)(\theta_{R,k} - 0) + \mathbf{R}_2(\theta_k),$$

where $\mathbf{R}_2(\theta_{R,k}) = \frac{1}{2}[\partial_\theta H_{RV}]\theta_{R,k}^\top \theta_{R,k}$ is the Taylor remainder. Assuming the Hessian $H_{RV}$ is $\rho_2$-Lipschitz continuous *w.r.t.* $\theta$, the remainder satisfies

$$\|\mathbf{R}_2(\theta_{V,k})\| \leq \frac{\rho_2}{2}\|\theta_k\|^2.$$

We find that the linear term is of order $\mathcal{O}(\|\theta_k\|)$, while the remainder is of order $\mathcal{O}(\|\theta_k\|^2)$. Therefore, as the parameters $\|\theta_k\|$ decay to minimum during optimization, the error term vanishes at a significantly faster rate than the linear term, ensuring the validity of Equation 10.

Therefore the GD update rules for the two subspaces under general loss are:

$$\theta_{V,k+1} = \theta_{V,k} - \eta\partial_{\theta_V}\widehat{L}(\theta_{V,k}, \theta_{R,k}) \approx (I - \eta H_{\text{Valley}}(\theta_k))\,\theta_{V,k} - \eta H_{VR}(\theta_k)\theta_{R,k}, \quad (11)$$

$$\theta_{R,k+1} = \theta_{R,k} - \eta\partial_{\theta_R}\widehat{L}(\theta_{V,k}, \theta_{R,k}) \approx \theta_{R,k} - \eta\left(H_{RV}(\theta_k)\theta_{V,k} - h_{R,k}\right). \quad (12)$$

Comparing Equation 11$\sim$12 with Equation 1$\sim$2, we find that the optimization dynamics under general loss can be viewed as evolving on a sequence of local quadratic landscapes, each defined by a time-varying Hessian $H(\theta_k)$.

**Derivation of the cumulative change in the river subspace.** Following the same procedure as in the quadratic case, we analyze the cumulative change in the river subspace over $K$ iterations:

$$\Delta\theta_{R,K} = \theta_{R,K} - \theta_{R,0} = \sum_{k=0}^{K-1}(\theta_{R,k+1} - \theta_{R,k})$$

$$\approx \sum_{k=0}^{K-1}\left(-\eta H_{RV}(\theta_k)\theta_{V,k} + \eta h_{R,k}\right)$$

$$\approx \eta\sum_{k=0}^{K-1}h_{R,k} - \eta\sum_{k=0}^{K-1}H_{RV}(\theta_k)\theta_{V,k}. \quad (13)$$

We define $C_{K,\text{gen}}$ to be the cumulative effect induced by the valley dynamics:

$$C_{K,\text{gen}} \triangleq \eta\sum_{k=0}^{K-1}H_{RV}(\theta_k)\theta_{V,k}.$$

The cumulative effect $C_{K,\text{gen}}$ depends on the trajectory of $\theta_{V,k}$. The recurrence for $\theta_{V,k}$ in Equation 11 can be solved as

$$\theta_{V,k} \approx \left(\prod_{j=0}^{k-1}\Phi_j\right)\theta_{V,0} - \eta\sum_{j=0}^{k-1}\left(\prod_{i=j+1}^{k-1}\Phi_i\right)b_j,$$

where $\Phi_k = I - \eta H_{\text{Valley}}(\theta_k)$ and $b_k = H_{VR}(\theta_k)\theta_{R,k}$. Similarly to Appendix G.2, we assume that the unforced update (the first term) is dominant, then

$$\theta_{V,k} \approx \left(\prod_{j=0}^{k-1}\Phi_j\right)\theta_{V,0}.$$

To analyze this, we introduce a effective Hessian $H^B$ with eigenvalues $\{\lambda_i^B\}$ and eigenvectors $\{v_i^B\}$, which satisfies $H^B \preceq H_{\text{Valley}}(\theta_j)$ for all $j$ (Assumption 2). Let $\Phi^B = I - \eta H^B$. This implies that $\|\Phi_j v\| \leq \|\Phi^B v\|$ for any vector $v$. Thus,

$$\|\theta_{V,k}\| \approx \left\|\left(\prod_{j=0}^{k-1}\Phi_j\right)\theta_{V,0}\right\| \leq \left\|(\Phi^B)^k\theta_{V,0}\right\|.$$

Let $H^B = Q^B \Lambda^B (Q^B)^\top$ be the spectral decomposition of this bounding Hessian, where $Q^B = [v_1^B, \ldots, v_{d_V}^B]$ is the orthonormal matrix of eigenvectors and $\Lambda^B = \mathrm{diag}(\lambda_1^B, \ldots, \lambda_{d_V}^B)$ is the diagonal matrix of corresponding eigenvalues. Let $\rho_i^B \triangleq 1 - \eta \lambda_i^B$ be the decay rate of the $i$-th component,

$$\|\theta_{V,k}\| \leq \left\|(\Phi^B)^k \theta_{V,0}\right\| = \left\|\sum_{i=1}^{d_V}(1-\eta\lambda_i^B)^k (v_i^B)^\top \theta_{V,0} v_i^B\right\| = \left\|\sum_{i=1}^{d_V}(\rho_i^B)^k (v_i^B)^\top \theta_{V,0} v_i^B\right\|.$$

Thus, under Assumption 3 and $\|\theta_{V,0}\| \leq \bar{\alpha}$,

$$\|C_{K,\mathrm{gen}}\| \approx \eta \sum_{k=0}^{K-1} \|H_{RV}(\theta_k)\theta_{V,k}\| \leq \eta \sum_{k=0}^{K-1}\left\|\sum_{i=1}^{d_V} H_{RV}(\theta_k)(\rho_i^B)^k (v_i^B)^\top \theta_{V,0} v_i^B\right\|$$

$$\leq \eta \sum_{k=0}^{K-1}\sum_{i=1}^{d_V} \|H_{RV}(\theta_k)v_i^B\|\,\|(\rho_i^B)^k (v_i^B)^\top \theta_{V,0}\|$$

$$\leq \eta \sum_{k=0}^{K-1}\sum_{i=1}^{d_V} \bar{h}_{\mathrm{gen}}\,\|(\rho_i^B)^k (v_i^B)^\top \theta_{V,0}\|$$

$$\leq \eta\,\bar{h}_{\mathrm{gen}} \sum_{k=0}^{K-1}\left(\sum_{i=1}^{d_V} |\rho_i^B|^k |(v_i^B)^\top \theta_{V,0}|\right)$$

$$= \eta\,\bar{h}_{\mathrm{gen}} \sum_{i=1}^{d_V} |(v_i^B)^\top \theta_{V,0}|\left(\sum_{k=0}^{K-1} |\rho_i^B|^k\right)$$

$$= \bar{h}_{\mathrm{gen}} \sum_{i=1}^{d_V} |(v_i^B)^\top \theta_{V,0}|\left(\frac{|1-(\rho_i^B)^K|}{|\lambda_i^B|}\right)$$

$$\leq \bar{h}_{\mathrm{gen}} \sum_{i=1}^{d_V} \frac{|(v_i^B)^\top \theta_{V,0}|}{|\lambda_i^B|}$$

$$\leq \sqrt{d_V}\,\bar{h}_{\mathrm{gen}}\,\bar{\alpha} \sum_{i=1}^{d_V} \frac{1}{|\lambda_i^B|} \triangleq \mathcal{C}_{\mathrm{gen}}.$$

It means that after $K$ iterations, the driving force from valley is limited to $\mathcal{C}_{\mathrm{gen}}$, determined by two factors: (a) $\bar{h}_{\mathrm{gen}}$, the supremum of the coupling strength which represents the most efficient effect of valley on the river; (b) $\{\lambda_i^B\}$, the eigenspectrum of valley subspace.

With the expression of cumulative force $\mathcal{C}_{\mathrm{gen}} = \sqrt{d_V}\,\bar{h}_{\mathrm{gen}}\,\bar{\alpha} \sum_{i=1}^{d_V} \frac{1}{|\lambda_i^B|}$, we compare two models.

The spectral experiments presented in Figure 27 (with $\epsilon = 0.02$) reveal that *Looped-Attn* exhibits a larger $\mathcal{E}(H_{\mathrm{Valley}})$ than *Single-Attn*. Thus with Definition 1, we summarize the characteristics of two models in Conjecture 1$\sim$2.

For *Single-Attn* with River-U-Valley, we have $\frac{1}{d_V^{(1)}} \sum_{i=1}^{d_V^{(1)}} \frac{1}{(\lambda_i^{B(1)})^2} \leq \zeta$. With inequality $\|x\|_1 \leq \sqrt{d}\|x\|_2$ for vector $x \in \mathbb{R}^d$, the maximal cumulative force satisfies

$$\mathcal{C}_{\mathrm{gen}}^{(1)} = \bar{h}_{\mathrm{gen}}\bar{\alpha}\sqrt{d_V^{(1)}} \sum_{i=1}^{d_V^{(1)}} \frac{1}{|\lambda_i^{B(1)}|} \leq \bar{h}_{\mathrm{gen}}\bar{\alpha}\sqrt{d_V^{(1)}}\sqrt{d_V^{(1)}}\sqrt{\sum_{i=1}^{d_V^{(1)}} \frac{1}{(\lambda_i^{B(1)})^2}} \leq \bar{h}_{\mathrm{gen}}\bar{\alpha}(d_V^{(1)})^{3/2}\sqrt{\zeta}.$$

For *Looped-Attn* with River-V-Valley, we have $\frac{1}{d_V^{(2)}} \sum_{i=1}^{d_V^{(2)}} \frac{1}{(\lambda_i^{B(2)})^2} \gg \zeta$. With inequality $\|x\|_1 \geq \|x\|_2$ for vector $x \in \mathbb{R}^d$, the maximal cumulative force satisfies

$$\mathcal{C}_{\mathrm{gen}}^{(2)} = \bar{h}_{\mathrm{gen}}\bar{\alpha}\sqrt{d_V^{(2)}} \sum_{i=1}^{d_V^{(2)}} \frac{1}{|\lambda_i^{B(2)}|} \geq \bar{h}_{\mathrm{gen}}\bar{\alpha}\sqrt{d_V^{(2)}}\sqrt{\sum_{i=1}^{d_V^{(2)}} \frac{1}{(\lambda_i^{B(2)})^2}} \gg \bar{h}_{\mathrm{gen}}\bar{\alpha}d_V^{(2)}\sqrt{\zeta}.$$

The valley dimensions of two model are typically of the same order, thus we conclude that

$$\mathcal{C}_{\text{gen}}^{(2)} \gg \mathcal{C}_{\text{gen}}^{(1)}.$$

In summary, under general loss, we prove that the V-shaped valley in *Looped-Attn* provides a larger potential for driving exploration in the river subspace.

**In the following, similar to Corollary 2, we can also connect with loss values.**

With the general loss form in Setting 2,

$$\widehat{L}(\theta_V, \theta_R) = \widehat{L}_{\text{Valley}}(\theta_V) + \widehat{L}_{\text{River}}(\theta_R) + \widehat{L}_{\text{Coupling}}(\theta_V, \theta_R).$$

Recall that, as $K \to \infty$ $\|C_{K,\text{gen}}\| \leq \bar{h}_{\text{gen}} \sum_{i=1}^{d_V} \frac{|(v_i^B)^\top \theta_{V,0}|}{|\lambda_i^B|}$, let $c_i^B = (v_i^B)^\top \theta_{V,0}$. With $\theta_{V,0} \sim \mathcal{N}(0, \bar{\alpha}^2 I / d_V)$ in Setting 2, we have

$$\mathbb{E}[c_i^B c_j^B] = \mathbb{E}\left[ (v_i^B)^\top \theta_{V,0} \theta_{V,0}^\top v_j^B \right] = (v_i^B)^\top \mathbb{E}\left[ \theta_{V,0} \theta_{V,0}^\top \right] v_j^B.$$

If $i = j$, $\mathbb{E}[(c_i^B)^2] = \bar{\alpha}^2 / d_V$. If $i \neq j$, $\mathbb{E}[(c_i^B)^2] = 0$. Then taking expectation over initialization, we have

$$\mathbb{E}\left[ \|C_{K,\text{gen}}\|^2 \right] \leq \mathbb{E}\left[ \bar{h}_{\text{gen}}^2 \left\langle \sum_{i=1}^{d_V} \frac{|(v_i^B)^\top \theta_{V,0}|}{|\lambda_i^B|}, \sum_{j=1}^{d_V} \frac{|(v_j^B)^\top \theta_{V,0}|}{|\lambda_j^B|} \right\rangle \right]$$

$$= \mathbb{E}\left[ \bar{h}_{\text{gen}}^2 \left\langle \sum_{i=1}^{d_V} \frac{|(c_i^B)|}{|\lambda_i^B|}, \sum_{j=1}^{d_V} \frac{|c_j^B|}{|\lambda_j^B|} \right\rangle \right]$$

$$= \bar{h}_{\text{gen}}^2 \sum_{i=1}^{d_V} \sum_{j=1}^{d_V} \frac{\mathbb{E}\left[ c_i^B c_j^B \right]}{\lambda_i^B \lambda_j^B}$$

$$= \bar{h}_{\text{gen}}^2 \sum_{i=1}^{d_V} \frac{\mathbb{E}\left[ (c_i^B)^2 \right]}{(\lambda_i^B)^2}$$

$$= \frac{\bar{\alpha}^2}{d_V} \bar{h}_{\text{gen}}^2 \sum_{i=1}^{d_V} \frac{1}{(\lambda_i^B)^2}.$$

We introduce Hessian $H^T$ with eigenvalues $\{\lambda_i^T\}$ and eigenvectors $\{v_i^T\}$, which satisfies $H_{\text{Valley}}(\theta_j) \preceq H^T$ for all $j$ (Assumption 2).

$$\|\theta_{V,k}\| \geq \|(\Phi^T)^k \theta_{V,0}\| = \left\| \sum_{i=1}^{d_V} (1 - \eta \lambda_i^T)^k (v_i^T)^\top \theta_{V,0} v_i^T \right\| = \left\| \sum_{i=1}^{d_V} (\rho_i^T)^k (v_i^T)^\top \theta_{V,0} v_i^T \right\|.$$

With Assumption 3, we can derive the lower bound of $\mathbb{E}\left[ \|C_{K,\text{gen}}\|^2 \right]$.

$$\mathbb{E}\left[ \|C_{K,\text{gen}}\|^2 \right] = \mathbb{E}\left[ \left\| \eta \sum_{k=0}^{K-1} H_{RV}(\theta_k) \theta_{V,k} \right\|^2 \right]$$

$$= \eta^2 \mathbb{E}\left[ \left\langle \sum_{k=0}^{K-1} H_{RV}(\theta_k) \theta_{V,k}, \sum_{t=0}^{K-1} H_{RV}(\theta_t) \theta_{V,t} \right\rangle \right]$$

$$= \eta^2 \mathbb{E}\left[ \sum_{k=0}^{K-1} \sum_{t=0}^{K-1} \theta_{V,k}^\top H_{RV}^\top(\theta_k) H_{RV}(\theta_t) \theta_{V,t} \right]$$

$$\geq \eta^2 \mathbb{E}\left[ \sum_{k=0}^{K-1} \theta_{V,k}^\top (\underline{H})^\top \underline{H} \theta_{V,k} \right]$$

$$\geq \eta^2 \mathbb{E}\left[ \sum_{k=0}^{K-1} \left( \sum_{i=1}^{d_V} (\rho_i^T)^k (v_i^T)^\top \theta_{V,0} v_i^T \right)^\top (\underline{H})^\top \underline{H} \left( \sum_{j=1}^{d_V} (\rho_j^T)^k (v_j^T)^\top \theta_{V,0} v_j^T \right) \right],$$

where the last inequality holds due to the stable valley eigenvectors in Assumption 2. As $K \to \infty$, we have

$$\mathbb{E}\left[\|C_{K,\text{gen}}\|^2\right] \geq \mathbb{E}\left[\left(\sum_{i=1}^{d_V} \frac{1}{\lambda_i^T}(v_i^T)^\top \theta_{V,0} v_i^T\right)^\top (\underline{H})^\top \underline{H} \left(\sum_{j=1}^{d_V} \frac{1}{\lambda_j^T}(v_j^T)^\top \theta_{V,0} v_j^T\right)\right]$$

$$= \mathbb{E}\left[\left(\sum_{i=1}^{d_V} \frac{1}{\lambda_i^T}\underline{H}(v_i^T)^\top \theta_{V,0} v_i^T\right)^\top \left(\sum_{j=1}^{d_V} \frac{1}{\lambda_j^T}\underline{H}(v_j^T)^\top \theta_{V,0} v_j^T\right)\right]$$

$$= \left(\sum_{i=1}^{d_V} \frac{c_i}{\lambda_i^T}u_i\right)^\top \left(\sum_{j=1}^{d_V} \frac{c_j}{\lambda_j^T}u_j\right),$$

where $c_i = (v_i^T)^\top \theta_{V,0} \in \mathbb{R}$ and $u_i = \underline{H}v_i^T \in \mathbb{R}^{d_R}$. Furthermore,

$$\mathbb{E}\left[\|C_{K,\text{gen}}\|^2\right] = \sum_{i=1}^{d_V} \frac{\mathbb{E}[c_i^2]}{(\lambda_i^T)^2}\|u_i\|^2 = \frac{\bar{\alpha}^2}{d_V}\sum_{i=1}^{d_V} \frac{\|u_i\|^2}{(\lambda_i^T)^2}$$

$$\geq \frac{\bar{\alpha}^2}{d_V}h_{\text{gen}}^2 \sum_{i=1}^{d_V} \frac{1}{(\lambda_i^T)^2}.$$

Thus,

$$\mathbb{E}\left[\|C_{K,\text{gen}}^{(2)}\|^2\right] \geq \frac{\bar{\alpha}^2}{d_V^{(2)}}h_{\text{gen}}^2 \sum_{i=1}^{d_V^{(2)}} \frac{1}{(\lambda_i^{T(2)})^2}, \quad \frac{\bar{\alpha}^2}{d_V}\bar{h}_{\text{gen}}^2 \sum_{i=1}^{d_V} \frac{1}{(\lambda_i^{B(1)})^2} \geq \mathbb{E}\left[\|C_{K,\text{gen}}^{(1)}\|^2\right].$$

With Definition 1 and Assumption 4, it leads to

$$\mathbb{E}\left[\|C_{K,\text{gen}}^{(2)}\|^2\right] \gg \mathbb{E}\left[\|C_{K,\text{gen}}^{(1)}\|^2\right].$$

Let $K$ be a number of iterations large enough such that the valley parameters for both models have converged to the bottom of their respective valleys.

The well-conditioned U-shaped valley of *Single-Attn* leads to converge rapidly in the valley subspace (within $K_1$ steps). The ill-conditioned V-shaped valley of *Looped-Attn* leads to slower convergence in the valley (within $K_2$ steps, where $K_2 > K_1$). We consider $K = K_2$.

At iteration $K$, for both models, the valley parameters are $\theta_{V,K}^{(1)} \approx \mathbf{0}$ and $\theta_{V,K}^{(2)} \approx \mathbf{0}$. Thus, $\widehat{L}_K^{(1)} \approx \widehat{L}_{\text{River},K}^{(1)}$ and $\widehat{L}_K^{(2)} \approx \widehat{L}_{\text{River},K}^{(2)}$.

We then analyze the change in the river loss, $\Delta\widehat{L}_{\text{River},K} = \widehat{L}_{\text{River},K} - \widehat{L}_{\text{River},0}$. With the Taylor expansion of $\widehat{L}(\theta)$ around $\theta_k$,

$$\widehat{L}(\theta) \approx \widehat{L}(\theta_k) + \partial_\theta \widehat{L}(\theta_k)^\top(\theta - \theta_k) + \frac{1}{2}(\theta - \theta_k)^\top H(\theta_k)(\theta - \theta_k). \tag{14}$$

Substitute $\theta = \theta_{k+1}$ and $\theta_{k+1} = \theta_k - \eta\partial_\theta\widehat{L}(\theta_k)$, we have

$$\widehat{L}(\theta_{k+1}) \approx \widehat{L}(\theta_k) - \eta\partial_\theta\widehat{L}(\theta_k)^\top \partial_\theta\widehat{L}(\theta_k) + \frac{\eta^2}{2}\widehat{L}(\theta_k)^\top H(\theta_k)\widehat{L}(\theta_k). \tag{15}$$

With a small learning rate, we approximate the above as $\widehat{L}(\theta_{k+1}) \approx \widehat{L}(\theta_k) - \eta\partial_\theta\widehat{L}(\theta_k)^\top \partial_\theta\widehat{L}(\theta_k)$. Thus

$$\Delta\widehat{L}_{\text{River},K} = \sum_{k=0}^{K-1}\left(\widehat{L}_{\text{River}}(\theta_{R,k+1}) - \widehat{L}_{\text{River}}(\theta_{R,k})\right) \approx -\eta\sum_{k=0}^{K-1}\left\|\partial_{\theta_R}\widehat{L}(\theta_k)\right\|^2.$$

From Equation 12, $\partial_{\theta_R}\widehat{L}(\theta_{V,k}, \theta_{R,k}) \approx H_{RV}(\theta_k)\theta_{V,k} - h_{R,k}$, we have

$$\Delta\widehat{L}_{\text{River},K} \approx -\eta \sum_{k=0}^{K-1} \|H_{RV}(\theta_k)\theta_{V,k} - h_{R,k}\|^2$$

$$= -\eta \sum_{k=0}^{K-1} \|H_{RV}(\theta_k)\theta_{V,k}\|^2 - \eta \sum_{k=0}^{K-1} \|h_{R,k}\|^2 + 2\eta \sum_{k=0}^{K-1} (h_{R,k})^\top (H_{RV}(\theta_k)\theta_{V,k}).$$

Assume that the river inherent gradient $h_R$ is the same during training for both models,

$$\mathbb{E}\left[\left(\Delta\widehat{L}_{\text{River},K}^{(1)}\right)^2 - \left(\Delta\widehat{L}_{\text{River},K}^{(2)}\right)^2\right]$$

$$=\mathbb{E}\left[\eta^2 \sum_{k=0}^{K-1} \left(\left\|H_{RV}^{(1)}(\theta_k^{(1)})\theta_{V,k}^{(1)}\right\|^4 - \left\|H_{RV}^{(2)}(\theta_k^{(2)})\theta_{V,k}^{(2)}\right\|^4\right)\right]$$

$$+ 4\eta^2 \sum_{k=0}^{K-1} \left([(h_{R,k}^{(1)})^\top (H_{RV}^{(1)}(\theta_k^{(1)})\theta_{V,k}^{(1)})]^2 - [(h_{R,k}^{(2)})^\top (H_{RV}^{(2)}(\theta_k^{(2)})\theta_{V,k}^{(2)})]^2\right)$$

$$=\mathbb{E}\left[\|C_{K,\text{gen}}^{(1)}\|^2 - \|C_{K,\text{gen}}^{(2)}\|^2\right]\mathbb{E}\left[\|C_{K,\text{gen}}^{(1)}\|^2 + \|C_{K,\text{gen}}^{(2)}\|^2\right] + 4\|h_R\|^2\mathbb{E}\left[\|C_{K,\text{gen}}^{(1)}\|^2 - \|C_{K,\text{gen}}^{(2)}\|^2\right].$$

As $K \to \infty$, we have $\mathbb{E}\left[\|C_{K,\text{gen}}^{(1)}\|^2\right] \ll \mathbb{E}\left[\|C_{K,\text{gen}}^{(2)}\|^2\right]$, then

$$\mathbb{E}\left[\left(\Delta\widehat{L}_{\text{River},K}^{(1)}\right)^2 - \left(\Delta\widehat{L}_{\text{River},K}^{(2)}\right)^2\right] \ll 0,$$

which yields $\mathbb{E}[(\Delta\widehat{L}_{\text{River},K}^{(1)})^2] \ll \mathbb{E}[(\Delta\widehat{L}_{\text{River},K}^{(2)})^2]$ and demonstrates that *Looped-Attn* achieves a significantly greater loss reduction. Starting from the same initialization, a greater loss reduction implies a lower final loss value $\mathbb{E}[(\widehat{L}_{\text{River},K}^{(2)})^2] \ll \mathbb{E}[(\widehat{L}_{\text{River,K}}^{(1)})^2]$, then

$$\mathbb{E}[(\widehat{L}_K^{(2)})^2] \ll \mathbb{E}[(\widehat{L}_K^{(1)})^2].$$

During the phase $K = K_2$, *Looped-Attn* has exhibited significant advantages over *Single-Attn*. Furthermore, for subsequent steps $K > K_2$, *Looped-Attn* continues to explore the river downstream while *Single-Attn* remains trapped in the flat valley.

## H  SHARED RIVER UPSTREAM

### H.1  ASSUMPTIONS AND USEFUL LEMMAS

#### H.1.1  ASSUMPTIONS

**Assumption 5** (**Diagonally Dominant and PSD Weight Matrices**). *Assume that the key, query, and value weight matrices* $(W_K, W_Q, W_V)$ *are diagonally dominant with Positive Semidefinite (PSD) diagonal matrices* $D_K$, $D_Q$, $D_V$, *we have*

$$W_K = D_K + \epsilon_K,$$
$$W_Q = D_Q + \epsilon_Q,$$
$$W_V = D_V + \epsilon_V,$$

*where* $\epsilon_K$, $\epsilon_Q$, $\epsilon_V$ *are dense matrices with significantly smaller spectral norm.*

**Remark 10** (**Justification of Assumption 5**). This assumption provides mathematical tractability for the formal analysis of composite matrix transformations, which is necessary for proving the positive alignment of gradients. Intuitively, it may approximate the behavior of the attention mechanism during the early stages of training, where the model first learns simple local dependencies before capturing more complex global interactions.

This assumption represents a relative idealization. In practice, the weight matrices of a well-trained, deep Transformer are typically dense and are not guaranteed to be PSD. In a practical setting, we posit that the gradients are more likely to be broadly aligned or at least non-negatively correlated, particularly during the initial phase of training. This weaker form of alignment is sufficient to support the theoretical basis for our SHIFT framework, ensuring that the parameters learned by *Single-Attn* provide a beneficial starting point for *Looped-Attn*.

**Assumption 6** (**Approximate PSD Property of Composite Transformations**). *Let* $D_A$ *and* $D_B$ *be Positive Semidefinite (PSD) diagonal matrices, and let* $P$ *be a general PSD matrix. We assume that their product,* $M = D_A P D_B$, *is approximately PSD. This means the matrix* $M$ *can be decomposed as:*

$$M = M_{PSD} + \epsilon,$$

*where* $M_{PSD}$ *is a PSD matrix that captures the dominant, direction-preserving behavior of the transformation, and* $\epsilon$ *is a perturbation matrix with a small norm relative to* $M_{PSD}$.

**Remark 11** (**Justification for Assumption 6**). A matrix is strictly PSD only if it is symmetric and its quadratic form is non-negative for all vectors. The composite transformation $M = D_A P D_B$ generally fails the first symmetric condition, and in rare extreme cases, may fail the second. The perturbation term $\epsilon$ accounts for these two sources of deviation from strict PSD properties.

**(a) Minor Fluctuation from Non-Symmetry.** The primary deviation arises from the non-commutativity of matrix multiplication, which breaks symmetry. The transpose of $M$ is $M^\top = D_B P D_A$, which is generally not equal to $M$. Therefore, $M$ is not symmetric. In a well-behaved system, we assume that this non-symmetry only introduce minor fluctuations rather than fundamentally altering the transformation's property.

**(b) Non-PSD Behavior from Extreme Anisotropic Scaling.** Another possible deviation can occur even in the symmetric part of $M$, *i.e.*, $M_{\text{sym}} = \frac{1}{2}(D_A P D_B + D_B P D_A)$. While the composition of direction-preserving operators $(D_A, P, D_B)$ is intuitively expected to remain direction-preserving, it is possible to construct extreme counterexamples. Such cases arises when the diagonal matrices $D_A$ and $D_B$ induce extreme anisotropic scaling (*i.e.*, some diagonal entries are very large while others are near-zero). This can significantly alter the direction of an arbitrary vector before and after the application of $P$, leading to a negative quadratic form. Our assumption posits that during the initial stage of training attention models, such extreme conditions are not common. We model these rare non-PSD behaviors as part of the small perturbation $\epsilon$, allowing our analysis to focus on the system's dominant, approximately PSD behavior captured by $M_{\text{PSD}}$.

#### H.1.2  GRADIENT CALCULATIONS

In this section, we present two key lemmas regarding the gradients of the cross-entropy loss function with respect to the key $(W_K)$ and query $(W_Q)$ matrices for the *Single-Attn* and *Looped-Attn* models.

**Lemma 1.** *For the Single-Attn model, the gradients of the empirical loss $\widehat{L}(\theta)$ with respect to the key matrix $W_K$ and query matrix $W_Q$ are given by:*

$$\nabla_{W_K}\widehat{L}(\theta) = \widehat{\mathbb{E}}\left[(A^\top \otimes b)W_h^\top(\mathsf{S}(\hat{y}) - \mathbf{e}_y)\right],$$

$$\nabla_{W_Q}\widehat{L}(\theta) = \widehat{\mathbb{E}}\left[(\tilde{b} \otimes \widetilde{A}^\top)W_h^\top(\mathsf{S}(\hat{y}) - \mathbf{e}_y)\right],$$

*where $A = W_V E_0 E_0^\top \in \mathbb{R}^{d\times d}$, $b = W_Q z_0 \in \mathbb{R}^d$, and $\widetilde{A} = W_V E_0 E_0^\top W_K^\top \in \mathbb{R}^{d\times d}$, $\tilde{b} = z_0 \in \mathbb{R}^d$.*

**Remark 12.** Recall that, $E_0 \in \mathbb{R}^{d\times n}$ is the input embedding matrix, $z_0 \in \mathbb{R}^d$ is the query vector, which is the last column of embedding $E_0$. $W_K, W_Q, W_V \in \mathbb{R}^{d\times d}$ are the key, query, and value weight matrices, respectively. $W_h$ is the prediction head parameters. Furthermore, $\mathsf{S}(\hat{y})$ represents the softmax probability vector of the logits, and $\mathbf{e}_y = [0, \cdots, 1, \cdots, 0]^\top$, *i.e.*, the value of the $y$-th component is 1, and 0 otherwise. The operator $\otimes$ denotes the Kronecker product.

*Proof.* Our objective is to compute the gradients $\nabla_{W_K}\widehat{L}(\theta)$ and $\nabla_{W_Q}\widehat{L}(\theta)$ for the *Single-Attn* model. We begin by recalling the definition of the empirical loss and the architecture of the *Single-Attn* model. The loss for one sequence is given by $\hat{l} = -\log(\mathsf{S}_y(\hat{y}))$, where logits $\hat{y} = W_h f_\theta(E_0, z_0)$ and the linear attention function is $f_\theta(E_0, z_0) = W_V E_0 E_0^\top W_K^\top W_Q z_0$. The overall empirical loss $\widehat{L}(\theta)$ is averaging over the training set.

The gradient calculations require the chain rule, which is summarized as follows:

1. The loss $\hat{l}$ is a function of the logit vector $\hat{y}$.

2. The logit vector $\hat{y}$ is a function of the final state $z_1$.

3. The final state $z_1$ is function of the attention output $f_\theta(E_0, z_0)$.

4. The attention output $f_\theta(E_0, z_0)$ is a function of the model parameters $W_K$ and $W_Q$.

We will compute the gradient for each component of the chain rule individually.

**Step 1: Gradient with respect to the logit vector $\hat{y}$.** We first compute the derivative of $\hat{l}$ with respect to an individual logit component $\hat{y}_k$. The softmax probability for the ground-truth token $y$ is defined as:

$$\log(\mathsf{S}_y(\hat{y})) = \log\left(\frac{e^{\hat{y}_y}}{\sum_{j=1}^V e^{\hat{y}_j}}\right) = \hat{y}_y - \log\sum_{j=1}^V e^{\hat{y}_j}.$$

When $k = y$,

$$\frac{\partial \log(\mathsf{S}_y(\hat{y}))}{\partial \hat{y}_y} = 1 - \frac{e^{\hat{y}_y}}{\sum_{j=1}^V e^{\hat{y}_j}} = 1 - \mathsf{S}_y(\hat{y}).$$

When $k \neq y$,

$$\frac{\partial \log(\mathsf{S}_y(\hat{y}))}{\partial \hat{y}_k} = 0 - \frac{e^{\hat{y}_k}}{\sum_{j=1}^V e^{\hat{y}_j}} = -\mathsf{S}_k(\hat{y}).$$

Combining these results,

$$\nabla_{\hat{y}}\log(\mathsf{S}_y(\hat{y})) = \mathbf{e}_y - \mathsf{S}(\hat{y}),$$

where $\mathbf{e}_y$ is a one-hot vector with a 1 at the position corresponding to the ground-truth token $y$, $\mathsf{S}(\hat{y})$ represents the softmax probability vector of the logits. Therefore, the gradient of the loss $\hat{l}$ with respect to $\hat{y}$ is:

$$\nabla_{\hat{y}}\hat{l} = -(\mathbf{e}_y - \mathsf{S}(\hat{y})) = \mathsf{S}(\hat{y}) - \mathbf{e}_y.$$

**Step 2: Gradient with respect to the final state $z_1$.** We then compute the derivative of the logit vector $\hat{y}$ with respect to the state $z_1$. With $\hat{y} = W_h z_1$, we have

$$\frac{\partial \hat{y}}{\partial z_1} = \frac{\partial W_h z_1}{\partial z_1} = W_h^\top.$$

**Step 3: Gradient with respect to the key matrix $W_K$.** We now compute the derivative of the final state $z_1$ with respect to the key matrix $W_K$.

With $z_1 = z_0 + f_\theta(E_0, z_0)$, we have

$$\frac{\partial z_1}{\partial W_K} = \frac{\partial f_\theta(E_0, z_0)}{\partial W_K} \frac{\partial z_1}{\partial f_\theta(E_0, z_0)} = \frac{\partial f_\theta(E_0, z_0)}{\partial W_K} \in \mathbb{R}^{d^2 \times d}.$$

Define $A = W_V E_0 E_0^\top \in \mathbb{R}^{d \times d}$ and $b = W_Q z_0 \in \mathbb{R}^d$. The attention function $f_\theta(E_0, z_0) = W_V E_0 E_0^\top W_K^\top W_Q z_0$ simplifies to $f_\theta(E_0, z_0) = A W_K^\top b$. We get

$$\frac{\partial f_\theta(E_0, z_0)}{\partial W_K} = \frac{\partial (A W_K^\top b)}{\partial W_K} = A^\top \otimes b \in \mathbb{R}^{d^2 \times d},$$

where $\otimes$ denotes the Kronecker product. Thus,

$$\frac{\partial z_1}{\partial W_K} = A^\top \otimes b \in \mathbb{R}^{d^2 \times d}.$$

Combining the above steps using the chain rule, we have

$$\nabla_{W_K} \widehat{L}(\theta) = \widehat{\mathbb{E}} \left[ -\nabla_{W_K} \log (\mathbf{S}_y(\hat{y})) \right]$$

$$= \widehat{\mathbb{E}} \left[ -\frac{\partial z_1}{\partial W_K} \frac{\partial \hat{y}}{\partial z_1} \nabla_{\hat{y}} \log(\mathbf{S}_y(\hat{y})) \right]$$

$$= \widehat{\mathbb{E}} \left[ (A^\top \otimes b) W_h^\top (\mathbf{S}(\hat{y}) - \mathbf{e}_y) \right],$$

where $A = W_V E_0 E_0^\top \in \mathbb{R}^{d \times d}, b = W_Q z_0 \in \mathbb{R}^d$.

**Step 4: Gradient with respect to the query matrix $W_Q$.** The process for computing the gradient with respect to $W_Q$ is similar.

Define $\widetilde{A} = W_V E_0 E_0^\top W_K^\top \in \mathbb{R}^{d \times d}$ and $\tilde{b} = z_0 \in \mathbb{R}^d$. The attention function $f_\theta(E_0, z_0) = W_V E_0 E_0^\top W_K^\top W_Q z_0$ can be written as $f_\theta(E_0, z_0) = \widetilde{A} W_Q \tilde{b}$. We have

$$\frac{\partial f_\theta(E_0, z_0)}{\partial W_Q} = \frac{\partial (\widetilde{A} W_Q \tilde{b})}{\partial W_Q} = \tilde{b} \otimes \widetilde{A}^\top \in \mathbb{R}^{d^2 \times d}.$$

Thus,

$$\frac{\partial z_1}{\partial W_Q} = \tilde{b} \otimes \widetilde{A}^\top \in \mathbb{R}^{d^2 \times d}.$$

Again, applying the chain rule by combining this result with Step 1 and Step 2, we have

$$\nabla_{W_Q} \widehat{L}(\theta) = \widehat{\mathbb{E}} \left[ -\nabla_{W_Q} \log (\mathbf{S}_y(\hat{y})) \right]$$

$$= \widehat{\mathbb{E}} \left[ -\frac{\partial z_1}{\partial W_Q} \frac{\partial \hat{y}}{\partial z_1} \nabla_{\hat{y}} \log(\mathbf{S}_y(\hat{y})) \right]$$

$$= \widehat{\mathbb{E}} \left[ (\tilde{b} \otimes \widetilde{A}^\top) W_h^\top (\mathbf{S}(\hat{y}) - \mathbf{e}_y) \right],$$

where $\widetilde{A} = W_V E_0 E_0^\top W_K^\top \in \mathbb{R}^{d \times d}, \tilde{b} = z_0 \in \mathbb{R}^d$. $\qquad \square$

**Lemma 2.** *For the Looped-Attn model, the gradients of the empirical loss $\widehat{L}(\theta)$ with respect to the key matrix $W_K$ and the query matrix $W_Q$ are given by:*

$$\nabla_{W_K} \widehat{L}(\theta) = \widehat{\mathbb{E}} \left[ \sum_{t=1}^T \left( A_{t-1}^\top \otimes b_{t-1} \right) W_h^\top (\mathbf{S}(\hat{y}) - \mathbf{e}_y) \right],$$

$$\nabla_{W_Q} \widehat{L}(\theta) = \widehat{\mathbb{E}} \left[ \sum_{t=1}^T \left( \tilde{b}_{t-1} \otimes \widetilde{A}_{t-1}^\top \right) W_h^\top (\mathbf{S}(\hat{y}) - \mathbf{e}_y) \right],$$

*where $A_{t-1} = W_V E_{t-1} E_{t-1}^\top \in \mathbb{R}^{d \times d}$, $b_{t-1} = W_Q z_{t-1} \in \mathbb{R}^d$, and $\widetilde{A}_{t-1} = W_V E_{t-1} E_{t-1}^\top W_K^\top \in \mathbb{R}^{d \times d}$, $\tilde{b}_{t-1} = z_{t-1} \in \mathbb{R}^d$ for each loop iteration $t$.*

**Remark 13.** Recall that, $E_{t-1}$ and $z_{t-1}$ are the intermediate representations during looping. The representations are updated by $z_t = z_{t-1} + f_\theta(E_{t-1}, z_{t-1})$ and $E_t = E_{t-1} + f_\theta(E_{t-1})$. Furthermore, $W_K, W_Q, W_V \in \mathbb{R}^{d \times d}$ are the key, query, and value weight matrices, respectively. $W_h$ is the prediction head parameters. $\mathsf{S}(\hat{y})$ represents the softmax probability vector of the logits, and $\mathbf{e}_y = [0, \cdots, 1, \cdots, 0]^\top$, *i.e.*, the value of the $y$-th component is 1, and 0 otherwise. The operator $\otimes$ denotes the Kronecker product.

*Proof.* We aim to compute the gradients $\nabla_{W_K} \widehat{L}(\theta)$ and $\nabla_{W_Q} \widehat{L}(\theta)$ for the *Looped-Attn* model. The final logit vector $\hat{y}$ is produced by applying the prediction head $W_h$ to the final state $z_T$, which is obtained by $T$ loops of the attention function.

The gradient calculations require the chain rule, which is summarized as follows:

1. The loss $\hat{l}$ is a function of the logit vector $\hat{y}$.

2. The logit vector $\hat{y}$ is a function of the final state $z_T$.

3. The final state $z_T$ is a function of the attention outputs $f_\theta(E_{t-1}, z_{t-1})$ from all preceding steps $t = 1, \ldots, T$.

4. Each attention output $f_\theta(E_{t-1}, z_{t-1})$ is a function of the model parameters $W_K$ and $W_Q$.

We proceed by computing the gradient for each component in this chain.

**Step 1: Gradient with respect to the logit vector $\hat{y}$.** This step is identical to the derivation for the *Single-Attn* model. The gradient of the loss $\hat{l}$ with respect to the logit vector $\hat{y}$ is:

$$\nabla_{\hat{y}} \hat{l} = \mathsf{S}(\hat{y}) - \mathbf{e}_y,$$

where $\mathsf{S}(\hat{y})$ is the softmax probability vector and $\mathbf{e}_y$ is the one-hot vector for the ground-truth token.

**Step 2: Gradient with respect to the final state $z_T$.** With $\hat{y} = W_h z_T$, we have

$$\frac{\partial \hat{y}}{\partial z_T} = \frac{\partial W_h z_T}{\partial z_T} = W_h^\top.$$

**Step 3: Gradient with respect to the key matrix $W_K$.** With the iteration $z_t = z_{t-1} + f_\theta(E_{t-1}, z_{t-1})$, we can derive a recursive defintion

$$z_T = z_0 + \sum_{t=1}^{T} f_\theta(E_{t-1}, z_{t-1}).$$

Then we have

$$\frac{\partial z_T}{\partial W_K} = \sum_{t=1}^{T} \frac{\partial f_\theta(E_{t-1}, z_{t-1})}{\partial W_K} \frac{\partial z_T}{\partial f_\theta(E_{t-1}, z_{t-1})}$$

$$= \sum_{t=1}^{T} \frac{\partial f_\theta(E_{t-1}, z_{t-1})}{\partial W_K} \in \mathbb{R}^{d^2 \times d}.$$

The derivative of the attention function $f_\theta(E_{t-1}, z_{t-1}) = W_V E_{t-1} E_{t-1}^\top W_K^\top W_Q z_{t-1}$ with respect to $W_K$ is structurally identical to the *Single-Attn* case, but with time-dependent inputs.

Define $A_{t-1} = W_V E_{t-1} E_{t-1}^\top$ and $b_{t-1} = W_Q z_{t-1}$ for each loop $t \in [T]$, then

$$\frac{\partial f_\theta(E_{t-1}, z_{t-1})}{\partial W_K} = A_{t-1}^\top \otimes b_{t-1}.$$

Thus

$$\frac{\partial z_T}{\partial W_K} = \sum_{t=1}^{T} (A_{t-1}^\top \otimes b_{t-1}) \in \mathbb{R}^{d^2 \times d}.$$

Combining the above results using the chain rule, we have

$$\nabla_{W_K}\widehat{L}(\theta) = \widehat{\mathbb{E}}\left[-\nabla_{W_K}\log\left(\mathbf{S}_y(\hat{y})\right)\right]$$

$$= \widehat{\mathbb{E}}\left[-\frac{\partial z_T}{\partial W_K}\frac{\partial \hat{y}}{\partial z_T}\nabla_{\hat{y}}\log(\mathbf{S}_y(\hat{y}))\right]$$

$$= \widehat{\mathbb{E}}\left[\sum_{t=1}^{T}\left(A_{t-1}^{\top}\otimes b_{t-1}\right)W_h^{\top}\left(\mathbf{S}(\hat{y}) - \mathbf{e}_y\right)\right],$$

where $A_{t-1} = W_V E_{t-1}E_{t-1}^{\top} \in \mathbb{R}^{d\times d}$, $b_{t-1} = W_Q z_{t-1} \in \mathbb{R}^d$.

**Step 4: Gradient with respect to the query matrix $W_Q$.** The derivation for $W_Q$ is similar to that for $W_K$.

Define $\widetilde{A}_{t-1} = W_V E_{t-1}E_{t-1}^{\top}W_K^{\top}$ and $\tilde{b}_{t-1} = z_{t-1}$ for each loop $t \in [T]$, then

$$\frac{\partial f_\theta(E_{t-1}, z_{t-1})}{\partial W_Q} = \tilde{b}_{t-1} \otimes \widetilde{A}_{t-1}^{\top}.$$

Thus

$$\frac{\partial z_T}{\partial W_Q} = \sum_{t=1}^{T}\left(\tilde{b}_{t-1}\otimes\widetilde{A}_{t-1}^{\top}\right) \in \mathbb{R}^{d^2\times d}.$$

Finally, applying the chain rule gives the gradient for $W_Q$:

$$\nabla_{W_Q}\widehat{L}(\theta) = \widehat{\mathbb{E}}\left[-\nabla_{W_Q}\log\left(\mathbf{S}_y(\hat{y})\right)\right]$$

$$= \widehat{\mathbb{E}}\left[-\frac{\partial z_T}{\partial W_Q}\frac{\partial \hat{y}}{\partial z_T}\nabla_{\hat{y}}\log(\mathbf{S}_y(\hat{y}))\right]$$

$$= \widehat{\mathbb{E}}\left[\sum_{t=1}^{T}\left(\tilde{b}_{t-1}\otimes\widetilde{A}_{t-1}^{\top}\right)W_h^{\top}\left(\mathbf{S}(\widehat{Y}) - \mathbf{e}_y\right)\right],$$

where $\widetilde{A}_{t-1} = W_V E_{t-1}E_{t-1}^{\top}W_K^{\top} \in \mathbb{R}^{d\times d}$, $\tilde{b} = z_{t-1} \in \mathbb{R}^d$. $\qquad\square$

### H.1.3 THE PRECONDITIONING EFFECT FOR *Looped-Attn*

In Lemma 1 and 2, we have derived the gradients for both *Single-Attn* and *Looped-Attn* models, we now directly compare them. This analysis reveals a crucial insight into the optimization dynamics of *Looped-Attn*.

**Lemma 3.** *Denote the empirical loss $\widehat{L}_1$ for Single-Attn and $\widehat{L}_2$ for Looped-Attn, then the gradient of the Looped-Attn model can be expressed as the preconditioned gradient of the Single-Attn model:*

$$\nabla_{W_K}\widehat{L}_2(\theta) = P_{W_K}\nabla_{W_K}\widehat{L}_1(\theta),$$

$$\nabla_{W_Q}\widehat{L}_2(\theta) = P_{W_Q}\nabla_{W_Q}\widehat{L}_1(\theta),$$

*where the preconditioners $P_{W_K}$ and $P_{W_Q}$ are defined as:*

$$P_{W_K} = I + \widehat{\mathbb{E}}\left[P_2 P_1^{+}\right],$$

*with $P_1 = A^{\top}\otimes b$, $P_2 = \sum_{t=2}^{T}\left(A_{t-1}^{\top}\otimes b_{t-1}\right)$, $P_1^{+}P_1 = I$, and $P_1^{+}$ is the Moore-Penrose pseudoinverse.*

$$P_{W_Q} = I + \widehat{\mathbb{E}}\left[\widetilde{P}_2\widetilde{P}_1^{+}\right],$$

*with $\widetilde{P}_1 = \tilde{b}\otimes\widetilde{A}^{\top}$, $\widetilde{P}_2 = \sum_{t=2}^{T}\left(\tilde{b}_{t-1}\otimes\widetilde{A}_{t-1}^{\top}\right)$, $\widetilde{P}_1^{+}\widetilde{P}_1 = I$, and $\widetilde{P}_1^{+}$ is the Moore-Penrose pseudoinverse.*

**Remark 14.** Recall that in Lemma 1 and 2, we define: $A_{t-1} = W_V E_{t-1}E_{t-1}^{\top}$, $b_{t-1} = W_Q z_{t-1}$, $A = W_V E_0 E_0^{\top}$, and $b = W_Q z_0$; $\widetilde{A}_{t-1} = W_V E_{t-1}E_{t-1}^{\top}W_K^{\top}$, $\tilde{b}_{t-1} = z_{t-1}$, $\widetilde{A} = W_V E_0 E_0^{\top}W_K^{\top}$,

and $\tilde{b} = z_0$. This Lemma shows that the gradient of the *Looped-Attn* model can be expressed as the gradient of the *Single-Attn* model multiplied by a specific linear operator. The operator acts as a preconditioner, effectively using information from the iterative refinement steps to adjust the magnitude and direction of the base gradient calculated from a single attention pass.

In Lemma 3, the full-rank assumption for $E_0 E_0^\top$ holds during the early stages of training. At initialization, the input embeddings $E_0 = W_{\mathrm{emb}} X$ utilize the full representation space and have not collapsed into a low-dimensional intrinsic subspace. The rank deficiency typically arises in the late training stage due to feature collapse where the dimension $d$ exceeds the intrinsic dimension.

Lemma 4 discusses the rank-deficiency case in the late training stage, where the residual terms emerge due to feature collapse. Specifically, the terms $I - \Pi$ or $I - \widetilde{\Pi}$ represent the null-space of *Single-Attn* where the model fails to acquire gradient information. The lemma reveals that *Looped-Attn* retains access to these directions via the residual terms $\mathcal{R}_{W_K}$ and $\mathcal{R}_{W_Q}$. Consequently, the recursive operations $P_2$ and $\widetilde{P}_2$ process these null-space signals, effectively recovering information lost by the non-recursive model.

While Lemma 4 addresses the rank-deficiency case in the late training stage, we utilize Lemma 3 for the derivation of Theorem 4. This is because Theorem 4 investigates gradient alignment in the initial descent phase within the valley subspace.

*Proof.* We prove this lemma by direct algebraic computation, starting with the gradient with respect to the key matrix $W_K$.

**Derivation for the key matrix $W_K$.** Recall the expressions for the *Single-Attn* gradient ($\nabla_{W_K} \widehat{L}_1$) and the *Looped-Attn* gradient ($\nabla_{W_K} \widehat{L}_2$):

$$\nabla_{W_K} \widehat{L}_1(\theta) = \widehat{\mathbb{E}} \left[ (A^\top \otimes b) W_h^\top (\mathbf{S}(\hat{y}) - \mathbf{e}_y) \right],$$

$$\nabla_{W_K} \widehat{L}_2(\theta) = \widehat{\mathbb{E}} \left[ \sum_{t=1}^{T} \left( A_{t-1}^\top \otimes b_{t-1} \right) W_h^\top (\mathbf{S}(\hat{y}) - \mathbf{e}_y) \right].$$

The core of the proof is to decompose the summation in the *Looped-Attn* gradient. We separate the first term of the series (for $t = 1$) from the subsequent terms (for $t = 2$ to $T$):

$$\nabla_{W_K} \widehat{L}_2(\theta) = \widehat{\mathbb{E}} \left[ \left( A_0^\top \otimes b_0 \right) W_h^\top (\mathbf{S}(\hat{y}) - \mathbf{e}_y) + \sum_{t=2}^{T} \left( A_{t-1}^\top \otimes b_{t-1} \right) W_h^\top (\mathbf{S}(\hat{y}) - \mathbf{e}_y) \right].$$

By the definitions, we have $A_0 = A$, $b_0 = b$. With our assumption, $\delta_1 = \mathbf{1}_d$ and $\mathrm{Diag}(\delta_1) = I$. Thus, the first term is exactly the gradient of the *Single-Attn* model:

$$\nabla_{W_K} \widehat{L}_2(\theta) = \nabla_{W_K} \widehat{L}_1(\theta) + \widehat{\mathbb{E}} \left[ \sum_{t=2}^{T} \left( A_{t-1}^\top \otimes b_{t-1} \right) W_h^\top (\mathbf{S}(\hat{y}) - \mathbf{e}_y) \right].$$

Define $P_1 = A^\top \otimes b$, $P_2 = \sum_{t=2}^{T} \left( A_{t-1}^\top \otimes b_{t-1} \right)$, then we derive that

$$\nabla_{W_K} \widehat{L}_2(\theta) = \nabla_{W_K} \widehat{L}_1(\theta) + \widehat{\mathbb{E}} \left[ \sum_{t=2}^{T} \left( A_{t-1}^\top \otimes b_{t-1} \right) W_h^\top (\mathbf{S}(\widehat{Y}) - \mathbf{e}_y) \right]$$

$$= \nabla_{W_K} \widehat{L}_1(\theta) + \widehat{\mathbb{E}} \left[ P_2 P_1^+ P_1 W_h^\top (\mathbf{S}(\widehat{Y}) - \mathbf{e}_y) \right]$$

$$= \nabla_{W_K} \widehat{L}_1(\theta) + \widehat{\mathbb{E}} \left[ P_2 P_1^+ \right] \nabla_{W_K} \widehat{L}_1(\theta)$$

$$= \left( I + \widehat{\mathbb{E}} \left[ P_2 P_1^+ \right] \right) \nabla_{W_K} \widehat{L}_1(\theta).$$

where $P_1^+ P_1 = I$, $P_1^+$ is the Moore-Penrose pseudoinverse with $b \neq \mathbf{0}$, $\mathrm{rank}(A^\top) = d$.

We can therefore identify the preconditioner for $W_K$ as:

$$P_{W_K} = I + \widehat{\mathbb{E}} \left[ P_2 P_1^+ \right].$$

**Derivation for the query matrix $W_Q$.** The derivation for the query matrix $W_Q$ follows an identical procedure. We begin by stating the gradients:

$$\nabla_{W_Q}\widehat{L}_1(\theta) = \widehat{\mathbb{E}}\left[(\tilde{b}\otimes\widetilde{A}^\top)W_h^\top(\mathsf{S}(\hat{y})-\mathbf{e}_y)\right],$$

$$\nabla_{W_Q}\widehat{L}_2(\theta) = \widehat{\mathbb{E}}\left[\sum_{t=1}^T\left(\tilde{b}_{t-1}\otimes\widetilde{A}_{t-1}^\top\right)W_h^\top(\mathsf{S}(\hat{y})-\mathbf{e}_y)\right].$$

Again, we split the summation and identify the first term as the *Single-Attn* gradient:

$$\nabla_{W_Q}\widehat{L}_2(\theta) = \widehat{\mathbb{E}}\left[\left(\tilde{b}_0\otimes\widetilde{A}_0^\top\right)W_h^\top(\mathsf{S}(\hat{y})-\mathbf{e}_y)+\sum_{t=2}^T\left(\tilde{b}_{t-1}\otimes\widetilde{A}_{t-1}^\top\right)W_h^\top(\mathsf{S}(\hat{y})-\mathbf{e}_y)\right]$$

$$= \nabla_{W_Q}\widehat{L}_1(\theta) + \widehat{\mathbb{E}}\left[\sum_{t=2}^T\left(\tilde{b}_{t-1}\otimes\widetilde{A}_{t-1}^\top\right)W_h^\top(\mathsf{S}(\hat{y})-\mathbf{e}_y)\right].$$

Define $\widetilde{P}_1 = \tilde{b}\otimes\widetilde{A}^\top$, $\widetilde{P}_2 = \sum_{t=2}^T\left(\tilde{b}_{t-1}\otimes\widetilde{A}_{t-1}^\top\right)$, then we derive that

$$\nabla_{W_Q}\widehat{L}_2(\theta) = \nabla_{W_Q}\widehat{L}_1(\theta) + \widehat{\mathbb{E}}\left[\sum_{t=2}^T\left(\tilde{b}_{t-1}\otimes\widetilde{A}_{t-1}^\top\right)W_h^\top(\mathsf{S}(\widehat{Y})-\mathbf{e}_y)\right]$$

$$= \nabla_{W_Q}\widehat{L}_1(\theta) + \widehat{\mathbb{E}}\left[\widetilde{P}_2\widetilde{P}_1^+\widetilde{P}_1 W_h^\top(\mathsf{S}(\widehat{Y})-\mathbf{e}_y)\right]$$

$$= \nabla_{W_Q}\widehat{L}_1(\theta) + \widehat{\mathbb{E}}\left[\widetilde{P}_2\widetilde{P}_1^+\right]\nabla_{W_Q}\widehat{L}_1(\theta)$$

$$= \left(I + \widehat{\mathbb{E}}\left[\widetilde{P}_2\widetilde{P}_1^+\right]\right)\nabla_{W_Q}\widehat{L}_1(\theta),$$

where $\widetilde{P}_1^+\widetilde{P}_1 = I$, $\widetilde{P}_1^+$ is the Moore-Penrose pseudoinverse with $\tilde{b}\neq\mathbf{0}$, $\mathrm{rank}(\widetilde{A}^\top) = d$.

We can therefore identify the preconditioner for $W_K$ as:

$$P_{W_Q} = I + \widehat{\mathbb{E}}\left[\widetilde{P}_2\widetilde{P}_1^+\right].$$

This completes the proof, demonstrating that the iterative updates in *Looped-Attn* introduce a pre-conditioning term to the standard single-pass attention gradient. $\qquad\square$

**Lemma 4.** *Denote the empirical loss $\widehat{L}_1$ for Single-Attn and $\widehat{L}_2$ for Looped-Attn, then the gradient of the Looped-Attn model can be expressed as the preconditioned gradient of the Single-Attn model in addition with a residual term:*

$$\nabla_{W_K}\widehat{L}_2(\theta) = P_{W_K}\nabla_{W_K}\widehat{L}_1(\theta) + \mathcal{R}_{W_K},$$

$$\nabla_{W_Q}\widehat{L}_2(\theta) = P_{W_Q}\nabla_{W_Q}\widehat{L}_1(\theta) + \mathcal{R}_{W_Q},$$

*where the preconditioners $P_{W_K}$, $P_{W_Q}$ and residual terms $\mathcal{R}_{W_K}$, $\mathcal{R}_{W_Q}$ are defined as:*

$$P_{W_K} = I + \widehat{\mathbb{E}}\left[P_2 P_1^+\right], \mathcal{R}_{W_K} = \widehat{\mathbb{E}}\left[P_2(I-\Pi)W_h^\top(\mathsf{S}(\widehat{Y})-\mathbf{e}_y)\right].$$

*with $P_1 = A^\top\otimes b$, $P_2 = \sum_{t=2}^T\left(A_{t-1}^\top\otimes b_{t-1}\right)$, $P_1^+P_1 \triangleq \Pi$, and $P_1^+$ is the Moore-Penrose pseudoinverse.*

$$P_{W_Q} = I + \widehat{\mathbb{E}}\left[\widetilde{P}_2\widetilde{P}_1^+\right], \mathcal{R}_{W_Q} = \widehat{\mathbb{E}}\left[\widetilde{P}_2(I-\widetilde{\Pi})W_h^\top(\mathsf{S}(\widehat{Y})-\mathbf{e}_y)\right].$$

*with $\widetilde{P}_1 = \tilde{b}\otimes\widetilde{A}^\top$, $\widetilde{P}_2 = \sum_{t=2}^T\left(\tilde{b}_{t-1}\otimes\widetilde{A}_{t-1}^\top\right)$, $\widetilde{P}_1^+\widetilde{P}_1 \triangleq \widetilde{\Pi}$, and $\widetilde{P}_1^+$ is the Moore-Penrose pseudoinverse.*

*Proof.* We proceed with the derivation for $W_K$ without assuming $P_1$ is full rank. Recall the decomposition of the summation:

$$\nabla_{W_K}\widehat{L}_2(\theta)$$

$$=\nabla_{W_K}\widehat{L}_1(\theta) + \widehat{\mathbb{E}}\left[\sum_{t=2}^{T}\left(A_{t-1}^{\top}\otimes b_{t-1}\right)W_h^{\top}(\mathsf{S}(\widehat{Y})-\mathbf{e}_y)\right]$$

$$=\nabla_{W_K}\widehat{L}_1(\theta) + \widehat{\mathbb{E}}\left[P_2(P_1^{+}P_1 + I - P_1^{+}P_1)W_h^{\top}(\mathsf{S}(\widehat{Y})-\mathbf{e}_y)\right]$$

$$=\nabla_{W_K}\widehat{L}_1(\theta) + \widehat{\mathbb{E}}\left[P_2 P_1^{+}P_1 W_h^{\top}(\mathsf{S}(\widehat{Y})-\mathbf{e}_y)\right] + \widehat{\mathbb{E}}\left[P_2(I - P_1^{+}P_1)W_h^{\top}(\mathsf{S}(\widehat{Y})-\mathbf{e}_y)\right]$$

$$=\nabla_{W_K}\widehat{L}_1(\theta) + \widehat{\mathbb{E}}\left[P_2 P_1^{+}\right]\nabla_{W_K}\widehat{L}_1(\theta) + \widehat{\mathbb{E}}\left[P_2(I - \Pi)W_h^{\top}(\mathsf{S}(\widehat{Y})-\mathbf{e}_y)\right]$$

$$=\left(I + \widehat{\mathbb{E}}\left[P_2 P_1^{+}\right]\right)\nabla_{W_K}\widehat{L}_1(\theta) + \mathcal{R}_{W_K},$$

where $P_1^{+}$ is the Moore-Penrose pseudoinverse and $P_1^{+}P_1 \triangleq \Pi$.

The derivation for the query matrix $W_Q$ follows an identical procedure:

$$\nabla_{W_Q}\widehat{L}_2(\theta)$$

$$=\nabla_{W_Q}\widehat{L}_1(\theta) + \widehat{\mathbb{E}}\left[\sum_{t=2}^{T}\left(\tilde{b}_{t-1}\otimes \widetilde{A}_{t-1}^{\top}\right)W_h^{\top}(\mathsf{S}(\widehat{Y})-\mathbf{e}_y)\right]$$

$$=\nabla_{W_Q}\widehat{L}_1(\theta) + \widehat{\mathbb{E}}\left[\widetilde{P}_2(\widetilde{P}_1^{+}\widetilde{P}_1 + I - \widetilde{P}_1^{+}\widetilde{P}_1)W_h^{\top}(\mathsf{S}(\widehat{Y})-\mathbf{e}_y)\right]$$

$$=\nabla_{W_Q}\widehat{L}_1(\theta) + \widehat{\mathbb{E}}\left[\widetilde{P}_2\widetilde{P}_1^{+}\widetilde{P}_1 W_h^{\top}(\mathsf{S}(\widehat{Y})-\mathbf{e}_y)\right] + \widehat{\mathbb{E}}\left[\widetilde{P}_2(I - \widetilde{P}_1^{+}\widetilde{P}_1)W_h^{\top}(\mathsf{S}(\widehat{Y})-\mathbf{e}_y)\right]$$

$$=\nabla_{W_Q}\widehat{L}_1(\theta) + \widehat{\mathbb{E}}\left[\widetilde{P}_2\widetilde{P}_1^{+}\right]\nabla_{W_Q}\widehat{L}_1(\theta) + \widehat{\mathbb{E}}\left[\widetilde{P}_2(I - \widetilde{\Pi})W_h^{\top}(\mathsf{S}(\widehat{Y})-\mathbf{e}_y)\right]$$

$$=\left(I + \widehat{\mathbb{E}}\left[\widetilde{P}_2\widetilde{P}_1^{+}\right]\right)\nabla_{W_Q}\widehat{L}_1(\theta) + \mathcal{R}_Q,$$

where $\widetilde{P}_1^{+}$ is the Moore-Penrose pseudoinverse and $\widetilde{P}_1^{+}\widetilde{P}_1 \triangleq \widetilde{\Pi}$.

Under the full-rank assumption ($\text{rank}(A) = d, b \neq 0$), $\Pi = I$, and the residual term $\mathcal{R}_{W_K}$ vanishes, recovering Lemma 3. $\qquad\square$

## H.2 PROOF FOR THEOREM 4

This section provides a formal analysis to demonstrate that the gradients of the *Single-Attn* and *Looped-Attn* models are positively aligned, a key theoretical foundation for the two-phase training strategy (SHIFT) proposed in our work. We establish this by proving that the inner product of the two gradient vectors is positive. This positive alignment ensures they point in a similar direction of descent. As both models make progress in the river direction during the initial phase of learning, this implies they explore a shared river upstream.

*Proof.* We begin by recalling the gradient expressions from Lemmas 1∼2, and the preconditioner relationship from Lemma 3. We have

$$\nabla_{W_K}\widehat{L}_1(\theta) = \widehat{\mathbb{E}}\left[(A^\top \otimes b)W_h^\top(\mathbb{S}(\hat{y}) - \mathbf{e}_y)\right],$$

$$\nabla_{W_K}\widehat{L}_2(\theta) = \widehat{\mathbb{E}}\left[\sum_{t=1}^{T}\left(A_{t-1}^\top \otimes b_{t-1}\right)W_h^\top(\mathbb{S}(\hat{y}) - \mathbf{e}_y)\right],$$

$$\nabla_{W_K}\widehat{L}_2(\theta) = P_{W_K}\nabla_{W_K}\widehat{L}_1(\theta).$$

We then analysis the directions of two gradients,

$$\langle\nabla_{W_K}\widehat{L}_1(\theta), \nabla_{W_K}\widehat{L}_2(\theta)\rangle = \mathrm{Tr}\left(\left(\nabla_{W_K}\widehat{L}_2(\theta)\right)^\top \nabla_{W_K}\widehat{L}_1(\theta)\right)$$

$$= \mathrm{Tr}\left(\left(P_{W_K}\nabla_{W_K}\widehat{L}_1(\theta)\right)^\top \nabla_{W_K}\widehat{L}_1(\theta)\right)$$

$$= \mathrm{Tr}\left(\nabla_{W_K}^\top\widehat{L}_1(\theta)P_{W_K}^\top\nabla_{W_K}\widehat{L}_1(\theta)\right).$$

The inner product is guaranteed to be non-negative if the matrix $P_{W_K}^\top$ is Positive Semidefinite (PSD), *i.e.*, $P_{W_K}^\top \succeq 0$. Our goal is to derive a set of sufficient conditions under which this holds.

From Lemma 3, we have

$$P_{W_K}^\top = I + \widehat{\mathbb{E}}[(P_1^+)^\top P_2^\top].$$

To ensure $P_{W_K}^\top \succeq 0$, we need to find conditions of $\widehat{\mathbb{E}}[(P_1^+)^\top P_2^\top] \succeq 0$. We analyze the term $(P_1^+)^\top P_2^\top$ for a single data sample. Using the properties of Kronecker products and pseudo-inverses, we have:

$$(P_1^+)^\top P_2^\top = (A^+ \otimes b^{+\top})\sum_{t=2}^{T}\left(A_{t-1} \otimes b_{t-1}^\top\right) = \sum_{t=2}^{T}(A^+ \otimes b^{+\top})\left[(A_{t-1} \otimes b_{t-1}^\top)\right].$$

To analyze this expression, we first establish recursive updates for $A_{t-1}$ and $b_{t-1}$.

**Recursive Updates of $A_{t-1}$.** The matrix $A_{t-1} = W_V E_{t-1}E_{t-1}^\top$ depends on the history of updates to the embedding matrix $E$. With $E_{t-1} = E_0 + \sum_{s=1}^{t-1}f(E_{s-1})$, we can write:

$$A_{t-1} = W_V E_{t-1}E_{t-1}^\top = W_V\left(E_0 + \sum_{s=1}^{t-1}f(E_{s-1})\right)\left(E_0 + \sum_{s=1}^{t-1}f(E_{s-1})\right)^\top$$

$$= W_V\left[E_0 E_0^\top + E_0\sum_{s=1}^{t-1}(f(E_{s-1}))^\top + \sum_{s=1}^{t-1}f(E_{s-1})E_0^\top + \sum_{s=1}^{t-1}\sum_{s'=1}^{t-1}f(E_{s-1})(f(E_{s'-1}))^\top\right]$$

$$= A + W_V\left[E_0\sum_{s=1}^{t-1}(f(E_{s-1}))^\top + \sum_{s=1}^{t-1}f(E_{s-1})E_0^\top + \sum_{s=1}^{t-1}\sum_{s'=1}^{t-1}f(E_{s-1})(f(E_{s'-1}))^\top\right].$$

We denote

$$\Delta A_{t-1} = W_V\left[E_0\sum_{s=1}^{t-1}(f(E_{s-1}))^\top + \sum_{s=1}^{t-1}f(E_{s-1})E_0^\top + \sum_{s=1}^{t-1}\sum_{s'=1}^{t-1}f(E_{s-1})(f(E_{s'-1}))^\top\right].$$

**Recursive Updates of $b_{t-1}$.** Similarly, the vector $b_{t-1} = W_Q z_{t-1}$ depends on the history of updates to the query vector $z$. With $z_{t-1} = z_0 + \sum_{s=1}^{t-1} f(E_{s-1}, z_{s-1})$, we can write:

$$b_{t-1} = W_Q z_{t-1} = W_Q \left( z_0 + \sum_{s=1}^{t-1} f(E_{s-1}, z_{s-1}) \right) = b + W_Q \sum_{s=1}^{t-1} f(E_{s-1}, z_{s-1}).$$

We denote $\Delta b_{t-1} = W_Q \sum_{s=1}^{t-1} f(E_{s-1}, z_{s-1})$.

**Substitute $A_{t-1}$ and $b_{t-1}$ into $(P_1^+)^\top P_2^\top$.** Let $A^+ = (W_V E_0 E_0^\top)^+$ and $b^{+\top} = (W_Q z_0)^{+\top}$. For each term in the summation ($t = 2$ to $T$), substitute $A_{t-1} = A + \Delta A_{t-1}$ and $b_{t-1} = b + \Delta b_{t-1}$, where $\Delta A_{t-1}$ and $\Delta b_{t-1}$ denote the recursive updates:

$$(A^+ \otimes b^{+\top})(A_{t-1} \otimes b_{t-1}^\top)$$
$$= (A^+ \otimes b^{+\top}) \left[ (A + \Delta A_{t-1}) \otimes (b + \Delta b_{t-1})^\top \right]$$
$$= (A^+ \otimes b^{+\top}) \left[ (A \otimes b^\top) + (A \otimes \Delta b_{t-1}^\top) + (\Delta A_{t-1} \otimes b^\top) + (\Delta A_{t-1} \otimes \Delta b_{t-1}^\top) \right].$$

For the first term,

$$(A^+ \otimes b^{+\top})(A \otimes b^\top) = (A^+ \otimes b^{+\top}) \sum_{k=1}^d e_k e_k^\top (A \otimes b^\top)$$
$$= \sum_{k=1}^d (A^+ \otimes b^{+\top}) e_k e_k^\top (A \otimes b^\top)$$
$$= \sum_{k=1}^d (A^+ \otimes b^{+\top})((e_k e_k^\top A) \otimes b^\top)$$
$$= \sum_{k=1}^d (A^+ e_k e_k^\top A) \otimes (b^{+\top} b^\top),$$

where $e_k = [0, \cdots, 1, \cdots, 0]^\top \in \mathbb{R}^d$, the $k$-th element is 1, and others is 0. $e_k e_k^\top (A \otimes b^\top)$ means that keeping the $k$-th row of matrix $A \otimes b^\top$ and others is 0. Similarly, $e_k e_k^\top A$ means that keeping the $k$-th row of matrix $A$, thus $e_k e_k^\top (A \otimes b^\top) = (e_k e_k^\top A) \otimes b^\top$.

$b$ is a vector and $b \neq \mathbf{0}$, then $b^+ = b^\top / b^\top b$,

$$(A^+ \otimes b^{+\top})(A \otimes b^\top) = \sum_{k=1}^d (A^+ e_k e_k^\top A) \otimes (b^{+\top} b^\top)$$
$$= (A^+ \sum_{k=1}^d e_k e_k^\top A) \otimes (b^{+\top} b^\top)$$
$$= (A^+ A) \otimes (b^{+\top} b^\top)$$
$$= \frac{1}{b^\top b} (A^+ A) \otimes (b b^\top).$$

For the second term,

$$(A^+ \otimes b^{+\top})(A \otimes \Delta b_{t-1}^\top) = (A^+ A) \otimes (b^{+\top} \Delta b_{t-1}^\top)$$
$$= \frac{1}{b^\top b} (A^+ A) \otimes (b \Delta b_{t-1}^\top).$$

For the third term,

$$(A^+ \otimes b^{+\top})(\Delta A_{t-1} \otimes b^\top) = (A^+ \Delta A_{t-1}) \otimes (b^{+\top} b^\top)$$
$$= \frac{1}{b^\top b} (A^+ \Delta A_{t-1}) \otimes (b b^\top).$$

For the fourth term,

$$(A^+ \otimes b^{+\top})(\Delta A_{t-1} \otimes \Delta b_{t-1}^\top) = (A^+ \Delta A_{t-1}) \otimes (b^{+\top} \Delta b_{t-1}^\top)$$
$$= \frac{1}{b^\top b}(A^+ \Delta A_{t-1}) \otimes (b \Delta b_{t-1}^\top).$$

Summarizing the decomposition for $(P_1^+)^\top P_2^\top$:

$$(P_1^+)^\top P_2^\top$$
$$= \sum_{t=2}^{T} [\text{Term1} + \text{Term2} + \text{Term3} + \text{Term4}]$$
$$= \frac{1}{b^\top b} \sum_{t=2}^{T} (A^+ A) \otimes (bb^\top) + (A^+ A) \otimes (b \Delta b_{t-1}^\top) + (A^+ \Delta A_{t-1}) \otimes (bb^\top) + (A^+ \Delta A_{t-1}) \otimes (b \Delta b_{t-1}^\top).$$

We now derive sufficient conditions for each term satisfies PSD.

**Term Analysis.** For Term1,

$$\text{Term1} = \frac{1}{b^\top b}(A^+ A) \otimes (bb^\top).$$

$bb^\top$ is a rank-1 PSD matrix. We also have $A^+ A = I \succeq 0$.

**For Term2**,

$$\text{Term2} = \frac{1}{b^\top b}(A^+ A) \otimes (b \Delta b_{t-1}^\top).$$

We have $A^+ A = I \succeq 0$. For $b \Delta b_{t-1}^\top$,

$$\Delta b_{t-1} = W_Q \sum_{s=1}^{t-1} f(E_{s-1}, z_{s-1})$$
$$f(E_{t-1}, z_{t-1}) = W_V E_{t-1} E_{t-1}^\top W_K^\top W_Q z_{t-1}$$
$$f(E_{t-1}) = W_V E_{t-1} E_{t-1}^\top W_K^\top W_Q E_{t-1}$$
$$E_{t-1} = E_0 + \sum_{s=1}^{t-1} f(E_{s-1})$$
$$z_{t-1} = z_0 + \sum_{s=1}^{t-1} f_\theta(E_{s-1}, z_{s-1}).$$

We need to prove there exists $\alpha \geq 0$ such that $\Delta b_{t-1} = \alpha b$.

Base Case: When $t = 2$, $s = 1$,

$$\Delta b_1 = W_Q f(E_0, z_0)$$
$$= W_Q(W_V E_0 E_0^\top W_K^\top W_Q z_0)$$
$$= W_Q(W_V E_0 E_0^\top W_K^\top b)$$
$$= (W_Q W_V E_0 E_0^\top W_K^\top)b$$
$$\triangleq \Phi_1 b,$$

where $E_0 E_0^\top \succeq 0$. With Assumption 5, $W_K, W_Q, W_V$ are approximately diagonal matrices,

$$W_K = D_K + \epsilon_K,$$
$$W_Q = D_Q + \epsilon_Q,$$
$$W_V = D_V + \epsilon_V,$$

where $D_K$, $D_Q$, $D_V$ are diagonal and $\epsilon_K$, $\epsilon_Q$, $\epsilon_V$ are dense matrices with extremely small elements. Thus we have

$$
\begin{aligned}
\Phi_1 &= W_Q W_V E_0 E_0^\top W_K^\top \\
&= (D_Q + \epsilon_Q)(D_V + \epsilon_V) E_0 E_0^\top (D_K + \epsilon_K) \\
&= (D_Q D_V + D_Q \epsilon_V + \epsilon_Q D_V + \epsilon_Q \epsilon_V) E_0 E_0^\top (D_K + \epsilon_K) \\
&= D_Q D_V E_0 E_0^\top D_K + \mathcal{O}(\epsilon_K, \epsilon_Q, \epsilon_V).
\end{aligned}
$$

With Assumption 6 ($D_A = D_Q D_V$, $D_B = D_k$, $P = E_0 E_0^\top$), we conclude that $\Phi_1$ is approximately PSD, and $\Delta b_1$ is co-directional with $b$.

Inductive Hypothesis: Assume that for $s = 1$ to $s = k-1$, $\Delta b_{k-1} = \Phi_{k-1} b$ where $\Phi_{k-1} \succeq 0$, i.e., $\Delta b_{k-1}$ is co-directional with $b$.

$$
\Delta b_{k-1} = W_Q \sum_{s=1}^{k-1} f(E_{s-1}, z_{s-1}) = \Phi_{k-1} b. \tag{16}
$$

Inductive Step: When $s = k$,

$$
\begin{aligned}
\Delta b_k &= W_Q \sum_{s=1}^{k} f(E_{s-1}, z_{s-1}) \\
&= W_Q \sum_{s=1}^{k-1} f(E_{s-1}, z_{s-1}) + W_Q f(E_{k-1}, z_{k-1}) \\
&= \Phi_{k-1} b + W_Q (W_V E_{k-1} E_{k-1}^\top W_K^\top W_Q z_{k-1}) \\
&= \Phi_{k-1} b + W_Q W_V E_{k-1} E_{k-1}^\top W_K^\top W_Q z_{k-1},
\end{aligned}
$$

where

$$
\begin{aligned}
z_{k-1} &= z_0 + \sum_{s=1}^{k-1} \delta_s \odot f_\theta(E_{s-1}, z_{s-1}) \\
&= W_Q^{-1} b + W_Q^{-1} \Phi_{k-1} b \\
&= W_Q^{-1}(I + \Phi_{k-1}) b,
\end{aligned}
$$

then $z_{k-1}$ is co-directional with $b$. Denote $\Phi_k' \triangleq W_Q \mathrm{Diag}(\delta_k) W_V E_{k-1} E_{k-1}^\top W_K^\top W_Q W_Q^{-1}(I + \Phi_{k-1})$, similarly with Assumption 5$\sim$6, we have $\Phi_k'$ is approximately PSD, and then

$$
\Delta b_k = \Phi_{k-1} b + \Phi_k' b.
$$

Thus, $\Delta b_k$ is co-directional with $b$.

Summary of Sufficient Condition: $W_K, W_Q, W_V$ are approximately diagonal matrices, $D_K$, $D_Q$, $D_V$ are PSD. These are summarized in Assumption 5$\sim$6.

**For Term3**,

$$
\text{Term3} = \frac{1}{b^\top b}(A^+ \Delta A_{t-1}) \otimes (bb^\top).
$$

$bb^\top$ is a rank-1 PSD matrix. We need to derive that the condition of $A^+ \Delta A_{t-1} \succeq 0$. With the definition of $\Delta A_{t-1}$,

$$
\begin{aligned}
\Delta A_{t-1} &= W_V \Big[ E_0 \sum_{s=1}^{t-1} (f(E_{s-1}))^\top + \sum_{s=1}^{t-1} f(E_{s-1}) E_0^\top + \sum_{s=1}^{t-1} \sum_{s'=1}^{t-1} f(E_{s-1})(f(E_{s'-1}))^\top \Big] \\
f(E_{t-1}) &= W_V E_{t-1} E_{t-1}^\top W_K^\top W_Q E_{t-1} \\
E_{t-1} &= E_0 + \sum_{s=1}^{t-1} f(E_{s-1}).
\end{aligned}
$$

We need to prove there exists $\Psi \succeq 0$ such that $\Delta A_{t-1} = \Psi A$, then $A^+ \Delta A_{t-1} \succeq 0$ can be derived.

Base Case: When $t = 2$, $s = 1$,

$$\Delta A_1 = W_V \left[ E_0(f(E_0))^\top + f(E_0)E_0^\top + f(E_0)(f(E_0))^\top \right].$$

(1) Substitute $A = W_V E_0 E_0^\top$ into $f(E_0) = W_V E_0 E_0^\top W_K^\top W_Q E_0$. Let $\Xi_1 \triangleq W_K^\top W_Q E_0$.

$$f(E_0) = W_V E_0 E_0^\top W_K^\top W_Q E_0 = A W_K^\top W_Q E_0 = A\Xi_1.$$

(2) Substitute

$$\Delta A_1$$

$$=W_V \left[ E_0(A\Xi_1)^\top + (A\Xi_1)E_0^\top + (A\Xi_1)(A\Xi_1)^\top \right]$$

$$=W_V E_0 \Xi_1^\top A^\top + W_V A\Xi_1 E_0^\top + W_V A\Xi_1 \Xi_1^\top A^\top$$

$$=W_V E_0 E_0^\top W_Q^\top W_K E_0 E_0^\top W_V^\top + W_V W_V E_0 E_0^\top W_K^\top W_Q E_0 E_0^\top + W_V A W_K^\top W_Q E_0 E_0^\top W_Q^\top W_K A^\top$$

$$= \underbrace{A W_Q^\top W_K A^\top}_{:T_1'} + \underbrace{W_V A W_K^\top W_Q W_V^{-1} A}_{:T_2'} + \underbrace{W_V A W_K^\top W_Q E_0 E_0^\top W_Q^\top W_K A^\top}_{:T_3'}$$

$$= \underbrace{A W_Q^\top W_K A^\top A^+ A}_{:T_1} + \underbrace{W_V A W_K^\top W_Q W_V^{-1} A}_{:T_2} + \underbrace{W_V A W_K^\top W_Q E_0 E_0^\top W_Q^\top W_K A^\top A^+ A}_{:T_3}$$

$$\triangleq \Psi_1 A,$$

where $A^+$ is the pseudoinverse matrix of $A$. Similarly with Assumption 5~6, when assuming that $W_K$, $W_Q$, $W_V$ are approximately diagonal matrices. $D_K, D_Q, D_V \succeq 0$, we have $\Delta A_1 = \Psi_1 A$ where $\Psi_1$ is approximately PSD.

Inductive Hypothesis: Assume that for $s = 1$ to $s = k - 1$, $\Delta A_{k-1} = \Psi_{k-1} A$ where $\Psi_{k-1} \succeq 0$.

$$\Delta A_{k-1} = W_V \Big[ E_0 \sum_{s=1}^{k-1} (\Delta_s \odot f(E_{s-1}))^\top + \sum_{s=1}^{k-1} (\Delta_s \odot f(E_{s-1})) E_0^\top$$

$$+ \sum_{s=1}^{k-1} \sum_{s'=1}^{k-1} (\Delta_s \odot f(E_{s-1}))(\Delta_{s'} \odot f(E_{s'-1}))^\top \Big] = \Psi_{k-1} A.$$

Inductive Step: When $s = k$,

$$\Delta A_k$$

$$= W_V \left[ E_0 \sum_{s=1}^{k} (f(E_{s-1}))^\top + \sum_{s=1}^{k} f(E_{s-1}) E_0^\top + \sum_{s=1}^{k} \sum_{s'=1}^{k} f(E_{s-1})(f(E_{s'-1}))^\top \right]$$

$$= \Psi_{k-1} A + W_V \left[ E_0(f(E_{k-1}))^\top + f(E_{k-1})E_0^\top + f(E_{k-1})(f(E_{k-1})^\top) \right]$$

$$= \Psi_{k-1} A + W_V E_0 f(E_{k-1})^\top + W_V f(E_{k-1}) E_0^\top + W_V f(E_{k-1}) f(E_{k-1})^\top$$

$$= \Psi_{k-1} A + W_V E_0 E_{k-1}^\top W_Q^\top W_K E_{k-1} E_{k-1}^\top W_V^\top + W_V W_V E_{k-1} E_{k-1}^\top W_K^\top W_Q E_{k-1} E_0^\top$$

$$\quad + W_V W_V E_{k-1} E_{k-1}^\top W_K^\top W_Q E_{k-1} E_{k-1}^\top W_Q^\top W_K E_{k-1} E_{k-1}^\top W_V^\top$$

$$= \Psi_{k-1} A + \underbrace{\left( W_V E_0 E_0^\top + W_V E_0 \Delta E_{k-1}^\top \right) W_Q^\top W_K E_{k-1} E_{k-1}^\top W_V^\top A^+ A}_{:M_1}$$

$$\quad + \underbrace{W_V W_V E_{k-1} E_{k-1}^\top W_K^\top W_Q \left( E_0 E_0^\top + \Delta E_{k-1} E_0^\top \right) A^+ A}_{:M_2}$$

$$\quad + \underbrace{W_V W_V E_{k-1} E_{k-1}^\top W_K^\top W_Q E_{k-1} E_{k-1}^\top W_Q^\top W_K E_{k-1} E_{k-1}^\top W_V^\top A^+ A}_{:M_3}$$

$$= \Psi_{k-1} A + M_1 A + M_2 A + M_3 A$$

$$= \Psi_k A,$$

where $E_{k-1} = E_0 + \sum_{s=1}^{k-1} f(E_{s-1})$, denote $\Delta E_{k-1} = \sum_{s=1}^{k-1} f(E_{s-1})$, similarly to $\Delta b = \Phi b$, we have $\Delta E_{k-1} = \Omega_{k-1} E_0$ and $\Omega_{k-1} \succeq 0$,

$$W_V E_0 E_{k-1}^\top = W_V E_0 E_0^\top + W_V E_0 \Delta E_{k-1}^\top,$$
$$E_{k-1} E_0^\top = E_0 E_0^\top + \Delta E_{k-1} E_0^\top.$$

Similarly to $\Delta A_1$, we have $\Psi_k = \Psi_{k-1} + M_1 + M_2 + M_3 \succeq 0$, thus we conclude that $\Delta A_k = \Psi_k A$.

Furthermore, using $\Delta A_k = \Psi_k A$ with $\Psi_k \succeq 0$, we then have $A^+ \Delta A \succeq 0$.

Summary of Sufficient Condition: $W_K$, $W_Q$, $W_V$ are approximately diagonal matrices, $D_K, D_Q, D_V \succeq 0$. These are summarized in Assumption 5$\sim$6.

**For Term4**,

$$\text{Term4} = \frac{1}{b^\top b} (A^+ \Delta A_{t-1}) \otimes (b \Delta b_{t-1}^\top).$$

Combining the analysis for Term2 and Term3, we need the conditions in Assumption 5$\sim$6.

Similarly to $W_K$, the conditions for preconditioner $P_{W_Q} \succeq 0$ are also Assumption 5$\sim$6.

Therefore, when with Assumption 5$\sim$6, the gradient updates on key and query matrices are co-directional between Single-Attn and Looped-Attn models:

$$\langle \nabla_{W_K} \widehat{L}_1(\theta), \nabla_{W_K} \widehat{L}_2(\theta) \rangle \geq 0, \quad \langle \nabla_{W_Q} \widehat{L}_1(\theta), \nabla_{W_Q} \widehat{L}_2(\theta) \rangle \geq 0.$$

$\square$

# I  USAGE OF LARGE LANGUAGE MODELS

In this work, we utilize Large Language Models (LLMs) for language polishing and grammar correction under our supervision. These suggestions are carefully reviewed and selectively adopted, ensuring consistency with our intended meaning and academic integrity. In addition, we use LLMs to generate the background visualizations for Figures 1(a)$\sim$1(b). The optimization trajectories presented in these figures are manually plotted by us.

