# OpenReview forum: "What Makes Looped Transformers Perform Better Than Non-Recursive Ones (Provably)"
_ICLR.cc/2026/Conference — ICLR 2026 Conference Withdrawn Submission_

### Official Review · Reviewer_p3UB · 2025-10-30

**Soundness:** 1
**Presentation:** 1
**Contribution:** 1
**Rating:** 0
**Confidence:** 4

**Summary:**

The paper studies looped versus non-looped (single) transformer architectures from a loss landscape perspective. Experiments on a synthetic Markov chain prediction task indicate that single attention stops learning after mastering simple patterns, while looped attention exhibits curriculum learning dynamics where the eigenspectrum goes through a 3-phase evolution. The paper conjectures that such inductive behaviors can be theoretically explained via a river-valley model. Moreover, the paper proposes a stage-wise framework that accelerates looped attention training.

**Strengths:**

The extended river-valley model may give some insights into empirically observed loss dynamics of attention architectures. The stage-wise training framework may also be of interest to practitioners.

**Weaknesses:**

The "theory" presented in the paper is unjustified and meaningless.

* While the paper claims to study single and looped attention architectures defined in Section 3, these model setups are never actually used in the analysis. After presenting some toy synthetic experiments with 1-layer attention on Markov chains, the authors simply assume each model's loss is equivalent to a U-valley and V-valley landscape (Conjecture 1,2) which satisfies a spectral condition (Assumption 1). None of these assumptions are rigorously justified even on the 1-layer linear attention model introduced in Section 3. Only empirical evidence on the aforementioned toy setup is given.
* Worse, the authors define an exact quadratic loss expression (Setting 1), then assume the single and looped attention losses satisfy this "setting" in their main results (Corollary 1,2). This is clearly false since the paper is studying cross-entropy next-token prediction loss with softmax output which is much more complicated than quadratic. Why are we considering this setting at all?
* Corollary 2 is also unjustified: even if Theorem 1 and Corollary 1 were correct, they only allow us to compare upper bounds $\mathcal{C}^{(1)},\mathcal{C}^{(2)}$ for $C_K^{(1)},C_K^{(2)}$. The argument that $\lVert C_K^{(i)}\rVert \approx \mathcal{C}^{(i)}$ given in the proof does not make sense.
* Theorem 2 seems to study a general loss function. However, the "proof" invokes a myriad of vague approximations (see the wording on p.34) and assumes global bounds of the Hessian over the entire training trajectory (Assumption 2,3 on p.28), which is completely unjustified, does not come with even an intuitive explanation and not even mentioned in the main text. This reduces the analysis to that of Theorem 1 and again does not give any insight into actual attention models.
* There is vague wording throughout the paper, both in the main body and proofs: e.g., "V-shaped valleys might convert hopping to significant progress along the river, which encourages to learn on the complex patterns"; "This matrix quantifies the critical interaction that allows movement in the valley to induce a gradient in the river"; "a U-shaped valley is characterized by a broad and flat floor through which the river flows. This valley is surrounded by uniformly steep cliffs, ensuring that movement in any direction within this subspace leads to a comparable loss. In contrast, a V-shaped valley is characterized by a narrow river channel, with cliffs of highly varied steepness." (What is being contrasted here?), etc.
* The overall logical structure (of lack thereof) and non-rigorous descriptive wording strongly suggest to me that the theoretical analysis was likely largely LLM-generated. It would be much better to leave out the theoretical claims completely and present the work as an experimental paper.

**Questions:**

See Weaknesses.

---

### Official Review · Reviewer_W45i · 2025-10-31

**Soundness:** 3
**Presentation:** 2
**Contribution:** 2
**Rating:** 4
**Confidence:** 4

**Summary:**

The paper argues that looped Transformers perform better than single‑pass ones because they have a different loss‑landscape geometry. The single‑pass models fall into broad, flat “U‑valleys,” whereas looped models create steep, “V‑valleys” that keep optimization moving along a low‑loss “river.” Empirically, looped models show a two‑phase learning pattern and distinctive Hessian dynamics on a synthetic Markov task, superior length generalization, and competitive results on algorithmic datasets. A block‑structured analysis explains why V‑valleys yield faster progress and lower loss, and the proposed SHIFT procedure, which is train single‑pass to a plateau, then switch to looped, preserves accuracy while reducing compute.

**Strengths:**

- The paper convincingly demonstrates through mechanistic studies on synthetic Markovian language tasks that Looped‑Attn exhibits two‑phase learning and three‑phase Hessian dynamics.
- Mathematical statements are rigorous, and the presented theory aligns well with the plots.
- The paper also provides a clear training recipe, with a simple criterion to get much of the looped model’s benefits at lower cost. This provides compute savings on synthetic sequence tasks, where SHIFT reduces training time vs Looped‑Attn across tasks with minimal accuracy loss.

**Weaknesses:**

- The paper’s theoretical setup for looped attention uses a simplistic linear‑attention block with additive updates. Whereas, in practice, a looped transformer iterates a full block and often adapts depth, hence the modeled dynamics deviate from actual looped architectures under these assumptions.

- The paper’s key claim that Looped‑Attn induces a V‑shaped valley and Single‑Attn a U‑shaped one remains a conjecture, supported only by toy‑model setting. Also, the title proposes “provably better” framing, yet this is conditional on the conjectured landscape and spectral dominance in Assumption 1. There is also no direct curvature analysis on deep nonlinear transformers to confirm this optimization shape distinctions.

- The mechanistic analyses of spectrum/entropy/MI are conducted on toy settings, the paper stops does not validate the similar behaviors in GPT‑2‑scale setups. Also, the non‑toy experiments target synthetic tasks only.

- While SHIFT shows, the performance depends on the hyperparameters which are thresholds $\delta_1,\delta_2$ and patience $W$, and they are not fully ablated/discussed in the paper.

- The captions for Figure 3, Figure 4 omit experimental setup details making them hard to interpret.

**Questions:**

I don't have any questions, please see weaknesses.

---

### Official Review · Reviewer_2jxE · 2025-10-31

**Soundness:** 1
**Presentation:** 2
**Contribution:** 2
**Rating:** 4
**Confidence:** 3

**Summary:**

The paper argues that looped transformers outperform standard single-pass transformers because recursion changes the loss landscape from a “River‑U‑Valley” (flat, trapping) geometry to a “River‑V‑Valley” (ill‑conditioned, hopping) geometry. The authors formalize the river/valley split by thresholding Hessian eigenvalues (Definition 1), posit two conjectures that map Single‑Attn to U‑valleys and Looped‑Attn to V‑valleys, and then prove that V‑valley geometry yields better optimization via a sequence of theorems. They also propose SHIFT, a two‑stage training procedure that switches from Single‑Attn to Looped‑Attn, and justify it with a “shared river upstream” result (gradient alignment) plus experiments on a Markov toy dataset and standard algorithmic tasks.

**Strengths:**

1. Clear narrative tying sample‑level behavior and Hessian dynamics to a geometric picture (Definition 1 and Conjectures 1–2). The figures and the matrix‑entropy/mutual‑information diagnostics make the empirical story easy to follow (Figure 3 and Figures 7–8).
2. The formal setup isolates a block‑structured Hessian and uses it to analyze valley/river coupling (Setting 1 and Definition 2 in Appendix F). This is a sensible abstraction for studying anisotropy.
3. SHIFT is a promising idea. On toy data it yields a measurable training speedup without sacrificing accuracy (Figure 4), and on algorithmic tasks it reduces wall‑clock time relative to a Looped‑Attn baseline (Figure 11).

**Weaknesses:**

1. “Formal characterization” of the landscape is largely conjectural: The paper “formally analyzes inductive bias” (Appendix A(b)), but the mapping from architectures to U‑ vs V‑valleys appears only as Conjecture 1 and Conjecture 2, backed by plots, not proofs. The formal part is the generic river/valley definition; the architectural link is not derived. The text explicitly labels these as conjectures with empirical justification (Section 4.2), and Appendix B acknowledges that proving the conjectures is left to future work. So the contribution is better described as a formalized hypothesis plus evidence, not a formal characterization.
2. “Provably superior performance” relies on strong and sometimes vague assumptions.
- Assumption 1 (Section 4.3 / Appendix F.1) presupposes that Looped‑Attn has (i) more low‑curvature eigenvalues and (ii) component‑wise smaller low eigenvalues than Single‑Attn. That is exactly the spectral dominance one would need to conclude Looped‑Attn is better. If you assume this degree of dominance, the later inequalities almost follow by construction. The “empirical justification” is a qualitative comparison of spectra (e.g., Figure 8(o) vs 7(j)), but the component‑wise dominance claim is not established. Your concern that it assumes too much is warranted.
- Theorem 1 defines an upper bound $C$ on the “cumulative force generated by the valley dynamics on the river subspace,” which is essentially a sum of $1/|\lambda_i|$ terms scaled by constants (Equation in Section 4.3 / Appendix F.2) The subsequent narrative states that this force is “dominated by the flattest directions” and that Looped‑Attn’s force is “significantly greater” than Single‑Attn’s ($>>$ in Corollary 1), but no quantitative constant factor or minimal gap is specified. The argument boils down to: if Looped‑Attn has more/smaller eigenvalues below a threshold, the sum of $1/\lambda$ is larger, which is true but tautological under Assumption 1, and stated non-quantitatively.
- Corollary 2 then claims lower training loss after $K$ steps for Looped‑Attn, but the proof sketch uses the first order Taylor approximation, without step-size conditions/lipschitz analysis, and treats second-order terms as negligible. This makes the conclusion sensitive to unlisted smoothness/learning-rate setup.
- Theorem 2 explicitly assumes the local forms $(\partial_{\theta_R} L(\theta_k)\approx H_{RV}(\theta_k)\theta_{V,k}-h_{R,k})$ and $(\partial_{\theta_V} L(\theta_k)\approx H_{Valley}(\theta_k)\theta_{V,k}+H_{VR}(\theta_k)\theta_{R,k})$, sets $(H_{River}\approx 0)$, and then further assumes the “unforced” part dominates (Appendix F.4). These are very strong assumptions: they are asserted after a Taylor expansion rather than proved to hold along the full training trajectory.
3. SHIFT’s “provable shared river upstream” depends on unrealistic structural assumptions: Theorem 3 proves gradient alignment for $W_K$ and $W_Q$ only if the attention weights are diagonally dominant with PSD diagonals and certain composite products are approximately PSD (Assumptions 4–5). Remark 6 concedes this is an idealization and not what we expect for trained transformers, offering only a heuristic that such alignment “may” hold early in training. This means the proof of “shared river upstream” rests on conditions unlikely to be met in practice, and covers only two of the three attention weight matrices (no result for $W_V$ or the head).
4. Additional proof‑level issues:
    - Vague definitions in key places.   U‑ vs V‑valley is defined via $\kappa(H_{Valley})\le 1+\delta$ vs $\kappa(H_{Valley})\gg 1$ with unspecified $\delta$ and no operational meaning for “$\gg$.” The river/valley split itself depends on a threshold $\epsilon$ without sensitivity analysis. This imprecision cascades into theorems that use those categories.
     - Neglecting residual terms.   The argument that the “residual” contribution from large eigenvalues is negligible (Appendix F.2) is asserted by intuition ($1/\lambda$ is small when $\lambda>\tau$) without bounding the   number   or   mass   of those eigenvalues. This is not a proof of negligibility.
      - Rank and pseudoinverse conditions.   Lemma 3’s preconditioner uses a Moore–Penrose pseudoinverse and assumes $\mathrm{rank}(A^\top)=d$ with $A=W_VE_0E_0^\top$. In realistic settings $E_0E_0^\top$ need not be full rank if (d) exceeds the intrinsic dimension of the embeddings, so the algebra depends on conditions that are not substantiated.
      - Ignoring $H_{River}$.   The general‑loss analysis sets $H_{River}\approx 0$ because the river is “extremely flat,” but the training path spends long stretches outside that idealized manifold, and no uniform upper bound on $|H_{River}|$ is provided to make the approximation justified
      - No non‑asymptotic control.   Claims like “cumulative force saturates as $K\to\infty$” are used to compare models at finite $K$ (e.g., choosing where their valley decays), but there are no   rates   or   step‑size   constraints that turn the asymptotic statements into practical guarantees.
5. Empirical scope vs theoretical claims: The experiments establish that Looped‑Attn can beat Single‑Attn on toy Markov sequences and algorithmic tasks and that SHIFT can be faster. But the paper’s   theory does not actually prove   that real, nonlinear looped transformers induce a V‑valley landscape; that link is conjectural. Even Appendix B states that extending the gradient‑alignment proof beyond the linearized toy setting is “beyond our current scope.” The “provably superior” phrasing in Appendix A(c) thus overstates what is established.

**Questions:**

1. Clarify the   quantitative thresholds   in your definitions. What concrete values of $\epsilon$, $\delta$, and the meaning of “$\gg$” separate U‑ from V‑valleys in your analyses? How sensitive are Theorem 1/2 conclusions to these choices?

2.   Assumption 1   is central. Can you provide   direct, per‑eigenvalue   evidence (not just histograms) that Looped‑Attn’s valley spectrum   component‑wise dominates   Single‑Attn’s in your experiments, and specify the threshold $\tau$ used? Also, can you show that this ordering persists across seeds and tasks?

3. In   Theorem 2  , you assume local linear forms for the river/valley gradients and dismiss $H_{River}$. Can you bound the   approximation error   of these assumptions along the optimization trajectory (e.g., Lipschitz constants, step sizes that make the Taylor remainder small)?

4. For   Theorem 3  , Assumptions 4–5 are acknowledged to be idealized. Can you empirically test them (diagonal dominance, “approx PSD” products) at different training stages and report quantitative deviations? Also, do you have any alignment result for $W_V$ or the output head, or for deeper stacks?

5.   Lemma 3   uses $\mathrm{rank}(A^\top)=d$. Under your experimental settings, is $E_0E_0^\top$ full rank? If not, how sensitive are the conclusions to rank deficiency, and can you replace the pseudoinverse step with a bound that does not require full rank?

6. The proofs repeatedly say the effect is “  dominated by the flattest directions  .” Can you turn this into an   explicit bound  , e.g., show that the small‑eigenvalue slice contributes at least a specified fraction of (C) under stated spectral conditions, instead of arguing informally that the residual is negligible?

7. SHIFT sometimes causes a   sharp accuracy drop at the switch   on practical tasks (Figures 12–16). How does this square with the claimed gradient alignment? Can you add ablations showing how the drop varies with the patience thresholds $\delta_1,\delta_2$ and with different stop criteria?

8. Your length‑generalization rationale is qualitative (downstream river implies flatter minima). Can you supply a   generalization bound   or a controlled synthetic where V‑valley geometry provably translates into lower error on longer sequences, beyond correlation?

---

### Note · Authors · 2025-11-25

**Comment:**

We sincerely thank the reviewers for taking the time to provide valuable feedback. Based on these suggestions, we have revised the manuscript to strengthen our work. While we have decided to withdraw the submission at this stage, we look forward to sharing the improved version with the community in the near future!

**Withdrawal Confirmation:**

I have read and agree with the venue's withdrawal policy on behalf of myself and my co-authors.